# Towards Second-Order Optimization in Learned Image Compression: Faster, Better, and More Deployable

## Abstract

Training learned image compression (LIC) models entails navigating a challenging optimization landscape defined by the fundamental trade-off between rate and distortion. Standard first-order optimizers, such as SGD and Adam, struggle with *gradient conflicts* arising from competing objectives, leading to slow convergence and suboptimal rate–distortion performance. In this work, we demonstrate that a simple switch to a second-order quasi-Newton optimizer, **SOAP**, dramatically improves both training efficiency and final performance across diverse LIC architectures. Our theoretical and empirical analyses reveal that SOAP's Newton preconditioning inherently resolves the intra-step and inter-step update conflicts intrinsic to the R–D objective, facilitating faster, more stable convergence. Beyond acceleration, we uncover a critical deployability benefit: SOAP-trained (non-diagonal) models exhibit significantly fewer activation and latent outliers. This improves entropy modeling and substantially enhances robustness to post-training quantization. Together, these results establish second-order optimization—achievable as a seamless drop-in replacement of the imported optimizer—as a powerful, practical tool for advancing the efficiency and real-world readiness of LICs. Code will be publicly available.

## 1 Introduction

Learned image compression (LIC) methods have attracted significant attention due to their impressive performance (He et al., 2022; Liu et al., 2023b; Li et al., 2024a; Feng et al., 2025; Jiang et al., 2023; Lu et al., 2025). Despite substantial advances, the *training dynamics* of LICs remain underexplored. The prevailing practice is straightforward: design a model, then train it with a rate–distortion (R–D) objective $\mathcal{L}_{\text{R-D}} = \mathbb{E}_{x \sim p_{\text{data}}} \left[ -\log_2 P(\hat{z}) + \lambda \, d(x, \hat{x}) \right]$, using a first-order optimizer (typically Adam (Kingma & Ba, 2014)). While this approach is generally effective, recent studies indicate that advanced LIC models converge slowly (demanding substantial GPU hours) (Li et al., 2025; Zhang et al., 2025b), and that the standard framework fails to address *gradient conflicts* between the rate and distortion terms, leading to suboptimal performance (Zhang et al., 2025c).

Li et al. (2025) attribute the slow convergence to challenges in learning energy compaction, proposing an *auxiliary transform* (AuxT) to facilitate feature decorrelation and energy compression, reducing training time by 47% without sacrificing performance. However, this approach slightly adds parameters and increases computational cost (GMACs), introducing additional development complexity. Concurrently, Zhang et al. (2025b) explore the low-dimensional hypothesis in LIC (CMD-LIC) by decomposing model parameters based on correlations. They progressively reduce trainable parameters based on stable affine coefficients to accelerate training, yielding a 40% acceleration. However, this approach requires tuning many hyperparameters, and poor choices can severely degrade performance. Additionally, Zhang et al. (2025c) explicitly address rate and distortion gradient conflicts by formulating a saddle-point problem and adaptively reweighting each gradient component (Balanced R-D), achieving a $-2\%$ BD-Rate improvement but incurs a substantial increase in per-step training time and high sensitivity to hyperparameter settings for advanced LIC models.

In summary, existing training strategies often: (i) increase model development complexity, (ii) rely on fragile hyperparameter tuning, or (iii) introduce non-trivial modifications to the training pipeline—limiting their practicality (a drop-in replacement is preferred).

This raises a natural question:

> **(Q)** Can we accelerate LIC training and mitigate gradient conflicts *without* sophisticated problem reformulation, training pipeline revision, additional architectural changes, or added development overhead?

The answer is *yes*. In this work, we demonstrate that adopting a recent efficient second-order quasi-Newton optimizer, **SOAP** (Vyas et al., 2024), addresses both challenges simultaneously via a seamless drop-in replacement of the imported optimizer in the standard training pipeline. Across four top-performing LIC models—ELIC (He et al., 2022), TCM (Liu et al., 2023b), LALIC (Feng et al., 2025), and DCAE (Lu et al., 2025)—SOAP delivers an average **70% reduction in training steps** and **57.7% reduction in wall-clock time** to achieve the same performance as Adam. Furthermore, when trained for an equal number of steps, SOAP-trained models achieve an average **3% BD-Rate improvement** over Adam baselines. Fig. 1 and 2 illustrate this accelerated and superior convergence. Moreover, we uncover an additional benefit of second-order optimization (non-diagonal optimizer) beyond fast convergence: SOAP-trained models exhibit *fewer outliers* in activation and latent spaces, making them more amenable to post-training quantization (PTQ) and thus easier to deploy on resource-constrained hardware.

Our contributions are summarized as follows:

- **Accelerated training with superior R–D performance:** We empirically demonstrate that SOAP substantially reduces both training steps and wall-clock time required to achieve the same performance, while simultaneously improving the final rate–distortion performance compared to first-order optimizers when trained for the same number of steps. (Sec. 3)

- **Gradient conflict resolution via Newton preconditioning:** We provide theoretical and empirical evidence that SOAP aligns gradients update from competing R–D loss terms and from consecutive update steps, enabling more effective optimization trajectories. (Sec. 4)

- **Practical deployability benefits:** We demonstrate that SOAP-trained (non-diagonal optimizer) models have significantly fewer feature outliers, which improves entropy modeling and substantially enhances robustness to post-training quantization. (Sec. 5)

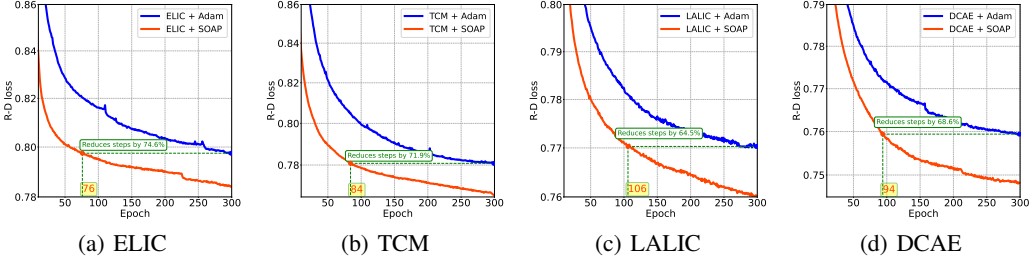

|  |  |  |  |
|---|---|---|---|
| (a) ELIC | (b) TCM | (c) LALIC | (d) DCAE |

Figure 1: **Comparison of Testing Loss: Epochs vs. R-D Loss for Various LICs.** First 10 epochs are omitted for better visualization. The SOAP optimizer demonstrates significantly faster convergence compared to Adam across multiple LICs. Evaluation is performed on the Kodak dataset with $\lambda = 0.013$; the R-D loss is computed as $\lambda \cdot 255^2 \cdot \text{MSE} + \text{Bpp}$.

## 2 PRELIMINARIES

### 2.1 RATE-DISTORTION IN LEARNED IMAGE COMPRESSION

Learned image compression seeks to efficiently encode an image $x$, sampled from a distribution $p_{\text{data}}(x)$, into a compact bitstream while minimizing the error in the reconstructed image

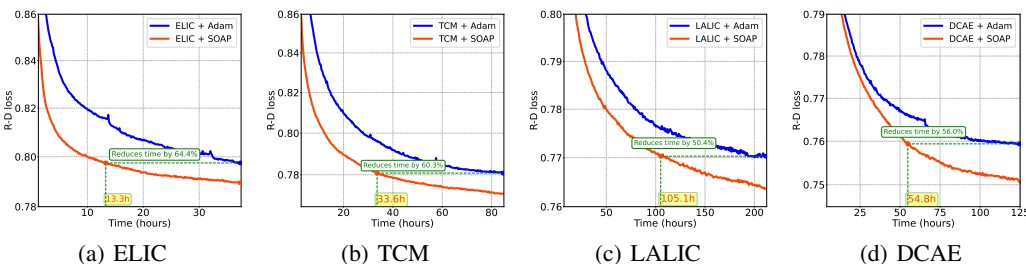

Figure 2: **Comparison of Testing Loss: Wall-Time vs. R-D Loss for Various LICs.** Training with the SOAP optimizer leads to much faster and more stable convergence than Adam when comparing wall-clock time. Results are measured on the Kodak dataset with $\lambda = 0.013$.

$\hat{x}$. This requires balancing two competing objectives: minimizing the bit rate (R) and minimizing the distortion (D). The standard transform coding architecture for LIC (Goyal, 2002; Ballé et al., 2018; Ballé et al., 2020) uses an encoder $e(\cdot)$, quantizer $Q(\cdot)$, and decoder $r(\cdot)$, such that $x \rightarrow \hat{z} = Q(e(x)) \rightarrow \hat{x} = r(\hat{z})$. The discrete latent $\hat{z}$ is compressed using entropy coding (Witten et al., 1987; Moffat, 2019), with an expected bit cost approximated by $-\log_2 P(\hat{z})$.

Training LICs involves minimizing the rate-distortion (R-D) loss:

$$\mathcal{L}_{\text{R-D}} = \mathbb{E}_{x \sim p_{\text{data}}} \left[ \underbrace{-\log_2 P(\hat{z})}_{\text{Rate}} + \lambda \underbrace{d(x, \hat{x})}_{\text{Distortion}} \right], \tag{1}$$

where $d(x, \hat{x})$ is a distortion metric (e.g., MSE, SSIM, LPIPS) and $\lambda$ controls the trade-off. Because the rate and distortion objectives often pull the model parameters in different directions, optimizing the R-D loss often leads to challenging gradient interactions and complex optimization dynamics.

## 2.2 Optimization Strategies: First-Order and Second-Order Methods

The choice of optimization algorithm profoundly impacts the efficiency and effectiveness of navigating the complex R-D loss landscape.

**First-Order Approaches.** First-order optimizers, such as SGD (Robbins & Monro, 1951) and Adam (Kingma & Ba, 2014), update parameters based on the gradient of the loss:

$$\theta_t \leftarrow \theta_{t-1} - \eta g_t. \tag{2}$$

While computationally efficient, these methods rely on local steepness and ignore the curvature of the loss landscape. Consequently, as we will demonstrate, they often struggle to resolve the competing gradients inherent in the R-D objective, leading to slow or unstable convergence.

**Second-Order Approaches.** In contrast, second-order optimization methods incorporate curvature information, aiming to better adapt parameter updates to the local geometry of the loss landscape. The classical Newton update (Boyd & Vandenberghe, 2004) is given by

$$\theta_t \leftarrow \theta_{t-1} - \eta H_t^{-1} g_t, \tag{3}$$

with $H_t$ denoting the Hessian matrix of second derivatives. In this paper, we demonstrate theoretically and empirically that such updates can help address gradient conflicts between the rate and distortion terms with effective descent directions.

**Main bottleneck.** The computational and memory demands of exact Newton steps are prohibitive for large neural networks, as the Hessian is an $n \times n$ matrix requiring $O(n^2)$ storage and at least $O(n^2)$ time to form, with inversion costing $O(n^3)$. To make second-order optimization tractable, practical algorithms such as Shampoo (Gupta et al., 2018; Eschenhagen et al., 2025; Morwani et al., 2024) approximate the Hessian inverse using structured preconditioners. SOAP (Vyas et al., 2024) extends this framework by introducing adaptive scaling reminiscent of Adam, but in the preconditioned space, resulting in an efficient quasi-Newton method suitable for deep models.

**Why Optimization Matters in LIC.** As we will demonstrate, the optimizer's ability to resolve the intrinsic gradient conflicts of the rate-distortion objective is key to effectively training a compressor. In particular, advanced second-order methods like SOAP can accelerate convergence, yield better rate-distortion results, and produce more stable representations—directly addressing both training efficiency and practical deployment challenges in LICs.

# 3 EMPIRICAL EVALUATION: ACCELERATING LIC TRAINING WITH SECOND-ORDER OPTIMIZATION

To empirically demonstrate the benefits of second-order optimization compared to first-order methods, we train several representative LICs using both Adam (Kingma & Ba, 2014) and SOAP (Vyas et al., 2024). A comparison with related training strategies, AuxT (Li et al., 2025), CMD-LIC (Zhang et al., 2025b), and Balanced-RD (Zhang et al., 2025c), is further provided in Appendix A.3.

**Evaluated Models.** We benchmark the following advanced LICs: **ELIC** (He et al., 2022), which incorporates unevenly grouped space-channel context models and stacked residual blocks; **TCM** (Liu et al., 2023b), which employs Transformer-CNN Mixture blocks to integrate both local and non-local information; **LALIC** (Feng et al., 2025), which utilizes Bi-RWKV blocks with linear attention; and **DCAE** (Lu et al., 2025), which adopts a dictionary-based cross-attention entropy model.

**Training Protocol.** All models are trained on the COCO 2017 dataset (Lin et al., 2014) using random $256 \times 256$ crops. Following CompressAI (Bégaint et al., 2020), we set $\lambda$ to $\{18, 35, 67, 130, 250, 483\} \times 10^{-4}$. EMA (Morales-Brotons et al., 2024) (decay=0.999) is enabled.

For both "+ Adam" and "+ SOAP" experiments, we use a batch size of 64 and an initial lr of $2 \times 10^{-4}$ with a ReduceLROnPlateau scheduler (patience 10, factor 0.5). Weight decay is set to 0, as no noticeable improvement is observed when it is applied (Sec. A.5). For SOAP, the preconditioner is updated every 10 steps following the default implementation (Sec. A.6). All models are trained for 300 epochs to ensure full convergence. These choices (lr, scheduler, update frequency, and other hyperparameters) follow standard defaults adopted in prior LIC and optimization studies (He et al., 2022; Liu et al., 2023b; Feng et al., 2025; Lu et al., 2025; Vyas et al., 2024).

**Evaluation Datasets.** Performance is evaluated on three widely used benchmarks: Kodak[1] ($768 \times 512$), Tecnick[2] ($1200 \times 1200$), and CLIC 2022[3] ($2048 \times 1365$).

**Evaluation Metrics.** We compare Adam and SOAP using the following criteria: **Steps-to-Adam** measures *step efficiency*—the ratio of training steps required by an optimizer to reach a target validation loss, normalized by the number required by Adam. Values less than 1.0 indicate superior step efficiency. **Time-to-Adam** assesses *wall-clock efficiency*—the ratio of training time to reach a target validation loss, relative to Adam. Values less than 1.0 reflect faster training in practice, accounting for per-step computational overhead. **BD-Rate after Convergence** reports the BD-Rate (Bjøntegaard, 2001), which quantifies average bitrate savings at matched image quality between models after full convergence, using the corresponding Adam-trained model as the anchor—a lower BD-Rate indicates better compression performance.

Please note that the additional VRAM overhead of SOAP relative to Adam is negligible (about a 1% increase in our setting) and is therefore not reported separately.

**Empirical Results.** Across all evaluated LIC architectures, as shown in Table 1, SOAP substantially accelerates convergence compared to Adam, both in terms of *step efficiency* and *wall-clock efficiency*. For instance, ELIC trained with SOAP reaches the target validation loss in only 25% of the steps and 35% of the time required by Adam, while TCM-S exhibits similar gains—requiring 28% of the steps and 39% of the time. These trends hold consistently for more complex and advanced LICs: LALIC and DCAE, where SOAP reduces training time by roughly 51–56% relative to Adam. Although each SOAP step incurs a slightly longer time cost, the drastic reduction in the number of steps leads to a net decrease in total training time. Figure A.2 illustrates and discusses the R-D curves for all methods.

---

[1] https://r0k.us/graphics/kodak/
[2] https://tecnick.com/?aiocp%20dp=testimages
[3] http://compression.cc/

Table 1: Computational Complexity and BD-Rate Compared to Adam

| Method | | Steps-to-Adam ↓ | Time-to-Adam ↓ | BD-Rate (%) ↓ | | | |
|---|---|---|---|---|---|---|---|
| | | | | Kodak | Tecnick | CLIC2022 | Avg. |
| ELIC (He et al., 2022) | + Adam | 1 | 1 | 0% | 0% | 0% | 0% |
| | + SOAP | **0.25** | **0.35** | **-3.49%** | **-3.52%** | **-4.01%** | **-3.67%** |
| TCM-S (Liu et al., 2023b) | + Adam | 1 | 1 | 0% | 0% | 0% | 0% |
| | + SOAP | **0.28** | **0.39** | **-2.86%** | **-2.40%** | **-3.01%** | **-2.76%** |
| LALIC (Feng et al., 2025) | + Adam | 1 | 1 | 0% | 0% | 0% | 0% |
| | + SOAP | **0.35** | **0.49** | **-2.44%** | **-3.31%** | **-3.51%** | **-3.09%** |
| DCAE (Lu et al., 2025) | + Adam | 1 | 1 | 0% | 0% | 0% | 0% |
| | + SOAP | **0.31** | **0.44** | **-2.26%** | **-2.03%** | **-2.06%** | **-2.12%** |

**Training Conditions**: $1 \times$ NVIDIA H100 GPU, $2 \times$ Intel Xeon Platinum 8480+ CPU, 1TB RAM. **Bold** indicates better performance. The "Avg." is the mean BD-Rate across Kodak, Tecnick, and CLIC2022.

Crucially, SOAP also achieves *better rate–distortion performance* after full convergence. On average across Kodak, Tecnick, and CLIC2022, SOAP improves BD-Rate by -3.67% for ELIC, -2.76% for TCM-S, -3.09% for LALIC, and -2.12% for DCAE relative to Adam. These improvements are consistent across datasets. Notably, for DCAE—which already achieves around -18% BD-Rate compared to VVC-intra—further improvement is especially meaningful. With SOAP, these gains are obtained without affecting the inference stage. Similarly, for smaller models such as TCM/ELIC, a 3% BD-Rate reduction is particularly impactful during development, further amplified by the enhanced robustness of SOAP-trained models to post-training quantization (see Section 5).

These results highlight a major advantage of SOAP: it can match Adam's final quality in less than half the training time across diverse LICs, while also delivering superior final R-D performance at the same steps. The benefits even extend to top-performing models such as DCAE and LALIC—where improving is notably challenging—suggesting that incorporating curvature information is especially valuable for navigating the complex optimization landscapes of LIC models.

## 4 NEWTON PRECONDITIONING ALIGNS CONFLICTING GRADIENTS IN RATE–DISTORTION OPTIMIZATION

We hypothesize that SOAP's empirical success stems from its ability to mitigate the inherent gradient conflicts in R-D optimization. First-order methods apply coordinate-wise rescaling to the gradient, leading to an inefficient compromise between rate and distortion objectives. In contrast, SOAP utilizes a second-order preconditioner that leverages curvature information to *rotate and scale* the gradient, producing an update vector that more effectively navigates the loss landscape.

In this section, we provide a theoretical analysis demonstrating how SOAP's Newton-like preconditioning resolves conflicts in two critical ways: (i) by aligning the rate and distortion update vectors within a single step (intra-step alignment), and (ii) by stabilizing the total update vector across consecutive steps (inter-step alignment). We then validate these theoretical insights empirically.

### 4.1 GRADIENT CONFLICT MEASUREMENT

Optimizing the R-D loss, $\mathcal{L}_{\text{R-D}} = \mathcal{L}_R + \lambda \mathcal{L}_D$, is fundamentally a multi-objective problem (Zhang et al., 2025c). Let $g_R = \nabla \mathcal{L}_R$ and $g_D = \nabla \mathcal{L}_D$ denote the *raw gradients*. Optimizers transform these gradients into *update vectors*; we denote the preconditioned update vectors corresponding to $g_R$ and $g_D$ as $p_R$ and $p_D$, respectively, and the total update vector as $p$.

Following Yu et al. (2020); Sener & Koltun (2018), we quantify conflict via cosine similarity:

$$\mathcal{S}(u, v) = \frac{\langle u, v \rangle}{\|u\| \, \|v\|} \in [-1, 1], \qquad (4)$$

for nonzero vectors $u, v$. We focus on two complementary metrics defined on the update vectors:

(a) **Inter-step score:** $\mathcal{S}_{\text{inter}}^t = \mathcal{S}(p^{t-1}, p^t)$ measures the consistency of the total update direction across consecutive steps, reflecting the stability of the optimization trajectory.

(b) **Intra-step score:** $\mathcal{S}_{\text{intra}}^t = \mathcal{S}(p_R^t, p_D^t)$ measures the alignment between the rate and distortion update vectors within step $t$.

## 4.2 GEOMETRIC INTUITION: WHY HIGHER COSINE ACCELERATES OPTIMIZATION

Before detailing the specific mechanism behind SOAP, it is crucial to understand intuitively *why* higher cosine similarity, both within a single update step and across consecutive steps, translates directly to the training acceleration observed. While we provide formal proofs linking cosine alignment to convergence in Appendix A.13, here we offer a geometric perspective on how "destructive interference" hampers standard optimizers and how "constructive alignment" resolves it.

**Intra-step: Resolving the Tug-of-War.** The total parameter update $p_t$ is effectively the vector sum of the preconditioned rate update $p_{R,t}$ and distortion update $p_{D,t}$. In first-order optimization (e.g., Adam), these vectors often point in divergent directions due to the competing nature of the R-D objective, creating a geometric tug-of-war. When $\mathcal{S}_{\text{intra}}^t$ is low or negative, significant portions of the gradient magnitudes are wasted as they cancel each other out; the optimizer burns computational energy pulling parameters in opposing directions while the net movement toward the Pareto frontier remains small. By identifying the curvature and rotating the optimization basis, SOAP aligns these update vectors ($\mathcal{S}_{\text{intra}}^t \approx 1$) so that they point towards a common descent direction. Geometrically, this ensures that the rate and distortion updates *constructively interfere*, effectively summing their magnitudes to take a larger, more efficient step.

**Inter-step: Straightening the Trajectory.** The efficiency is also determined by the path taken through the loss landscape. Complex rate-distortion landscapes are often characterized by narrow, curving valleys (ill-conditioned curvature) (Ma et al., 2022). First-order optimizers, unable to account for parameter correlations, typically oscillate across the walls of these valleys. An unstable inter-step cosine indicates this "zigzagging" behavior, where the update at step $t+1$ partially undoes the progress of step $t$. This results in a long, winding path to traverse a short Euclidean distance. In contrast, SOAP's Newton-style preconditioning aims to jump directly to the bottom of the local quadratic approximation. This linearizes the trajectory ($\mathcal{S}_{\text{inter}}^t \approx 1$), allowing the model to traverse the landscape along a smooth path, thereby requiring significantly fewer steps to reach convergence.

## 4.3 HOW SOAP RESOLVES GRADIENT CONFLICTS

**SOAP as a local Newton preconditioner.** While exact Newton updates are intractable for large models, SOAP efficiently approximates a quasi-Newton step (Vyas et al., 2024; Morwani et al., 2024). By utilizing the Kronecker-factored structure of the Gauss-Newton matrix, SOAP applies an Adam-style preconditioner within a rotated basis. As formally derived in Appendix A.7 (Theorem 1), this operation is equivalent to performing a local Newton step in the original parameter space:

$$p \approx -H^{-1}g \quad \text{(locally, under Assumptions (A1)–(A4)).} \tag{5}$$

This Newton-like behavior is key to resolving gradient conflicts.

**Inter-step alignment (Stability).** Newton preconditioning inherently stabilizes the optimization trajectory by adapting to local curvature. Consider the Newton-like update $p_t = -H_t^{-1}g_t$, where $H_t = \nabla^2 \mathcal{L}(\theta_t) \succ 0$, and parameters evolve as $\theta_{t+1} = \theta_t + \eta p_t$.

**Lemma 1** (Inter-step alignment for Newton). *If the Hessian varies smoothly (Lipschitz continuous), then the Newton direction changes very slowly between steps. Specifically, we show in Appendix A.8 that for $\eta$ sufficiently small,*

$$\left|1 - \mathcal{S}(p_t, p_{t+1})\right| \leq C_1\,\eta\,\|p_t\| + C_2\,\eta^2\,\|p_t\|^2, \tag{6}$$

*for constants $C_1, C_2$. In particular, $\mathcal{S}(p_t, p_{t+1}) \to 1$ as $\eta \to 0$ (or $\|p_t\| \to 0$).*

Lemma 1 guarantees that consecutive updates point in nearly the same direction, explaining the smooth, non-oscillatory progress observed when using SOAP.

**Intra-step alignment (Cooperation).** Beyond stabilizing the trajectory, SOAP also aligns the competing objectives within each step. Near a nondegenerate minimizer $\theta^*$, the component gradients linearize as $g_R \approx H_R(\theta - \theta^*)$ and $g_D \approx H_D(\theta - \theta^*)$ (Nocedal & Wright, 1999). Although the raw gradients $g_R$ and $g_D$ may point in different directions, they share the underlying curvature of the

model. If SOAP uses a shared preconditioner (due to combined loss) approximating the inverse of the Hessian $H$, and the component Hessians ($H_R$, $H_D$) share sufficient structure with $H$ (if they are locally proportional or jointly diagonalizable), the preconditioner effectively rotates both gradients toward the common solution $\theta^*$. We detail this justification in Appendix A.9, leading to:

**Proposition 1** (SOAP aligns component steps near the optimum). *Under the structural conditions described above (proof in Appendix A.9,*

$$\lim_{\theta \to \theta^*} \mathcal{S}(p_R, p_D) = 1. \tag{7}$$

Intuitively, the preconditioner ensures that both $p_R$ and $p_D$ point toward the optimum along $-(\theta - \theta^*)$ up to vanishing error, ensuring that R and D are optimized cooperatively rather than adversarially.

Together, these results highlight a central reason for SOAP's superiority: by aligning updates both across steps and between objectives, SOAP ensures that progress made in one iteration is not undone in the next, and that rate and distortion are optimized in a more cooperative rather than adversarial manner. This dual alignment reduces optimization inefficiency, avoids oscillatory behavior common in first-order methods, and leads to faster, more stable convergence with better final R-D tradeoffs.

**Why Adam struggles.** In contrast, Adam's fundamental limitation is its diagonal constraint. As we formalize in Appendix A.10 (Proposition 2), Adam locally approximates a diagonally preconditioned step: $p \propto -\operatorname{diag}(H)^{-1}g$. While this scales coordinates individually, it cannot utilize off-diagonal curvature information to *rotate* the update vector. Because the conflict between rate and distortion is rarely axis-aligned, diagonal scaling is insufficient to align the gradients. Our analysis in Appendix A.11 demonstrates that at initialization, this can lead to orthogonal updates, while Appendix A.12 shows that even near the optimum, Adam's updates can remain misaligned or adversarial (negative cosine similarity).

### 4.4 EMPIRICAL VALIDATION

To validate these theoretical predictions, we track the intra-step ($\mathcal{S}_{\text{intra}}^t$) and inter-step ($\mathcal{S}_{\text{inter}}^t$) scores for the ELIC model trained with Adam and SOAP. We initialize from a pretrained model to observe behavior near a local minimum, using a small learning rate ($1e{-}5$) and $\lambda = 0.013$.

The results, shown in Figure 3, strongly support our analysis:

- **SOAP achieves high alignment:** SOAP maintains consistently high positive alignment for both metrics. The inter-step score remains near $1.0$, indicating a highly stable trajectory, while the intra-step score remains strongly positive, indicating cooperative optimization of rate and distortion. This is consistent with the Newton-like behavior described in Lemma 1 and Proposition 1.

- **Adam exhibits significant conflict:** Adam shows low and highly oscillatory alignment. The intra-step score frequently dips toward $-1.0$ (strong opposition between rate and distortion updates), while the inter-step score fluctuates wildly around zero, indicating an unstable, inefficient trajectory. This confirms Adam's inability to resolve the inherent conflicts of the R-D objective characterized in Appendix A.10, A.11, and A.12.

These empirical findings substantiate our central claim: SOAP's performance gains arise from resolving both intra- and inter-step gradient conflicts in rate–distortion optimization.

## 5 SOAP SUPPRESSES LATENT AND ACTIVATION OUTLIERS

Beyond accelerating convergence and improving rate-distortion (R-D) performance, we observe a second advantage: SOAP reduces extreme values (*outliers*) in both latents and intermediate activations. Outlier suppression tightens entropy models and improves robustness to post-training quantization (PTQ), where large dynamic ranges are a primary failure mode (Bondarenko et al., 2021; Dettmers et al., 2022; Xiao et al., 2023a; Ashkboos et al., 2024; Nrusimha et al., 2024).

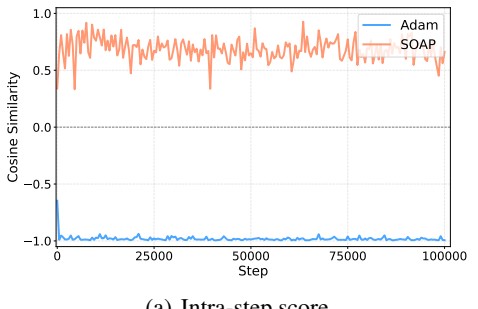 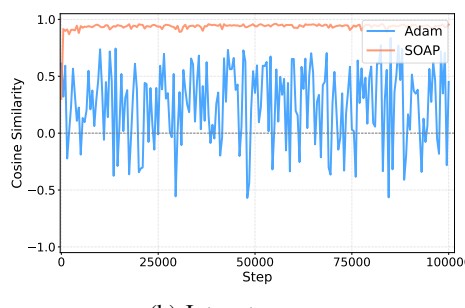

(a) Intra-step score            (b) Inter-step score

Figure 3: **Evolution of intra-step and inter-step gradient scores** for ELIC trained with Adam vs. SOAP. SOAP achieves high intra-step and inter-step scores, while Adam exhibits negative intra-step scores and oscillatory inter-step scores, highlighting SOAP's ability to suppress gradient conflicts.

## 5.1 OUTLIER MEASUREMENT

**Metrics.** Following prior work on neural feature analysis (Bondarenko et al., 2021; Elhage et al., 2023; He et al., 2024), we quantify outliers using two complementary, scale-invariant statistics. Let $\mathbf{X} \in \mathbb{R}^{n \times d}$ denote latents or activations, rescaled such that the second moment $m_2(\mathbf{X}) \triangleq \frac{1}{nd}\|\mathbf{X}\|_F^2 = 1$. We define the root mean square (RMS) per channel as $s_j = \sqrt{\frac{1}{n}\sum_{\alpha=1}^{n} X_{\alpha j}^2}$. We use:

$$\text{Kurt}(\mathbf{X}) = \frac{\frac{1}{d}\sum_{j=1}^{d} s_j^4}{\left(\frac{1}{d}\sum_{j=1}^{d} s_j^2\right)^2} \quad \text{and} \quad \text{MaxMed}(\mathbf{X}) = \frac{1}{n}\sum_{\alpha=1}^{n} \frac{\max_j |X_{\alpha j}|}{\text{median}_j |X_{\alpha j}|}. \tag{8}$$

$\text{Kurt}(\mathbf{X})$ measures the *tailedness* (heavy-tailed distributions imply more outliers) of channel energies, while $\text{MaxMed}(\mathbf{X})$ captures per-sample extreme values relative to typical magnitudes.

## 5.2 HOW SOAP SUPPRESSES OUTLIERS

The mechanism behind SOAP's outlier suppression lies in how its Newton-like updates interact with the underlying feature distributions during training.

**Newton preconditioning redistributes update energy.** SOAP applies a layerwise quasi-Newton step (Gupta et al., 2018; Anil et al., 2020; Vyas et al., 2024)

$$\Delta W = -\eta\, H_W^{-1} G, \tag{9}$$

where $G$ is the gradient and $H_W \succ 0$ is an SPD curvature proxy (Sec. A.7). In the eigenbasis $H_W = U\Lambda U^\top$, SOAP scales principal directions by $\Lambda^{-1}$ and rotates back via $U$, *coupling channels within a layer*. This rotation+rescaling compresses per-direction step size dispersion compared to diagonally preconditioned Adam/AdaFactor (Kingma & Ba, 2014; Shazeer & Stern, 2018), limiting runaway growth along isolated high-variance directions that otherwise produce outliers.

**A conserved-quantity view from signal propagation.** We can further understand this phenomenon through the lens of Signal Propagation theory (Schoenholz et al., 2016; Noci et al., 2022), which studies the input-wise Gram matrix ($\Sigma_I = \mathbf{X}\mathbf{X}^\top$) and how $\Sigma_I$ evolves in deep NNs. A key property is that the total energy of the feature correlations is conserved under rotation. Specifically, using the cyclicity of the trace ($\text{Tr}(\Sigma_F^2) = \text{Tr}(\Sigma_I^2)$) (Petersen et al., 2008), we derive an identity in Appendix A.14 that links feature kurtosis to cross-channel correlations:

$$\underbrace{n^2 d \cdot \text{Kurt}(\mathbf{X})}_{\text{Diagonal (Kurtosis)}} + \underbrace{\sum_{i \neq j} (\Sigma_F)_{ij}^2}_{\text{Off-Diagonal (Cross-Channel)}} = \underbrace{\sum_{\alpha,\beta} (\Sigma_I)_{\alpha\beta}^2}_{\text{Input Correlation Energy}}. \tag{10}$$

*Intuition.* The right-hand side, $\text{Tr}(\Sigma_I^2)$, measures the total "input correlation energy." When inputs are highly correlated, $\Sigma_I$ develops large off-diagonal entries, and this energy increases. Because the trace identity enforces conservation, the extra energy must manifest somewhere in the feature

statistics. A diagonal optimizer like Adam is inefficient at moving energy into the off-diagonal terms $((\Sigma_F)^2_{ij})$. Consequently, it forces the energy into the diagonal term, inflating the kurtosis and creating outliers. In contrast, SOAP rotates the basis, allowing it to redistribute this correlation energy into the off-diagonal terms, thereby keeping the kurtosis (and outliers) low. In essence: *Adam isolates outlier items, while SOAP diffuses variance across directions.*

**Small-step bound.** We can further quantify this by analyzing how the kurtosis grows during a single update step. Kurtosis is driven by the fourth moment ($L_4$ norm) of the parameter updates. Because Adam scales coordinates individually, it tends to produce axis-aligned updates that maximize this norm. SOAP, however, computes updates in a curvature-aligned eigenbasis and rotates them back, effectively "diffusing" the update energy across multiple physical channels. In Appendix A.14, we prove that the dominant second-order contribution to kurtosis growth for SOAP is upper-bounded:

$$\mathbb{E}[\Delta\text{Kurt}(\mathbf{X})]_{\text{SOAP}} \leq \mathbb{E}[\Delta\text{Kurt}(\mathbf{X})]_{\text{Diag}}. \tag{11}$$

This inequality (which holds up to negligible $O(\eta^3)$ terms) guarantees that diagonal optimizers represent the worst-case baseline for outlier generation. In non-diagonal landscapes, SOAP's rotational mixing ensures strictly lower growth.

## 5.3 EMPIRICAL VALIDATION

We measure Kurt($\mathbf{X}$) and MaxMed($\mathbf{X}$) for latents $z$, which is the feature after the last layer of the encoder, and feature activations[4] on Kodak, $\lambda = 0.013$. PTQ robustness is assessed via $\Delta$BD-Rate (%, lower is better) across all $\lambda$ for W8A8 (int8 weights and activations) quantization, using AdaRound (Nagel et al., 2020). Activation quantization is implemented as a non-learnable, dynamic channel-wise quantization approach that is applied on-the-fly during inference following (Shi et al., 2023). More specifically, for each channel independently, it computes the minimum and maximum values from the current activation data, then uses these to define an asymmetric 8-bit uniform quantization range where the zero-point equals the channel minimum and the scale factor is determined by the range divided by 255. The floating-point values are then quantized by subtracting the zero-point, dividing by the scale, rounding to the nearest integer, clamping to the 0-255 range, and finally dequantizing back by multiplying by the scale and adding the zero-point. Critically, this entire process is non-learnable as activation quantization serves as a fixed, statistical operation applied during each forward pass.[5] We also visualize the latent scaled deviation map (Xie et al., 2021; Feng et al., 2025) for the ELIC model between $\hat{y}$ and $y$ (Fig. 4), defined as $\varepsilon = \frac{|\hat{y}-y|}{\sum y}$, where lower values denote fewer outliers.

Table 2: Outlier metrics and PTQ robustness: Adam vs. SOAP. Metrics averaged on Kodak ($\lambda = 0.013$). PTQ robustness reported as $\Delta$BD-Rate (%); lower is better.

| Model + Optimizer | Latents | | Activations | | W8A8 PTQ |
| --- | --- | --- | --- | --- | --- |
| | Kurt($\mathbf{X}$) $\downarrow$ | MaxMed($\mathbf{X}$) $\downarrow$ | Kurt($\mathbf{X}$) $\downarrow$ | MaxMed($\mathbf{X}$) $\downarrow$ | $\Delta$BD-Rate $\downarrow$ |
| ELIC + Adam | 151.76 | 194.65 | 64.96 | 48.34 | 7.67% |
| ELIC + SOAP | **128.89** | **99.25** | **4.28** | **8.01** | **5.96%** |
| TCM + Adam | 127.99 | 182.32 | 12.26 | 18.27 | 7.75% |
| TCM + SOAP | **93.07** | **89.45** | **1.10** | **4.36** | **5.66%** |
| LALIC + Adam | 142.25 | 221.10 | 108.47 | 94.31 | 8.06% |
| LALIC + SOAP | **80.80** | **46.37** | **32.27** | **24.13** | **6.02%** |
| DCAE + Adam | 133.32 | 178.69 | 23.01 | 29.38 | 8.98% |
| DCAE + SOAP | **101.9** | **90.70** | **1.57** | **5.25** | **6.98%** |

Table 2 reports outlier metrics and PTQ robustness across four representative architectures. SOAP consistently yields substantially lower latent and activation kurtosis as well as reduced MaxMed($\mathbf{X}$) values compared to Adam. For example, on ELIC, SOAP reduces latent kurtosis from 151.76 to 128.89 and activation kurtosis from 64.96 to 4.28, yielding a nearly 2% BD-Rate gain under

---

[4]Randomly selected as the fourth layer at the encoder.

[5]We use AdaRound for illustration, following implementation at `https://github.com/Eric-qi/RDO-PTQ`; more advanced PTQ methods (Shi et al., 2023) are likely to yield even stronger results.

W8A8 quantization. Similar improvements hold across TCM, LALIC, and DCAE, demonstrating that SOAP's outlier suppression effect is architecture-agnostic. In the challenging W8A8 setting, quantization penalties consistently drop by about 2% BD-Rate across models.

Fig. 4 shows scaled deviation maps for the ELIC model. Under Adam, latents exhibit scattered extreme deviations (bright orange patches), reflecting concentrated outliers in a few positions. SOAP-trained latents, by contrast, display smoother and more uniform deviation maps with significantly lower peak values, directly corroborating the statistical improvements.

These empirical findings support the theoretical perspective in Sec. 5: By coupling channels via Newton-like scaling and rotations, SOAP redistributes variance across directions rather than concentrating it in a few, preventing outlier formation. This yields more regular feature statistics, improving entropy modeling and stabilizing activations, with the downstream benefit of enhanced PTQ robustness. Thus, SOAP not only accelerates training and improves R–D trade-offs but also produces models that are substantially easier to deploy on constrained hardware.

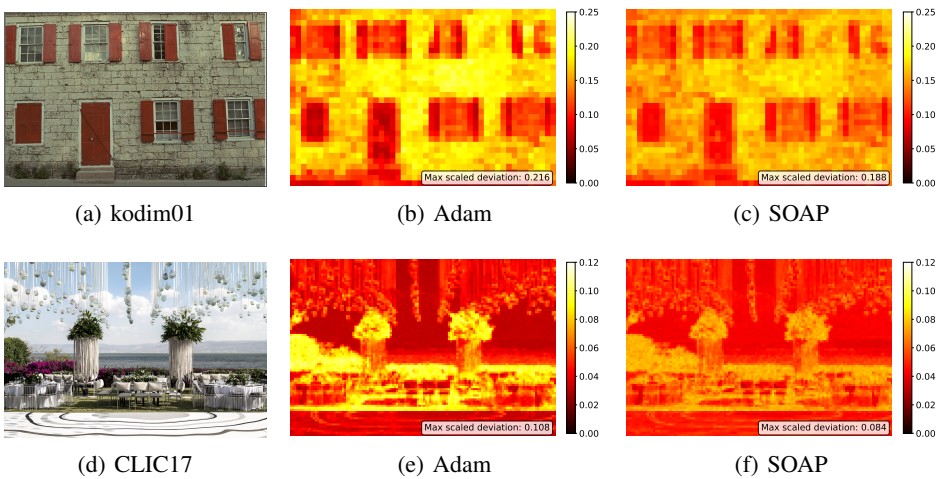

(a) kodim01    (b) Adam    (c) SOAP

(d) CLIC17    (e) Adam    (f) SOAP

Figure 4: **Scaled deviation maps for ELIC latent representations.** Each row shows the input image (left), latent scaled deviation with Adam (middle), and SOAP (right). SOAP consistently suppresses extreme values and yields lower maximum scaled deviation. *(Best viewed zoomed in.)*

## 6   CONCLUSION AND FUTURE WORK

In this work, we demonstrated that a simple two-line code modification yields faster training, as well as improved R-D performance across advanced LICs (ELIC, TCM, LALIC, and DCAE). Our theoretical and empirical analyses reveal that SOAP's Newton-style preconditioning effectively resolves the inherent gradient conflicts of the R-D objective by aligning updates both between the competing terms (intra-step) and across iterations (inter-step). Furthermore, we uncovered a critical practical benefit: SOAP-trained models exhibit significantly fewer activation and latent outliers, which enhances robustness to post-training quantization, making the models more deployable.

Looking forward, we identify several promising research directions: (i) developing hybrid optimization strategies that combine second-order information with complementary techniques (e.g., energy compression or feature decorrelation); (ii) extending second-order training to other domain compression methods, such as videos (Li et al., 2024b; Jia et al., 2025) and 3d representations (Wang et al., 2025a;b; Gao et al., 2025), where training costs (wall-time) are even higher; (iii) investigating adaptive R-D Hessian decomposition strategies to explicitly model and exploit the specific curvature interactions between rate and distortion terms; (iv) strengthening the theoretical foundations by relaxing assumptions (e.g., joint-diagonalization), quantifying curvature drift, and formally connecting outlier suppression to PTQ error bounds. We hope these results encourage the community to recognize optimization strategy as a critical pillar, alongside architecture and algorithm design, for advancing practical learned compression.

## 7 LLM USAGE DISCLOSURE

During the writing of this paper, we used GPT-5 to check and improve grammar and wording. No substantive content, research ideas, analysis, or results were generated by the model. We, as the authors, remain fully responsible for the accuracy and integrity of the work.

## 8 ETHICS STATEMENT

This work adheres to the ICLR Code of Ethics. No human subjects, personally identifiable data, or sensitive information were used in the research. All datasets employed are publicly available, properly licensed, and used in accordance with their intended purpose. We took care to verify dataset provenance and avoid unintended privacy or security risks.

We considered potential risks related to bias, fairness, and downstream misuse. While our methods are intended to advance scientific understanding, as with many machine learning techniques, they could potentially be applied in unintended ways. We encourage responsible use and further community evaluation of the societal impacts.

No conflicts of interest or external sponsorship influenced the research. The authors remain fully responsible for the integrity, accuracy, and transparency of the work.

## 9 REPRODUCIBILITY STATEMENT

We have made efforts to ensure reproducibility by providing detailed descriptions of training procedures, hyperparameters, and evaluation protocols in Sec. 3, 4, and 5 of the main paper, where dataset sources are also documented. Complete proofs of theoretical results are provided in Appendices A.7, A.8, A.9, A.10, A.11, A.12, and A.14 . To further support reproducibility, we will release the full implementation, including training and evaluation scripts, in an open-source repository upon acceptance of the paper.

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

# A APPENDIX

This appendix provides additional details regarding our methods, discussions, and comparisons.

## A.1 RELATED WORK

Learned image compression (LIC) methods are typically developed within the nonlinear transform coding framework (Ballé et al., 2020; Goyal, 2002), aiming to balance the compressed bit rate (R) and the reconstruction error (D).

**Advances in Transform Modules.** A wide variety of architectures have been explored to enhance the expressive power of encoder–decoder transforms. Examples include residual networks (He et al., 2022; Cheng et al., 2020), deformable convolutions (Fu et al., 2024a), designs based on frequency decomposition (Fu et al., 2024b; Ma et al., 2020), invertible neural networks (Xie et al., 2021; Cai et al., 2024), and contextual clustering (Qi et al.). Recently, transformers and Mamba architectures have gained traction, offering strong performance gains (Liu et al., 2023b; Zhu et al., 2022b; Zou et al., 2022; Koyuncu et al., 2022; Qian et al., 2022; Li et al., 2024a; Qin et al., 2024; Wu et al., 2025; Zeng et al., 2025; Feng et al., 2025; Lu et al., 2025).

**Entropy Modeling Improvements.** On the probabilistic side, research has sought to design more accurate entropy models for the latent space. This includes hierarchical priors (Ballé et al., 2018; Hu et al., 2020; Duan et al., 2023), autoregressive models operating over spatial (Minnen et al., 2018) or channel dimensions (Minnen & Singh, 2020), as well as hybrid models that capture joint spatial–channel dependencies (Jiang et al., 2023; Ma et al., 2021). Further refinements make use of checkerboard-based decoding (He et al., 2021), codebooks (Zhu et al., 2022a), and (lattice) vector quantization techniques (Zhang & Wu, 2023; Feng et al., 2023; Lei et al., 2024; Xu et al., 2025).

**Efficiency-Oriented Methods.** Another body of work seeks to reduce training and inference cost while maintaining compression quality. Notable examples include slimmable sub-networks (Tao et al., 2023), variable-bit-rate codecs (Guo-Hua et al., 2023; Kamisli et al., 2024), and knowledge distillation strategies (Fu et al., 2024a). Lightweight decoding has been pursued through shallow or linear decoders (Yang & Mandt, 2023), while new loss formulations such as causal context (Han et al., 2024) or latent decorrelation penalties (Ali et al., 2023) have also been proposed. Recent studies further introduce rate–distortion–complexity analysis (Minnen & Johnston, 2023; Gao et al., 2024), explicitly incorporating computational cost into the optimization objective.

**From Rate–Distortion to Multi-Objective Optimization.** Although rate–distortion training is often cast as minimizing a scalarized loss $R + \lambda D$, it is fundamentally a multi-objective problem: decreasing rate typically worsens distortion, and vice versa. This observation motivates the adoption of multi-objective optimization (MOO) techniques, which are designed to handle multiple conflicting criteria. One influential MOO approach is the Multiple Gradient Descent Algorithm (MGDA) (Désidéri, 2012; Sener & Koltun, 2018; Fliege et al., 2019). MGDA determines an update direction by combining gradients from different objectives with non-negative weights that minimize the squared norm of their sum, subject to a simplex constraint. The resulting direction guarantees improvement for all objectives simultaneously. MOO methods have seen wide adoption in multi-task learning (Yu et al., 2020; Liu et al., 2021; Momma et al., 2022; Navon et al., 2022; Zhou et al., 2022; Senushkin et al., 2023; Fernando et al., 2023; Liu et al., 2023a; Chen et al., 2023; Xiao et al., 2023b; Hu et al., 2024), where they are used to balance competing gradients across tasks and mitigate conflicts during training.

**Training Dynamics Approaches for LIC.** Building on these insights, recent studies in LIC have begun to focus on R-D optimization dynamics. Zhang et al. (2025c) introduced the Balanced-RD framework, which explicitly regularizes the interaction between rate and distortion gradients. Other

approaches, such as CMD-LIC (Zhang et al., 2025b) and Auxiliary Transform (AuxT) methods (Li et al., 2025), also reformulate training to accelerate convergence or stabilize optimization. Together, these works highlight optimization strategy as another pillar of LIC research, alongside architectural and entropy modeling advances.

**Our Perspective.** Our work aligns with this emerging line of training-dynamics-based methods and is closely related to Balanced-RD, CMD-LIC, and AuxT. Balanced-RD explicitly regulates the interaction between rate and distortion gradients to promote stable convergence in the R–D trade-off. CMD-LIC accelerates optimization by reducing training space dimensions. AuxT, on the other hand, introduces architectural constraints on energy compaction and feature decorrelation to improve convergence behavior. Distinct from these approaches, our study focuses on second-order optimization—specifically the SOAP method—which leverages curvature information (Newton-like update) to jointly accelerate convergence and reduce gradient conflicts. Beyond faster optimization, SOAP also suppresses activation and latent outliers, tightening entropy modeling and improving PTQ robustness, thereby enhancing both the stability and deployability of learned compressors.

## A.2 R-D FIGURES

Fig. 5 illustrates the R-D curves of all methods. The SOAP-trained models consistently outperform their Adam-trained counterparts, with the performance gap particularly pronounced in the challenging high-bpp regime.

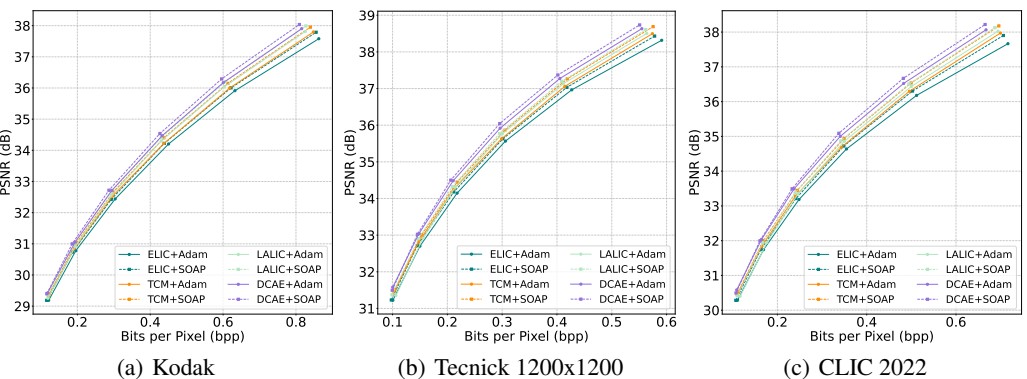

(a) Kodak      (b) Tecnick 1200x1200      (c) CLIC 2022

Figure 5: **R-D curves of various methods.** *Please zoom in for more details.*

## A.3 COMPARISON WITH OTHER METHODS

As discussed in Sec. 1 and A.1, SOAP can be compared against (i) *acceleration methods*, such as Auxiliary Transform (AuxT) (Li et al., 2025) and CMD-LIC (Zhang et al., 2025b); and (ii) *gradient-conflict mitigation methods*, such as Balanced-RD (Zhang et al., 2025c). Since the released Auxiliary Transform code[6] is implemented for TCM, we adopt the TCM model for fair comparison. Balanced-RD results are reproduced following the official implementation[7] ($\gamma$ values for Balanced-RD are swept to find the best results.), while CMD-LIC results are obtained from the authors of CMD-LIC. Additionally, to verify the additiveness of SOAP to other acceleration techniques, we further applied the SOAP to AuxT, termed as AuxT + SOAP. All the experiments follow the protocol in Sec. 3.

Table 3 and Fig. 6 reveal a clear trend: while existing acceleration methods (AuxT, CMD-LIC) and gradient-conflict mitigation (Balanced-RD) provide modest gains, SOAP consistently delivers stronger improvements in both convergence speed and final R-D performance. On TCM-S, SOAP alone reduces the number of steps and wall-clock time to reach Adam's performance by about 72% and 61%, respectively, compared to 51–57% step reductions for AuxT and CMD-LIC and even slower convergence for Balanced-RD. SOAP also outperforms Balanced-RD by more than 1% BD-Rate on average across Kodak, Tecnick, and CLIC2022. Moreover, combining SOAP with AuxT

---

[6]https://github.com/qingshi9974/AuxT
[7]https://gitlab.com/viper-purdue/balanced-rd

Table 3: Computational Complexity and BD-Rate Comparison on TCM-S

| Method | | Steps-to-Adam ↓ | Time-to-Adam ↓ | BD-Rate (%) ↓ | | | |
|---|---|---|---|---|---|---|---|
| | | | | Kodak | Tecnick | CLIC2022 | Avg. |
| TCM-S (Liu et al., 2023b) | + Adam | 1 | 1 | 0% | 0% | 0% | 0% |
| | + AuxT (Li et al., 2025) | 0.43 | 0.46 | -1.11% | -1.24% | -1.66% | -1.34% |
| | + CMD-LIC (Zhang et al., 2025b) | 0.49 | 0.50 | -0.47% | -0.55% | -0.68% | -0.57% |
| | + Balanced-RD (Zhang et al., 2025c) | 0.67 | 0.81 | -1.37% | -1.91% | -1.87% | -1.71% |
| | + SOAP | 0.28 | 0.39 | -2.86% | -2.40% | -3.01% | -2.76% |
| | + AuxT + SOAP | **0.23** | **0.35** | **-2.97%** | **-2.53%** | **-3.22%** | **-2.91%** |

**Training Conditions**: $1 \times$ NVIDIA H100 GPU, $2 \times$ Intel Xeon Platinum 8480+ CPU, 1TB RAM. **Bold** indicates the best performance. The "Avg." column reports the mean BD-Rate across Kodak, Tecnick, and CLIC2022.

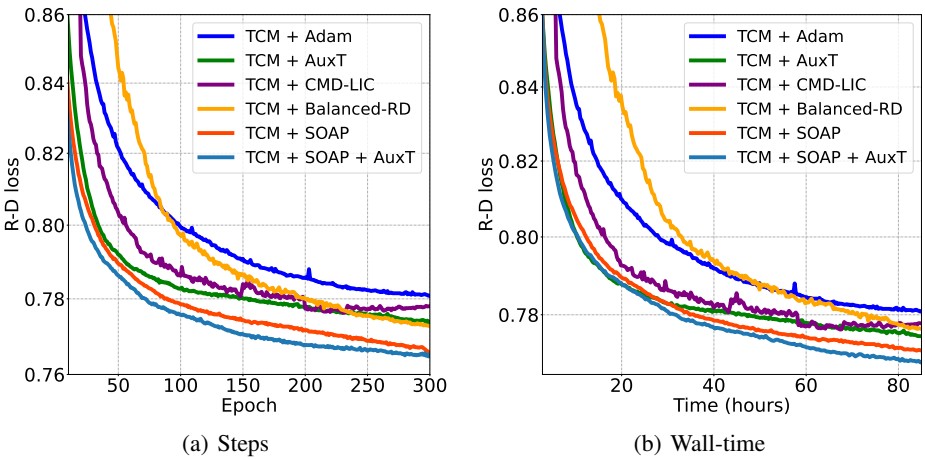

(a) Steps

(b) Wall-time

Figure 6: **Comparison of Testing Loss: Epochs/Wall-time vs. R-D Loss for Various LIC Methods.** *(Best viewed zoomed in.)* The first 10 epochs are omitted for better visualization. The SOAP optimizer demonstrates significantly faster convergence compared to Adam, AuxT, CMD-LIC, and Balanced-RD, and the AuxT + SOAP combination further accelerates convergence. SOAP not only accelerates training but also achieves a lower final R-D loss. Evaluation is performed on the Kodak dataset with $\lambda = 0.013$; the R-D loss is computed as $\lambda \cdot 255^2 \cdot \text{MSE} + \text{Bpp}$.

(AuxT + SOAP) yields the best overall performance, further reducing the steps- and time-to-Adam ratios to $0.23$ and $0.35$, and improving the average BD-Rate to $-2.91\%$. These results indicate that SOAP is not only effective on its own but also complementary to existing acceleration techniques. Unlike prior approaches, SOAP requires no auxiliary networks, progressive parameter freezing, loss reweighting, or extensive hyperparameter tuning, making it easy to integrate into existing training pipelines. Overall, the empirical evidence supports that incorporating second-order curvature information is a direct and effective way to accelerate training and mitigate gradient conflicts in learned image compression.

### A.4 A Preliminary Exploration for Learned Video Compression

Since our analysis is not closely constrained by image sources, we believe it is generally applicable to R-D problems, such as video compression. To further demonstrate the effectiveness and the generalization of SOAP and our analysis, we also performed a preliminary exploration on DCVC (Li et al., 2021).

#### A.4.1 DCVC

Since the DCVC training code is not open-sourced, we use an online reproduced version available at https://gitlab.com/viper-purdue/opendcvcs.

**Training Data:** We use the training partition of the Vimeo-90k septuplet dataset (Xue et al., 2019) as the source of training samples. During training, video sequences are randomly cropped into $256 \times 256$ patches.

**Testing Data:** For testing, we evaluate our models on benchmark datasets widely used in the video compression literature: HEVC Class B (Boyce et al., 2018); UVG (uvg, 2021); MCL-JCV (Wang et al., 2016).

**Test Conditions:** We test 96 frames for each video, and the intra period is set as 32. The low delay encoding setting is used. During both training and testing, all the frames are converted to the YUV444 color space by the ITU-R BT.709 transform matrix, and distortion loss is a weighted version in both RGB and YUV420 color spaces (Jia et al., 2025). We follow the progressive training strategy (Li et al., 2021). For illustration purposes, we only train $\lambda = 256$ models.

**Results:**

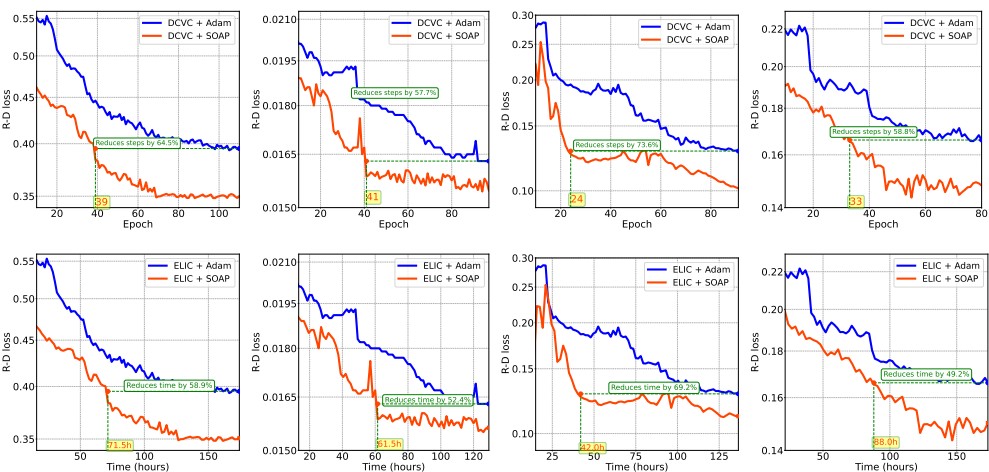

Figure 7: **Training dynamics across stages.** Top row: Loss vs Epochs for Stage 1, Stage 2, Stage 3, and Stage 4. Bottom row: Loss vs Wall time for Stage 1, Stage 2, Stage 3, and Stage 4. SOAP consistently converges faster and more stably than Adam across all stages.

As shown in Fig. 7, SOAP achieves faster convergence and more stable training dynamics than Adam across all stages of DCVC. The improvement holds whether progress is measured in terms of epochs or wall-clock time, indicating that the added per-step overhead of SOAP is easily offset by the substantial reduction in total steps required for convergence.

Importantly, the benefits of SOAP extend beyond acceleration. The final rate–distortion performance (R-D loss) achieved by SOAP is consistently stronger, suggesting that curvature-aware optimization is particularly valuable in the highly complex setting of video compression, where gradient conflicts are even more pronounced. This corroborates our central claim: by resolving intra- and inter-step conflicts, SOAP not only speeds up training but also yields higher-quality solutions.

These preliminary findings suggest that second-order optimization via SOAP generalizes effectively from LIC to learned video compression. While additional large-scale experiments are warranted, the results highlight SOAP as a promising optimizer for future research in video and other high-dimensional compression domains.

### A.5    WILL WEIGHT DECAY MAKE ADAM/ADAMW DIFFERENT?

To investigate whether weight decay is beneficial for LICs, we use the ELIC model (He et al., 2022) as a baseline and follow the same training and evaluation protocol described in Sec. 3. For illustrative purposes, we use $\lambda = 0.013$. We compare two optimizers: Adam (Kingma & Ba, 2014) and AdamW

(Adam with decoupled weight decay) (Loshchilov & Hutter, 2017)[8]. Weight decay values of {0.01, 0.001, 0.0001} are tested. Note that weight decay introduces no noticeable wall-time overhead.

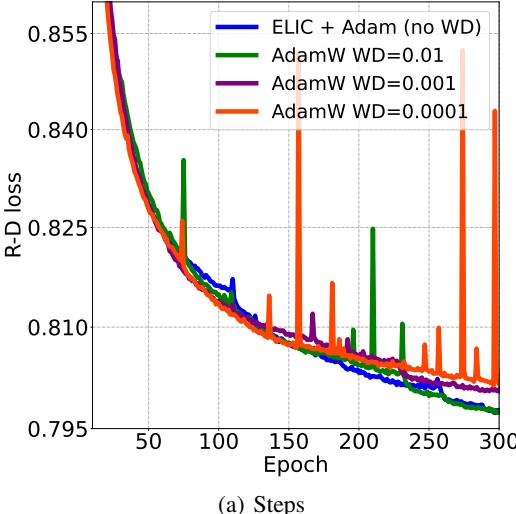

(a) Steps

Figure 8: **Comparison of Testing Loss: Epochs vs. R-D Loss under Different Weight Decay Settings.** *(Best viewed zoomed in.)* The first 10 epochs are omitted for clarity. Evaluation is conducted on the Kodak dataset with $\lambda = 0.013$. The R-D loss is computed as $\lambda \cdot 255^2 \cdot \text{MSE} + \text{Bpp}$.

From Fig. 8, we observe that training with Adam or AdamW *without* weight decay yields the most stable optimization and best final convergence. When weight decay is applied, Adam fails to converge properly:

- Adam + WD = 0.01 stalls at R-D loss $\approx 4.7$

- Adam + WD = 0.001 stalls at $\approx 2.5$

- Adam + WD = 0.0001 stalls at $\approx 1.0$

All of these results are significantly worse than the converged value of $\approx 0.795$. We exclude these curves from Fig. 8 for better visualization.

For AdamW, using weight decay produces results that are either similar to or slightly worse or more unstable than training without weight decay. Given that neither Adam nor AdamW benefits from weight decay in this setting, we choose **not to apply weight decay in any of our experiments**.

### A.6 WHAT IS THE IMPACT OF PRECONDITIONER UPDATE FREQUENCY?

A key hyperparameter of the SOAP optimizer (Vyas et al., 2024) is the *preconditioner update frequency*, which controls the trade-off between computational efficiency and preconditioner accuracy. A smaller frequency value updates the preconditioner more frequently, improving its accuracy but increasing computational overhead. Conversely, a larger frequency reduces the update cost, potentially speeding up training, but risks using a stale preconditioner that may slow convergence.

To empirically study this trade-off, we use the ELIC model (He et al., 2022) as a baseline and follow the same training and evaluation protocol described in Sec. 3. For illustrative purposes, we fix $\lambda = 0.013$ and test update frequencies of {5, 10, 20, 50}. The default implementation is 10.

As shown in Fig. 9, varying the preconditioner update frequency does not significantly affect either step efficiency or wall-clock efficiency. The differences between frequencies are marginal, and thus we adopt the default setting of 10 throughout all experiments.

---

[8]When no weight decay is applied, Adam and AdamW are mathematically equivalent, as their difference lies solely in how weight decay is implemented.

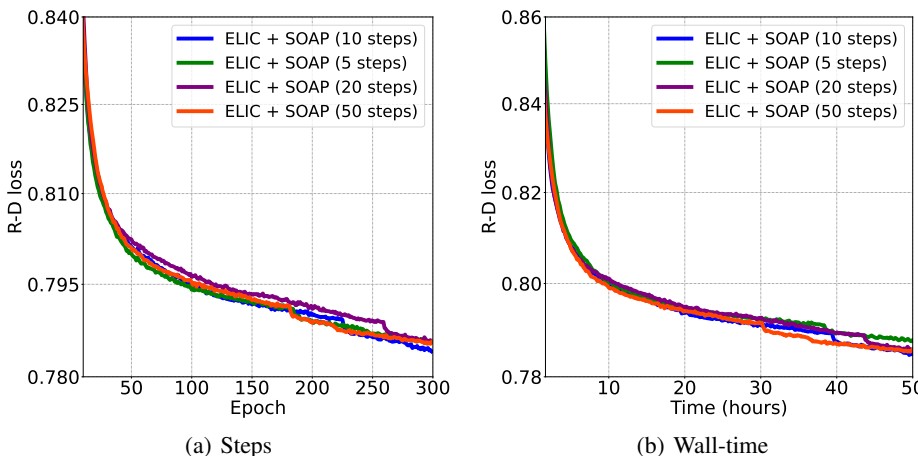

(a) Steps  (b) Wall-time

Figure 9: **Comparison of Testing Loss: Epochs/Wall-time vs. R-D Loss for Various Update Frequencies.** *(Best viewed zoomed in.)* The first 10 epochs are omitted for clarity. Evaluation is performed on the Kodak dataset with $\lambda = 0.013$; the R-D loss is computed as $\lambda \cdot 255^2 \cdot \mathrm{MSE} + \mathrm{Bpp}$.

While Fig. 9 (and the main results across all four LICs considered in this work) indicate that update frequencies in the range $\{5, 10, 20, 50\}$ are both numerically stable and nearly indistinguishable in terms of convergence speed in our setting, practitioners may in principle encounter numerical issues (e.g., exploding activations or `NaN`/`Inf` values) when using very frequent preconditioner updates in marginal situations. From an optimization perspective, more frequent updates are generally desirable: they allow the preconditioner to track changes in the local curvature more closely, making the quasi-Newton step more faithful to the current Hessian and potentially improving convergence in highly non-stationary regimes. However, in architectures where gradient statistics are particularly noisy, this increased reactivity can also make the preconditioner more sensitive to transient spikes. In such cases, a simple mitigation is to increase the update interval (e.g., from 10 to 100 or even 1000), so that each preconditioner refresh aggregates curvature information over more optimization steps. This yields smoother Kronecker-factored curvature estimates and makes their (approximate) inverse less sensitive to transient gradient spikes, thereby acting as a more conservative and stable preconditioner.

### A.7 SOAP as an Approximation to Newton's Method

We argue that the SOAP update can behave like a Newton step *locally and under specific modeling assumptions*, i.e., $p \approx - H^{-1} g$. The derivation proceeds through standard curvature approximations and a rotated-basis view in which SOAP applies an Adam-style preconditioner.

**Under standard assumptions:**

**(A1) Gauss–Newton (GN) surrogate.** The Hessian is well-approximated by its GN component (Bishop & Nasrabadi, 2006; Martens & Grosse, 2015; Martens et al., 2010; Morwani et al., 2024; Zhang et al., 2025a; Schraudolph et al., 2007):

$$H \approx H_{\mathrm{GN}} \qquad \text{(GN approximation)}. \tag{12}$$

**(A2) Layerwise Kronecker structure.** For a single layer with weight matrix $W$ (vectorized as $\mathrm{vec}(W)$), the GN is well-approximated by a Kronecker product of second-moment factors built from forward activations $a_t$ and backpropagated sensitivities $\delta_t$ (Grosse & Martens, 2016; Martens & Grosse, 2015; Li, 2017; Martens, 2020; Morwani et al., 2024; Gupta et al., 2018; Vyas et al., 2024):

$$H_{\mathrm{GN}} \approx R_t \otimes L_t, \quad L_t = \mathbb{E}[\delta_t \delta_t^\top], \quad R_t = \mathbb{E}[a_t a_t^\top]. \tag{13}$$

(An $L \otimes R$ parameterization is equivalent; only the rotation/diagonalization matters.)

**(A3) Rotated-basis diagonalization.** With eigendecompositions $L_t = Q_L \Lambda_L Q_L^\top$ and $R_t = Q_R \Lambda_R Q_R^\top$, the $(Q_L \otimes Q_R)$ rotation makes the GN surrogate (nearly) diagonal:

$$\tilde{H}_{\text{GN}} = (Q_L \otimes Q_R)^\top H_{\text{GN}} (Q_L \otimes Q_R) \approx \Lambda_R \otimes \Lambda_L, \tag{14}$$

which is diagonal because it is the Kronecker product of diagonal matrices.

**(A4) Adam-as-diagonal preconditioner (local).** In the rotated basis and sufficiently close to a (nondegenerate) local minimum, the Adam/Adafactor-style update acts like preconditioning by the *diagonal* curvature (Kingma & Ba, 2014; Reddi et al., 2019; Vyas et al., 2024):

$$\tilde{p} \approx -\operatorname{diag}(\tilde{H}_{\text{GN}})^{-1} \tilde{g}, \tag{15}$$

up to standard damping ($\varepsilon I$), EMAs, and step-size factors.

**Rotated-space argument.** Under (A1)–(A3), $\tilde{H}_{\text{GN}}$ is diagonal, so $\operatorname{diag}(\tilde{H}_{\text{GN}})^{-1} = \tilde{H}_{\text{GN}}^{-1}$. By (A4),

$$\tilde{p} \approx -\tilde{H}_{\text{GN}}^{-1} \tilde{g}. \tag{16}$$

Because $(Q_L \otimes Q_R)$ is orthogonal, applying a preconditioned step in the rotated space is equivalent to applying the corresponding step in the original coordinates:

$$p = (Q_L \otimes Q_R) \tilde{p} \approx -(Q_L \otimes Q_R) \tilde{H}_{\text{GN}}^{-1} (Q_L \otimes Q_R)^\top g. \tag{17}$$

Finally, by (A1),

$$(Q_L \otimes Q_R) \tilde{H}_{\text{GN}}^{-1} (Q_L \otimes Q_R)^\top \approx H^{-1}, \tag{18}$$

yielding the claimed local Newton approximation.

**Theorem 1** (Conditional Newton approximation for SOAP). *Under (A1)–(A4) and with standard damping and stable moment estimates, the SOAP layer update is a local approximation to the Newton update:*

$$p \approx -H^{-1} g. \tag{19}$$

**Remarks and limitations.** (i) The Adam preconditioner tracks (diagonal) second moments of gradients (Fisher-like), not the exact Hessian diagonal; the identification in equation 15 is a *local* approximation strongest when $\operatorname{diag}(H_{\text{GN}}) \approx \operatorname{diag}(H)$ near the optimum. (ii) Finite-sample EMAs, infrequent preconditioner updates, and regularization ($+\varepsilon I$) introduce additional approximation error. (iii) The argument is layerwise and ignores inter-layer curvature; nonetheless, in practice, the rotated-space diagonalization substantially improves conditioning compared to first-order methods. (iv) For common losses (e.g., MSE, cross-entropy, typical distortion, and rate losses), the Fisher information matrix and GN coincide and provide a PSD approximation to the true Newton matrix under standard assumptions, which are widely used in practice as stable surrogates for second-order optimization.

## A.8 PROOF OF LEMMA 1

Assume $f$ has an $L_H$-Lipschitz Hessian in a neighborhood of $\theta_t$, and $H_t = \nabla^2 f(\theta_t)$ is SPD with $\|H_t^{-1}\| \le \kappa$. For the Newton update

$$p_t = -H_t^{-1} g_t, \qquad \theta_{t+1} = \theta_t + \eta p_t, \quad 0 < \eta < 1, \tag{20}$$

there exist constants $C_1, C_2$ (depending on $L_H$ and uniform bounds on $\|H_t\|, \|H_t^{-1}\|$) such that, whenever $\|g_t\|$ is sufficiently small,

$$\left| 1 - \mathcal{S}(p_t, p_{t+1}) \right| \le C_1 \, \eta \, \|p_t\| + C_2 \, \eta^2 \, \|p_t\|^2. \tag{21}$$

In particular, as $\|p_t\| \to 0$ (or as $\eta \to 0$), $\mathcal{S}(p_t, p_{t+1}) \to 1$.

*Proof.* By the Lipschitz continuity of the Hessian (Taylor expansion),

$$g_{t+1} = g(\theta_{t+1}) = g_t + H_t(\theta_{t+1} - \theta_t) + r_t, \quad \|r_t\| \le \frac{L_H}{2} \|\theta_{t+1} - \theta_t\|^2. \tag{22}$$

Since $\theta_{t+1} - \theta_t = \eta p_t = -\eta H_t^{-1} g_t$ and $\|H_t^{-1}\| \le \kappa$,

$$g_{t+1} = (1 - \eta) g_t + r_t, \qquad \|r_t\| \le \frac{L_H}{2} \kappa^2 \, \eta^2 \|g_t\|^2. \tag{23}$$

Now the next Newton update is

$$p_{t+1} = -H_{t+1}^{-1} g_{t+1}. \tag{24}$$

Add and subtract $H_t^{-1}$:

$$p_{t+1} = -H_t^{-1} g_{t+1} \; - \; (H_{t+1}^{-1} - H_t^{-1}) g_{t+1}. \tag{25}$$

Using $g_{t+1} = (1-\eta) g_t + r_t$ and $p_t = -H_t^{-1} g_t$, we get

$$p_{t+1} = (1-\eta) p_t - H_t^{-1} r_t \; - \; (H_{t+1}^{-1} - H_t^{-1}) g_{t+1}. \tag{26}$$

Lipschitzness implies $\|H_{t+1} - H_t\| \le L_H \|\theta_{t+1} - \theta_t\| = L_H \eta \|p_t\|$, hence

$$\|H_{t+1}^{-1} - H_t^{-1}\| \; \le \; \|H_t^{-1}\| \, \|H_{t+1} - H_t\| \, \|H_{t+1}^{-1}\| \; \le \; C \, \eta \|p_t\| \tag{27}$$

for $C = \kappa^2 L_H$ (assuming $\|H_{t+1}^{-1}\|$ remains bounded) in a small neighborhood. Combining these bounds yields

$$p_{t+1} = (1-\eta) p_t + e_t, \qquad \|e_t\| \; \le \; C_1' \, \eta \, \|p_t\|^2 + C_2' \, \eta^2 \, \|p_t\|^3. \tag{28}$$

Writing $u = p_t / \|p_t\|$ and $p_{t+1} = (1-\eta) \|p_t\| \, u + e_t$, a standard cosine perturbation bound gives

$$\big| 1 - \mathcal{S}(p_t, p_{t+1}) \big| \; \le \; C_1 \, \eta \, \|p_t\| \; + \; C_2 \, \eta^2 \, \|p_t\|^2, \tag{29}$$

as claimed. $\qquad\square$

### A.9 PROOF OF PROPOSITION 1

Let $\theta^*$ be a nondegenerate local minimizer with Hessian $H \succ 0$. Assume that, in a neighborhood of $\theta^*$, the component gradients admit quadratic models (Nocedal & Wright, 1999; Boyd & Vandenberghe, 2004):

$$g_R(\theta) \approx H_R(\theta - \theta^*), \qquad g_D(\theta) \approx H_D(\theta - \theta^*), \tag{30}$$

and that SOAP uses a single (shared) preconditioner that locally approximates $H^{-1}$, i.e.,

$$p \approx -H^{-1} g \quad \text{(cf. Sec. A.7).} \tag{31}$$

Suppose, moreover, that the component Hessians are *locally proportional* to $H$:

$$H_R(\theta) \; = \; \alpha_R(\theta) \, H(\theta) + E_R(\theta), \qquad H_D(\theta) \; = \; \alpha_D(\theta) \, H(\theta) + E_D(\theta), \tag{32}$$

where $\alpha_R, \alpha_D > 0$ are continuous near $\theta^*$ and $\|E_R(\theta)\|, \|E_D(\theta)\| = o(1)$ as $\theta \to \theta^*$. Then

$$\lim_{\theta \to \theta^*} \mathcal{S}(p_R(\theta), \, p_D(\theta)) \; = \; 1, \tag{33}$$

where $p_R \approx -H^{-1} g_R$ and $p_D \approx -H^{-1} g_D$ are the SOAP update vectors corresponding to the rate and distortion gradients.

*Proof.* Using the shared preconditioner and the quadratic models,

$$p_R \; \approx \; -H^{-1} H_R (\theta - \theta^*) = -\alpha_R(\theta) \, (\theta - \theta^*) \; + \; -H^{-1} E_R(\theta) \, (\theta - \theta^*). \tag{34}$$

Because $\|E_R(\theta)\| = o(1)$ and $\|H^{-1}\|$ is bounded near $\theta^*$, we have

$$\|H^{-1} E_R(\theta)(\theta - \theta^*)\| = o(\|\theta - \theta^*\|). \tag{35}$$

An identical argument yields

$$p_D \; \approx \; -\alpha_D(\theta) \, (\theta - \theta^*) \; + \; o(\|\theta - \theta^*\|). \tag{36}$$

Thus both update vectors $p_R$ and $p_D$ are colinear with $-(\theta - \theta^*)$ up to a vanishing error. Hence their cosine similarity converges to 1 as $\theta \to \theta^*$. $\qquad\square$

**Remark.** Without proportionality, the update vectors $p_R = -H^{-1} H_R(\theta - \theta^*)$ and $p_D = -H^{-1} H_D(\theta - \theta^*)$ need not be parallel. A weaker (sufficient) condition is that $H, H_R, H_D$ are jointly diagonalizable near $\theta^*$ and that the ratios $\lambda_R^i / \lambda^i$ and $\lambda_D^i / \lambda^i$ are constant on the (active) eigenspaces visited by $(\theta - \theta^*)$, which again renders the two vectors colinear. However, these assumptions serve as sufficient conditions that provide essential theoretical intuition for why second-order preconditioning aids alignment. In the context of R-D optimization, it is plausible that rate and distortion objectives share significant curvature structure, as both depend on the capacity and fidelity of the underlying transform. The strong empirical alignment observed in practice (Fig. 3a) suggests that the R-D optimization landscape possesses enough shared structure for SOAP to effectively exploit, even if these idealized conditions are not perfectly met. The Newton preconditioner inherently seeks a shared descent direction by accounting for how the objectives interact locally.

### A.10 LIMITATIONS OF ADAM FOR GRADIENT ALIGNMENT

Adam is powerful and widely used, but its effectiveness is inherently limited by its *diagonal* preconditioner. Because it scales coordinates independently, it cannot exploit off–diagonal curvature that encodes interactions among parameters—precisely what is needed to resolve non–axis-aligned gradient conflicts in multi-objective settings such as rate–distortion (R–D) optimization. The following proposition formalizes a standard local approximation behind this limitation, following Molybog et al. (2023); Martens & Grosse (2015).

**Proposition 2** (Local diagonal-preconditioner approximation). *In a neighborhood of a nondegenerate local minimum $\theta^*$ where the loss is well-approximated by a quadratic and the Hessian $H \succ 0$ is close to diagonal (diagonally dominant), the Adam update vector is approximately a diagonally preconditioned gradient step:*

$$p_{\text{Adam}}(g) = c \, \text{diag}(H)^{-1}g + o(\|g\|), \tag{37}$$

*for some scalar $c > 0$ that absorbs stepsize, bias-correction, and damping factors.*

*Proof.* Adam (Kingma & Ba, 2014) maintains

$$m_t = \beta_1 m_{t-1} + (1 - \beta_1)g_t,$$
$$v_t = \beta_2 v_{t-1} + (1 - \beta_2)(g_t \odot g_t),$$

$$\theta_{t+1} = \theta_t - \eta \frac{\hat{m}_t}{\sqrt{\hat{v}_t} + \epsilon},$$

with bias-corrected $\hat{m}_t, \hat{v}_t$ and elementwise operations. For local conditioning it suffices to (i) linearize $g_t \approx H(\theta_t - \theta^*)$ and (ii) use $m_t \approx g_t$ to expose the preconditioner. Under small, approximately isotropic perturbations near $\theta^*$, $\mathbb{E}[(\theta_t - \theta^*)(\theta_t - \theta^*)^\top] \approx \sigma^2 I$, giving

$$\mathbb{E}[g_t g_t^\top] \approx \sigma^2 H H^\top. \tag{38}$$

Hence

$$v_t \approx \text{diag}(\sigma^2 H H^\top), \qquad \sqrt{v_t} \approx \sqrt{\sigma^2 \, \text{diag}(H H^\top)}. \tag{39}$$

Diagonal dominance implies $\text{diag}(H H^\top)_{ii} = \sum_k H_{ik}^2 \approx H_{ii}^2$, so

$$\sqrt{v_t} \approx \sigma \, \text{diag}(H), \tag{40}$$

(using $H \succ 0$). Therefore the Adam update vector is

$$p_{\text{Adam}}(g_t) \approx \frac{g_t}{\sigma \, \text{diag}(H)} = c \, \text{diag}(H)^{-1}g_t, \tag{41}$$

with $c = 1/\sigma$, as claimed. $\qquad\square$

**Why a diagonal preconditioner fails.** The core limitation of Adam in this context is structural. For multi-objective problems (e.g., R-D), parameter couplings are encoded in the *off-diagonal* entries of $H$ (Das et al., 2024). A diagonal preconditioner cannot mix coordinates and therefore cannot *rotate* the update direction $p_{\text{Adam}}$ toward a descent direction that resolves conflicting objectives. This remains true regardless of how accurately Adam's second-moment estimate $v_t$ approximates the true Hessian diagonal (which itself relies on strong assumptions like diagonal dominance used in the proof above). This inability to rotate the update leads to inherent intra-step conflicts and poor inter-step alignment, often manifesting as oscillatory trajectories in practice. In contrast, SOAP's block-diagonal curvature approximation preserves within-block off-diagonal structure, enabling within-layer rotations that align conflicting updates and accelerate convergence.

### A.11 ADAM'S GRADIENT CONFLICT IN A SIMPLIFIED R-D SETTING AT INITIALIZATION

We now make the above limitation concrete in a toy R-D problem. Consider a linear autoencoder (Saxe et al., 2013) with encoder $e$ and decoder $d$. For a scalar input $x$, the latent is $z = W_e x$ and the reconstruction is $\hat{x} = W_d z$, where $W_e \in \mathbb{R}^{M \times 1}$ and $W_d \in \mathbb{R}^{1 \times M}$. Let $\theta = (\mathbf{w}_e, \mathbf{w}_d)$ denote the vectorized parameters. The R-D loss balances distortion and rate,

$$\mathcal{L}(\theta) = \underbrace{\mathbb{E}_{x \sim \mathcal{U}[-1,1]}[(\hat{x} - x)^2]}_{\mathcal{L}_D(\theta)} + \lambda \underbrace{\mathbb{E}_{x \sim \mathcal{U}[-1,1]}[\|z\|^2]}_{\mathcal{L}_R(\theta)}. \tag{42}$$

Write $C = \mathbb{E}[x^2] = 1/3$.

**Assumption 1.** Small random initialization. *Entries of $\mathbf{w}_e, \mathbf{w}_d$ are i.i.d. $\mathcal{N}(0, \epsilon^2)$ with $\epsilon = o(1)$, and $\bar{\mathbf{w}}_e = \epsilon^{-1}\mathbf{w}_e$, $\bar{\mathbf{w}}_d = \epsilon^{-1}\mathbf{w}_d$.*

**Proposition 3.** *Under Assumption 1, let $p_R = \text{Adam}(\nabla\mathcal{L}_R)$ and $p_D = \text{Adam}(\nabla\mathcal{L}_D)$ denote Adam's update vectors at initialization. In the wide-latent limit $M \to \infty$ (Jacot et al., 2018; Yang & Hu, 2020),*

$$\mathcal{S}(p_R, p_D) \xrightarrow[M \to \infty]{a.s.} 0. \tag{43}$$

*Thus, Adam updates rate and distortion in asymptotically orthogonal directions, inducing an inefficient trajectory.*

*Proof.* **Leading-order gradients.** The rate term is

$$\mathcal{L}_R = \mathbb{E}\|W_e x\|^2 = \mathbb{E}[x^2]\,\|\mathbf{w}_e\|^2 = C\|\mathbf{w}_e\|^2, \tag{44}$$

so $\nabla_{\mathbf{w}_e}\mathcal{L}_R = 2C\,\mathbf{w}_e = 2C\epsilon\,\bar{\mathbf{w}}_e$ and $\nabla_{\mathbf{w}_d}\mathcal{L}_R = \mathbf{0}$. The distortion term is

$$\mathcal{L}_D = \mathbb{E}[(W_d W_e x - x)^2] = C\,(\mathbf{w}_d\cdot\mathbf{w}_e - 1)^2. \tag{45}$$

Because $\mathbf{w}_d\cdot\mathbf{w}_e = \epsilon^2(\bar{\mathbf{w}}_d\cdot\bar{\mathbf{w}}_e) = O(\epsilon^2)$, we obtain

$$\nabla_{\mathbf{w}_e}\mathcal{L}_D = -2C\epsilon\,\bar{\mathbf{w}}_d + O(\epsilon^3),$$
$$\nabla_{\mathbf{w}_d}\mathcal{L}_D = -2C\epsilon\,\bar{\mathbf{w}}_e + O(\epsilon^3).$$

Collecting terms for $\theta = (\mathbf{w}_e, \mathbf{w}_d)$,

$$\nabla_\theta\mathcal{L}_R \approx (2C\epsilon\,\bar{\mathbf{w}}_e, \mathbf{0}), \qquad \nabla_\theta\mathcal{L}_D \approx (-2C\epsilon\,\bar{\mathbf{w}}_d, -2C\epsilon\,\bar{\mathbf{w}}_e). \tag{46}$$

**Adam's initial updates.** Early in training, Adam's elementwise scaling makes the update direction close to $\text{sign}(g)$ (Balles & Hennig, 2018). Thus,

$$p_R \propto (\text{sign}(\bar{\mathbf{w}}_e), \mathbf{0}), \quad p_D \propto (-\text{sign}(\bar{\mathbf{w}}_d), -\text{sign}(\bar{\mathbf{w}}_e)). \tag{47}$$

**Alignment.** Let $u_R = (\text{sign}(\bar{\mathbf{w}}_e), \mathbf{0})$ and $u_D = (-\text{sign}(\bar{\mathbf{w}}_d), -\text{sign}(\bar{\mathbf{w}}_e))$. Then

$$\langle u_R, u_D \rangle = -\sum_{i=1}^{M} \text{sign}(\bar{w}_{e,i})\,\text{sign}(\bar{w}_{d,i}), \tag{48}$$

$$\|u_R\|^2 = M, \quad \|u_D\|^2 = 2M. \tag{49}$$

Hence

$$\mathcal{S}(u_R, u_D) = -\frac{1}{M\sqrt{2}}\sum_{i=1}^{M} \text{sign}(\bar{w}_{e,i}\bar{w}_{d,i}). \tag{50}$$

Under Assumption 1, the signs are i.i.d. Rademacher variables with mean zero, so the average converges a.s. to 0 as $M \to \infty$ by the strong law of large numbers, proving the claim. $\qquad\square$

**Takeaway.** In this stylized R-D setting, Adam's diagonal preconditioning makes the rate and distortion *updates* nearly orthogonal at initialization, degrading joint progress. Methods that capture within-layer off-diagonal curvature (e.g., SOAP's block-diagonal preconditioner) can rotate updates to better align competing objectives, yielding more direct descent paths (Martens et al., 2010; Balles & Hennig, 2018; Wang et al., 2025c).

**Note (positive per–coordinate scalings).** The orthogonality conclusion in Proposition 3 is unchanged if Adam's elementwise normalization introduces arbitrary *positive* scalings that are independent of the signs of the initialized weights. Concretely, let

$$u_R = \big(a_i\,\text{sign}(\bar{w}_{e,i})\big)_{i=1}^{M} \oplus \mathbf{0}, \qquad u_D = \big(-b_i\,\text{sign}(\bar{w}_{d,i})\big)_{i=1}^{M} \oplus \big(-c_i\,\text{sign}(\bar{w}_{e,i})\big)_{i=1}^{M}, \tag{51}$$

where $a_i, b_i, c_i > 0$ are any (possibly random) scalings produced by Adam's second-moment terms and damping, assumed independent of $\text{sign}(\bar{w}_{e,i}), \text{sign}(\bar{w}_{d,i})$. Then

$$\mathcal{S}(u_R, u_D) = -\frac{\frac{1}{M}\sum_{i=1}^{M} a_i b_i\,\text{sign}(\bar{w}_{e,i}\bar{w}_{d,i})}{\sqrt{\left(\frac{1}{M}\sum_{i=1}^{M} a_i^2\right)\left(\frac{1}{M}\sum_{i=1}^{M}(b_i^2 + c_i^2)\right)}}. \tag{52}$$

If the empirical second moments converge, i.e.,

$$\frac{1}{M}\sum_i a_i^2 \to \bar{a}^2 > 0 \quad \text{and} \quad \frac{1}{M}\sum_i (b_i^2 + c_i^2) \to \bar{b}^2 + \bar{c}^2 > 0, \tag{53}$$

and the Rademacher variables $\mathrm{sign}(\bar{w}_{e,i}\bar{w}_{d,i})$ are i.i.d. with mean 0 (and independent of $a_i, b_i$), then by the strong law of large numbers

$$\frac{1}{M}\sum_{i=1}^{M} a_i b_i \ \mathrm{sign}(\bar{w}_{e,i}\bar{w}_{d,i}) \ \xrightarrow{\text{a.s.}} \ 0, \tag{54}$$

while the denominator converges almost surely to $\sqrt{\bar{a}^2(\bar{b}^2 + \bar{c}^2)}$. Hence

$$\mathcal{S}(u_R, u_D) \xrightarrow{\text{a.s.}} 0. \tag{55}$$

Thus, the asymptotic orthogonality persists under any positive, sign-independent coordinate scalings induced by Adam.

### A.12 Adam's Alignment Behaviour Near a Nondegenerate Optimum

We complement Secs. A.10 and A.11 by *formally* characterizing Adam's (i) inter-step alignment $\mathcal{S}(p_t, p_{t+1})$ and (ii) intra-step alignment $\lim_{\theta \to \theta^*} \mathcal{S}(p_R, p_D)$ in a neighborhood of a nondegenerate optimum.

**Standing assumptions.** Throughout we adopt the standard local model used for diagonal adaptive methods:

**(B1) Quadratic model near $\theta^*$.** Writing $e_t = \theta_t - \theta^*$, the total loss satisfies $g_t = \nabla\mathcal{L}(\theta_t) = He_t$ with $H \succ 0$ constant locally (Nocedal & Wright, 1999).

**(B2) Frozen second moments.** Adam's second–moment accumulator and damping are (locally) stationary, yielding a fixed positive diagonal matrix $D \succ 0$ and a scalar $c > 0$ (absorbing step-size, bias correction, $\varepsilon$). Hence the Adam update is the diagonally preconditioned step (Kingma & Ba, 2014; Reddi et al., 2019; Zaheer et al., 2018)

$$p_t = -cD^{-1}g_t = -Ae_t, \qquad A := cD^{-1}H. \tag{56}$$

**(B3) Small step regime.** Parameters evolve by $\theta_{t+1} = \theta_t + \eta p_t$ with $0 < \eta < 1$ so that the linearization remains valid.

**A symmetric reparameterization.** To analyze the dynamics, we introduce a reparameterization using the fixed diagonal preconditioner $D$. Let

$$B := cD^{-1/2}HD^{-1/2} \succ 0, \qquad q_t := D^{1/2}p_t, \qquad y_t := D^{1/2}e_t.$$

Here, $B$ represents the Hessian in the $D$-whitened coordinate space. Then the local dynamics (equation 56) implies

$$q_t = -By_t, \qquad y_{t+1} = (I - \eta B)y_t, \qquad q_{t+1} = (I - \eta B)q_t. \tag{57}$$

Since $B$ is symmetric positive definite, inter-step cosines in the $q$–space admit closed forms; cosines in the original coordinates are equivalent up to constants depending only on $\kappa(D)$ (the condition number of D) (Petersen et al., 2008).

#### A.12.1 Inter-step cosine for Adam

**Lemma 2** (Local inter-step cosine for diagonal preconditioning). *Under (B1)–(B3), with $u_t = q_t/\|q_t\|$ and Rayleigh statistics*

$$\mu_1(u_t) := u_t^\top B u_t, \qquad \mu_2(u_t) := u_t^\top B^2 u_t,$$

*the* exact *inter-step cosine in the q–space is*

$$\mathcal{S}_q(q_t, q_{t+1}) = \frac{1 - \eta\,\mu_1(u_t)}{\sqrt{1 - 2\eta\,\mu_1(u_t) + \eta^2\,\mu_2(u_t)}}. \tag{58}$$

*In particular, for small $\eta$,*

$$\mathcal{S}_q(q_t, q_{t+1}) = 1 - \tfrac{1}{2}\eta^2\big(\mu_2(u_t) - \mu_1(u_t)^2\big) + O(\eta^3) = 1 - \tfrac{1}{2}\eta^2\,\mathrm{Var}_{u_t}(B) + O(\eta^3). \tag{59}$$

*Proof.* From equation 57, $q_{t+1} = (I - \eta B)q_t$. Therefore

$$\mathcal{S}_q(q_t, q_{t+1}) = \frac{\langle q_t, (I - \eta B)q_t \rangle}{\|q_t\| \, \|(I - \eta B)q_t\|} = \frac{1 - \eta \, u_t^\top B u_t}{\sqrt{1 - 2\eta \, u_t^\top B u_t + \eta^2 \, u_t^\top B^2 u_t}},$$

yielding equation 58. A Taylor expansion of the denominator gives equation 59. The term $\mathrm{Var}_{u_t}(B)$ represents the variance of the eigenvalues of $B$ (the whitened Hessian) with respect to the direction $u_t$. □

**Consequences.**

- *No automatic alignment as $\|p_t\| \to 0$.* Unlike Newton (Lemma 1), the deviation $1 - \mathcal{S}_q$ is *second order in $\eta$* and controlled by curvature anisotropy $\mathrm{Var}_{u_t}(B)$, not by $\|p_t\|$. Thus $\mathcal{S}(p_t, p_{t+1})$ need not approach 1 near the optimum unless $B$ is a scalar multiple of $I$ or $u_t$ is an eigenvector of $B$.

- *Oscillation threshold.* If $\eta \, \mu_1(u_t) > 1$, the numerator in equation 58 is negative and $\mathcal{S}_q < 0$. Hence whenever $\eta \, \lambda_{\max}(B) > 1$, there exist directions with *negative* inter-step cosine (flip–flop behaviour).

- *Back to original coordinates.* Since $q_t = D^{1/2} p_t$ and $D \succ 0$ is fixed locally, Euclidean cosines of $(p_t, p_{t+1})$ and $(q_t, q_{t+1})$ are equivalent up to constants depending on $\kappa(D)$; all qualitative conclusions transfer to $\mathcal{S}(p_t, p_{t+1})$.

*Intuition.* In the R-D setting, Adam's diagonal preconditioning cannot remove curvature anisotropy: the inter-step cosine is governed by the variance of eigenvalues rather than by step size alone. As a result, update directions often fail to align even near convergence. In practice, this manifests as oscillatory trajectories—updates pulling in different directions—rather than the smooth progress observed under Newton-like SOAP. In other words, *Adam has no guarantee of alignment, while SOAP actively suppresses this oscillation behaviour.*

A.12.2    INTRA-STEP COSINE FOR ADAM NEAR THE OPTIMUM

Near $\theta^*$, the component gradients linearize (Nocedal & Wright, 1999) as

$$g_R \approx H_R(\theta - \theta^*), \qquad g_D \approx H_D(\theta - \theta^*),$$

with $H_R, H_D \succeq 0$. Under **(B2)**, the shared Adam preconditioner $D$ is (locally) fixed, so

$$p_R(\theta) \approx -c \, D^{-1} H_R(\theta - \theta^*), \qquad p_D(\theta) \approx -c \, D^{-1} H_D(\theta - \theta^*).$$

Let $e = \theta - \theta^*$ and $u = e/\|e\|$.

**Proposition 4** (Exact intra-step limit for Adam). *Fix any sequence $\theta_k \to \theta^*$ such that $u_k = (\theta_k - \theta^*)/\|\theta_k - \theta^*\| \to u$ with $\|u\| = 1$. Under (B2),*

$$\lim_{k \to \infty} \mathcal{S}\big(p_R(\theta_k), p_D(\theta_k)\big) = \frac{\langle D^{-1} H_R u, \, D^{-1} H_D u \rangle}{\|D^{-1} H_R u\| \, \|D^{-1} H_D u\|} =: \rho_{\mathrm{Adam}}(u). \tag{60}$$

*Moreover:*

(i) *$\rho_{\mathrm{Adam}}(u) = 1$ iff $D^{-1} H_R u$ and $D^{-1} H_D u$ are colinear, i.e., $D^{-1} H_R u = \alpha \, D^{-1} H_D u$ for some $\alpha > 0$. A sufficient condition is that $H_R$ and $H_D$ are locally proportional on the $D$-whitened direction $D^{1/2} u$.*

(ii) *If $H_R, H_D, D$ are jointly diagonalizable, then $\rho_{\mathrm{Adam}}(u) \in [0, 1]$ for all $u$, and $\rho_{\mathrm{Adam}}(u) = 1$ iff the per-coordinate ratios are constant on the support of $u$ (the same condition that yields SOAP's alignment in Prop. 1).*

(iii) *In general (non-commuting case), $\rho_{\mathrm{Adam}}(u)$ can take any value in $(-1, 1)$. In particular, there exist SPD triples $(H_R, H_D, D)$ and $u$ such that $\rho_{\mathrm{Adam}}(u) \leq 0$.*

*Proof.* Substitute the linearizations

$$p_R(\theta) \approx -c \, D^{-1} H_R e, \qquad p_D(\theta) \approx -c \, D^{-1} H_D e, \quad e = \theta - \theta^*,$$

and cancel the common positive factor $c/\|e\|$. This yields the expression in equation 60. Continuity then guarantees the limit along any sequence $\theta_k \to \theta^*$ with normalized directions $u_k \to u$.

*Case (i).* If $D^{-1}H_R u$ and $D^{-1}H_D u$ are colinear with positive scalar $\alpha$, then the cosine is exactly 1. Conversely, if the cosine is 1, the two vectors must be positively colinear by definition.

*Case (ii).* If $H_R, H_D, D$ are jointly diagonalizable, choose the common eigenbasis. In this basis, $D^{-1}H_R$ and $D^{-1}H_D$ are diagonal with positive entries. For any $u$, the inner product is non-negative, so $\rho_{\mathrm{Adam}}(u) \in [0,1]$. Equality $\rho_{\mathrm{Adam}}(u) = 1$ requires that the coordinatewise ratios $(H_R)_{jj}/(H_D)_{jj}$ be constant on the support of $u$, ensuring proportionality of the two preconditioned vectors.

*Case (iii).* In the general non-commuting case, $D^{-1}H_R$ and $D^{-1}H_D$ need not share eigenvectors, and their images of $u$ can point in very different directions. To see that negative cosines are possible, set $D = I$ in $\mathbb{R}^2$ and take

$$H_R = \begin{bmatrix} 10 & 0 \\ 0 & 1 \end{bmatrix}, \qquad H_D = R(\vartheta) \begin{bmatrix} 10 & 0 \\ 0 & 1 \end{bmatrix} R(\vartheta)^\top,$$

with $R(\vartheta)$ a rotation by $\vartheta \simeq 57°$. For $u = 2^{-1/2}(1,-1)$, a direct calculation gives $\langle H_R u, H_D u \rangle < 0$, so $\rho_{\mathrm{Adam}}(u) < 0$ even though $H_R, H_D$ are SPD. $\qquad\square$

**Empirical note.** In practice, we find that Adam's intra-step cosine rarely falls in $[0,1]$ as in the commutative case, but instead is often *strongly negative*. Around local minima of the ELIC model, the measured $\mathcal{S}_{\mathrm{intra}}^t$ values concentrate near $-1$ (see Fig. 3(a)), confirming that Adam is unable to align the updates of rate and distortion objectives. This behaviour is consistent with Proposition 4 (case (iii)) and explains the inefficient dynamics observed empirically.

**Remarks** Nonzero momentum ($\beta_1 > 0$) produces a linear two-term recurrence in the $q$–space; all qualitative conclusions above persist with $B$ replaced by an $O(1)$ affine function of $B$. If $D$ evolves slowly rather than remaining fixed, equation 58–equation 60 apply between preconditioner refreshes with the current $D_t$.

**Takeaways vs. SOAP.** Near $\theta^*$, Adam's inter-step misalignment is governed by curvature *anisotropy* via $\mathrm{Var}_{u_t}(B)$ and does *not* vanish with $\|p_t\|$ (Lemma 2), whereas the Newton-like SOAP bound (Lemma 1) decays as $O(\eta\|p_t\|)$. For the intra-step metric, SOAP yields $\mathcal{S}(p_R, p_D) \to 1$ under mild proportionality/diagonalization conditions (Prop. 1), while Adam's limit $\rho_{\mathrm{Adam}}(u)$ in equation 60 generally depends on the *approach direction* $u$ and can be $\leq 0$ unless the component Hessians align in the $D$-whitened geometry.

### A.13 WHY HIGH COSINE ACCELERATES OPTIMIZATION

A natural question arises regarding the optimization of the rate-distortion objective: given that rate ($\mathcal{L}_R$) and distortion ($\mathcal{L}_D$) are intrinsically conflicting objectives, one might expect the cosine similarity between their update directions to be small or negative (Yu et al., 2020). While raw gradient conflict is indeed a characteristic of the problem, we formally show here that a larger cosine similarity between the *preconditioned* update vectors ($p_R$ and $p_D$) is strictly beneficial for convergence speed.

We demonstrate that the lower bound of the loss reduction at each step depends monotonically on the intra-step cosine. Consequently, an optimizer (like SOAP) that induces high cosine effectively resolves the "destructive interference" between competing gradients, maximizing the effective step size for a given gradient magnitude.

**Proposition 5** (Alignment Maximizes Descent Efficiency)**.** *Let the total loss function $\mathcal{L}(\theta) = \mathcal{L}_R(\theta) + \lambda\mathcal{L}_D(\theta)$ be L-smooth. Consider the update $\theta_{t+1} = \theta_t + \eta p_t$, where the total update $p_t = p_{R,t} + p_{D,t}$ is composed of preconditioned rate and distortion components ($p_{k,t} = -P_t\nabla\mathcal{L}_k(\theta_t)$). Assume the preconditioner $P_t$ is positive definite with eigenvalues bounded by $0 < \mu \leq \lambda_i(P_t) \leq M$.*

*For a learning rate $\eta < \frac{2}{LM}$, the reduction in loss $\Delta_t = \mathcal{L}(\theta_t) - \mathcal{L}(\theta_{t+1})$ is lower-bounded by:*

$$\Delta_t \geq C(\eta) \cdot \left( \|p_{R,t}\|^2 + \|p_{D,t}\|^2 + 2\|p_{R,t}\|\|p_{D,t}\| \cdot \mathcal{S}_{\mathrm{intra}}^t \right), \tag{61}$$

where $C(\eta) = \frac{\eta}{M} - \frac{L\eta^2}{2} > 0$. *Thus, strictly increasing $\mathcal{S}_{\text{intra}}^t$ strictly increases the guaranteed loss reduction.*

*Proof.* **Step 1: Quadratic Upper Bound.** By the $L$-smoothness of $\mathcal{L}$, the Descent Lemma guarantees (Boyd & Vandenberghe, 2004):

$$\mathcal{L}(\theta_{t+1}) - \mathcal{L}(\theta_t) \leq \langle \nabla\mathcal{L}(\theta_t), \eta p_t \rangle + \frac{L}{2}\|\eta p_t\|^2. \tag{62}$$

**Step 2: Linking Gradient to Update Norm.** Let $g_t = \nabla\mathcal{L}(\theta_t)$. Since $p_t = -P_t g_t$, we have $g_t = -P_t^{-1} p_t$. The linear term becomes:

$$\langle g_t, p_t \rangle = -p_t^\top P_t^{-1} p_t. \tag{63}$$

Using the eigenvalue bounds of $P_t$, the eigenvalues of $P_t^{-1}$ are at least $1/M$. Therefore, $p_t^\top P_t^{-1} p_t \geq \frac{1}{M}\|p_t\|^2$. Substituting this into the inequality:

$$\mathcal{L}(\theta_{t+1}) - \mathcal{L}(\theta_t) \leq -\eta\frac{1}{M}\|p_t\|^2 + \frac{L\eta^2}{2}\|p_t\|^2 = -\left(\frac{\eta}{M} - \frac{L\eta^2}{2}\right)\|p_t\|^2. \tag{64}$$

**Step 3: Geometric Decomposition.** Let $\Delta_t = \mathcal{L}(\theta_t) - \mathcal{L}(\theta_{t+1})$. Provided $\eta < \frac{2}{LM}$, the coefficient $C(\eta) = \frac{\eta}{M} - \frac{L\eta^2}{2}$ is positive. Thus:

$$\Delta_t \geq C(\eta)\|p_t\|^2 = C(\eta)\|p_{R,t} + p_{D,t}\|^2. \tag{65}$$

Expanding the squared norm using the cosine definition $\langle a, b \rangle = \|a\|\|b\|\mathcal{S}(a,b)$ yields the final bound:

$$\Delta_t \geq C(\eta)\left(\|p_{R,t}\|^2 + \|p_{D,t}\|^2 + 2\|p_{R,t}\|\|p_{D,t}\|\mathcal{S}_{\text{intra}}^t\right). \tag{66}$$

$\square$

**Interpretation.** The proposition highlights that optimization efficiency depends not just on gradient magnitudes, but critically on their vector alignment.

- **Destructive Interference** ($\mathcal{S}_{\text{intra}}^t < 0$)**:** When update vectors conflict, the cross-term becomes negative. The optimizer expends the magnitude of the individual updates ("energy") merely to cancel each other out, resulting in a small effective step $\|p_t\|$ and minimal loss reduction. This corresponds to the "zigzagging" often seen with Adam.
- **Constructive Synergy** ($\mathcal{S}_{\text{intra}}^t \to 1$)**:** When SOAP aligns the updates via curvature correction, the cross-term is maximized. The rate and distortion updates effectively sum up, producing the largest possible descent step for the given gradient magnitudes.

Thus, while the *objectives* ($\mathcal{L}_R, \mathcal{L}_D$) are conflicting, an optimal preconditioner must rotate the space such that the *updates* are cooperative. The high intra-step cosine observed with SOAP (Fig. 3) confirms it successfully achieves this constructive synergy.

We now show that alignment between consecutive updates (inter-step) is equally critical for maximizing the effective displacement along the descent path.

**Proposition 6** (Trajectory Coherence Maximizes Descent)**.** *Consider the cumulative loss reduction over two consecutive steps, $\Delta_{2,t} = \mathcal{L}(\theta_{t-1}) - \mathcal{L}(\theta_{t+1})$. Adopting the same assumptions as Proposition 5 (L-smoothness and spectral bound M), further assume the preconditioner $P$ is locally isotropic ($P^{-1} \approx \frac{1}{\sigma}I$) for the cross-term approximation. For a learning rate $\eta < \frac{1}{L\sigma}$, the cumulative reduction is lower-bounded by:*

$$\Delta_{2,t} \geq C_1(\eta)\left(\|p_{t-1}\|^2 + \|p_t\|^2\right) + C_2(\eta)\|p_{t-1}\|\|p_t\|\mathcal{S}_{\text{inter}}^t, \tag{67}$$

*where $C_2(\eta) > 0$. Thus, maximizing the inter-step cosine $\mathcal{S}_{\text{inter}}^t$ strictly increases the guaranteed loss reduction by preventing trajectory cancellation.*

*Proof.* **Step 1: Two-step Descent Lemma.** By $L$-smoothness, the loss reduction over the total displacement $\Delta\theta = \theta_{t+1} - \theta_{t-1} = \eta(p_{t-1} + p_t)$ is bounded by:

$$\mathcal{L}(\theta_{t+1}) \leq \mathcal{L}(\theta_{t-1}) + \langle \nabla\mathcal{L}(\theta_{t-1}), \Delta\theta \rangle + \frac{L}{2}\|\Delta\theta\|^2. \tag{68}$$

Let $g_{t-1} = \nabla\mathcal{L}(\theta_{t-1})$ and $\Delta_{2,t} = \mathcal{L}(\theta_{t-1}) - \mathcal{L}(\theta_{t+1})$. Rearranging yields:

$$\Delta_{2,t} \geq \underbrace{-\eta\langle g_{t-1}, p_{t-1} + p_t \rangle}_{\text{Linear Gain}} - \underbrace{\frac{L\eta^2}{2}\|p_{t-1} + p_t\|^2}_{\text{Quadratic Penalty}}. \tag{69}$$

**Step 2: Bounding the Linear Term.** Recall $p_{t-1} = -P_{t-1}g_{t-1}$, so $g_{t-1} = -P_{t-1}^{-1}p_{t-1}$. The linear term splits into:

$$-\eta\langle g_{t-1}, p_{t-1} + p_t \rangle = \eta\left(p_{t-1}^\top P_{t-1}^{-1}p_{t-1} + p_{t-1}^\top P_{t-1}^{-1}p_t\right). \tag{70}$$

Using the spectral lower bound (consistent with Proposition 5), $p_{t-1}^\top P_{t-1}^{-1}p_{t-1} \geq \frac{1}{M}\|p_{t-1}\|^2$. For the cross-term, under the local isotropy assumption ($P_{t-1}^{-1} \approx \frac{1}{\sigma}I$), we approximate:

$$p_{t-1}^\top P_{t-1}^{-1}p_t \approx \frac{1}{\sigma}p_{t-1}^\top p_t = \frac{1}{\sigma}\|p_{t-1}\|\|p_t\|\mathcal{S}_{\text{inter}}^t. \tag{71}$$

**Step 3: Combining with Quadratic Penalty.** We expand the quadratic penalty norm $\|p_{t-1} + p_t\|^2$ using the cosine law. Substituting back:

$$\Delta_{2,t} \geq \left[\frac{\eta}{M}\|p_{t-1}\|^2 + \frac{\eta}{\sigma}\|p_{t-1}\|\|p_t\|\mathcal{S}_{\text{inter}}^t\right]$$
$$- \frac{L\eta^2}{2}\left[\|p_{t-1}\|^2 + \|p_t\|^2 + 2\|p_{t-1}\|\|p_t\|\mathcal{S}_{\text{inter}}^t\right]. \tag{72}$$

**Step 4: Grouping by Cosine.** Collecting the terms multiplied by $\mathcal{S}_{\text{inter}}^t$:

$$\Delta_{2,t} \geq C_{\text{mag}} + \underbrace{\left(\frac{\eta}{\sigma} - L\eta^2\right)}_{C_2(\eta)}\|p_{t-1}\|\|p_t\|\mathcal{S}_{\text{inter}}^t. \tag{73}$$

For the alignment coefficient $C_2(\eta)$ to be positive, we require $\eta < \frac{1}{L\sigma}$. Under this condition, a higher inter-step cosine $\mathcal{S}_{\text{inter}}^t$ strictly increases the lower bound of the cumulative loss reduction. $\qquad\square$

## A.14 NEWTON PRECONDITIONING AND OUTLIER SUPPRESSION

We detail the derivations supporting Sec. 5 and make explicit the assumptions under which SOAP (quasi-Newton) limits kurtosis growth relative to diagonal methods.

**Setup and identity.** Let $\mathbf{X} \in \mathbb{R}^{n \times d}$ with $m_2(\mathbf{X}) = 1$, and define $\Sigma_F = \mathbf{X}^\top\mathbf{X}$, $\Sigma_I = \mathbf{X}\mathbf{X}^\top$. By trace cyclicity, $\text{Tr}(\Sigma_F^2) = \text{Tr}(\Sigma_I^2)$. Writing $\Sigma_F$'s diagonal in terms of per-channel RMS $s_j^2 = \frac{1}{n}\sum_\alpha X_{\alpha j}^2$,

$$\sum_{j=1}^d (\Sigma_F)_{jj}^2 = \sum_{j=1}^d \left(\sum_{\alpha=1}^n X_{\alpha j}^2\right)^2 = n^2 \sum_{j=1}^d s_j^4 = n^2 d \cdot \text{Kurt}(\mathbf{X}), \tag{74}$$

since $\frac{1}{d}\sum_j s_j^2 = m_2(\mathbf{X}) = 1$. Hence equation 10 follows:

$$n^2 d \cdot \text{Kurt}(\mathbf{X}) + \sum_{i \neq j}(\Sigma_F)_{ij}^2 = \sum_{\alpha,\beta}(\Sigma_I)_{\alpha\beta}^2. \tag{75}$$

This identity holds *for any* $\mathbf{X}$ (not only at initialization), so it ties feature-wise kurtosis to input-wise correlation energy throughout training.

**Small-step bound.** Consider a linearized local map $\mathbf{X} = \mathbf{H}W$ at a given layer (holding the upstream activation $\mathbf{H}$ fixed during the step). A SOAP update gives $\Delta W = -\eta\, H_W^{-1}G$; thus $\Delta\mathbf{X} = \mathbf{H}\,\Delta W$.

Expanding $\|\mathbf{X} + \Delta\mathbf{X}\|_F^4$ to second order in $\eta$ and taking expectations over minibatches yields the $O(\eta^2)$ contribution

$$u_{4,2} \leq C\,nd\,\eta^2\,\|\mathbf{H}\|_2^2\,\|H_W^{-1}\|_2^2\,\|G\|_F^2, \tag{76}$$

for a constant $C$ independent of $\eta$ (the precise value depends only on fourth-moment combinatorics). Replacing $H_W^{-1}$ by a diagonal preconditioner's effective scaling $D^{-1/2}$ yields the analogous diagonal bound. Therefore, with identical $\eta$ and damping chosen so that $\|H_W^{-1}\|_2 \leq \|D^{-1/2}\|_2$ and then,

$$\mathbb{E}[\Delta\mathrm{Kurt}(\mathbf{X})]_{\mathrm{SOAP}} \leq \mathbb{E}[\Delta\mathrm{Kurt}(\mathbf{X})]_{\mathrm{Diag}} \tag{77}$$

holds up to negligible $O(\eta^3)$ terms. Intuitively, Newton preconditioning narrows the spread of per-direction step sizes by working in (and rotating back from) the curvature eigenbasis (Martens & Grosse, 2015; Gupta et al., 2018; Anil et al., 2020; Vyas et al., 2024). This curbs single-direction amplification and suppresses outliers, in line with our empirical findings.

## A.15 COMPARISON WITH OTHER OPTIMIZATION PARADIGMS

To contextualize the performance of SOAP, we evaluate it against a broader spectrum of optimization strategies, ranging from basic first-order methods to other advanced second-order approximations. The results are visualized in Fig. 10. Please note that for the compared optimizers, the hyperparameters (lr, momentum, and update frequency) are swept to get the best possible results.

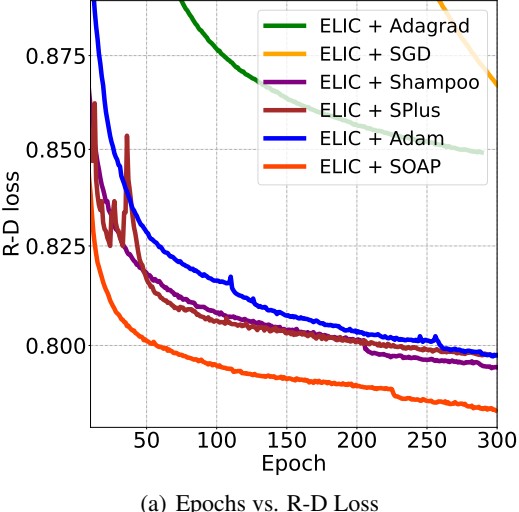

(a) Epochs vs. R-D Loss

Figure 10: **Comparison of Testing Loss for Various Optimizers.** *(Best viewed zoomed in.)* We compare SOAP against SGD (First-Order), Adagrad (Diagonal Root-Inverse), Shampoo (Structured Root-Inverse), and Adam. The first 10 epochs are omitted for clarity. Evaluation is conducted on the Kodak dataset with $\lambda = 0.013$.

**First-Order Methods (SGD).** SGD represents the baseline with no curvature information and no additional information estimation. Without the ability to rescale or rotate gradients based on the loss landscape geometry, it is theoretically unable to mitigate the gradient conflicts inherent in the R-D objective. Empirically, as expected, we observe that SGD performs worse than Adam and fails to reach a competitive rate-distortion performance within the same training period [9].

**Root-Inverse Methods ($H^{-1/2}$).** A distinct class of adaptive optimizers approximates the inverse square root of the Hessian ($H^{-1/2}$) rather than the full inverse ($H^{-1}$) used by Newton-like methods (SOAP). Theoretically, the full inverse is required to completely "whiten" the local landscape into a spherical shape where gradient alignment is maximized (see Sec. A.7). The square root inverse only partially corrects the curvature, which limits its ability to fully resolve intra-step conflicts.

---

[9]It is widely known that SGD requires much more steps to converge

- **Adagrad:** Adagrad (Duchi et al., 2011) approximates a diagonal $H^{-1/2}$ using the sum of squares of gradients. While it provides adaptive scaling, its diagonal formulation lacks the off-diagonal information to rotate updates and the full-inverse scaling to whiten them. Empirically, we found its performance consistently outperformed by Adam and SOAP, which is as expected due to the simple design mechanism of Adagrad.

- **Shampoo/SPlus:** Shampoo (Gupta et al., 2018) and SPlus (Frans et al., 2025) utilize Kronecker products to approximate a structured $H^{-1/2}$. While they capture more correlations than Adagrad, the root-inverse formulation still falls short of the perfect alignment offered by the full inverse. In our experiments, Shampoo/SPlus achieved slightly faster convergence than Adam but remained slower and less effective than SOAP. Furthermore, we observed the known instability issues (Anil et al., 2020); these methods required careful tuning and gradient crafting to avoid divergence, whereas SOAP served as a stable drop-in replacement.

**Muon (Momentum Orthogonal Optimizer).** Muon is an emerging optimizer designed for transport in LLMs that also conceptually approximates an orthogonalizing $H^{-1/2}$ update[10]. However, it faces specific structural challenges in LIC:

1. **Dimensionality Mismatch:** Muon is defined for 2D parameters (matrices). For 1D parameters (e.g., biases), it falls back to AdamW. Crucially, LIC models rely heavily on 4D Convolutional kernels ($C_{out} \times C_{in} \times K \times K$). To apply Muon, these must be flattened into 2D matrices (e.g., $C_{out} \times (C_{in} \cdot K \cdot K)$), potentially disrupting the spatial inductive bias.

2. **Divergence:** Despite extensive hyperparameter tuning (learning rates, momentum, and flattening strategies), we were unable to achieve stable convergence with Muon in the setting of learned image compression. We hypothesize that Muon's specific orthogonalization constraints may conflict with the initialization or dynamic range requirements of LIC modules. Future research is needed to adapt such constraints to convolutional architectures.

### A.16    WILL A LONGER TRAINING PERIOD MAKE ANY DIFFERENCE?

To ensure that the superior performance of SOAP is not simply due to the optimizer requiring more training steps to converge, we conducted ablation studies with extended training durations (up to 1000 epochs) using the ELIC model. The results are summarized in Table 4.

First, we observed that extending training beyond 300 epochs yields negligible improvements. This is because the `ReduceLROnPlateau` scheduler monitors the validation loss; by epoch 300, the learning rate has typically decayed to values less than $5 \times 10^{-6}$. At this magnitude, the optimization updates become small, and the model has effectively reached a stationary point. Consequently, training for 1000 epochs results in a statistically insignificant BD-Rate improvement compared to the 300-epoch baseline.

Second, to investigate if the specific choice of scheduler limited the baseline's convergence, we implemented a "Half Constant + Cosine" scheduler over 300 and 500 epochs. In this setting, the learning rate is held constant at the initial value for 150 or 250 epochs before undergoing cosine decay. Even with this prolonged period of high learning rate, the final converged rate-distortion performance did not show significant differences compared to the standard setting. These results confirm that the performance gap between SOAP and Adam is fundamental to how they navigate the optimization landscape—specifically SOAP's ability to resolve gradient conflicts—rather than a result of insufficient training time for the baseline.

### A.17    HOW DO ADAM AND SOAP INTERACT WITH TRAINING STABILIZERS?

Training models often rely on a suite of heuristic stabilizers to prevent divergence and ensure smooth convergence. Here, we investigate the sensitivity of Adam and SOAP to three common techniques: gradient clipping, learning rate warmup, and Exponential Moving Average (EMA). We use the ELIC model on the Kodak dataset ($\lambda = 0.013$) as the testbed.

---

[10]https://kellerjordan.github.io/posts/muon/

Table 4: Comparisons of ELIC performance trained with different durations and schedulers. Evaluated on Kodak.

| Optimizer | Epochs | Scheduler | BD-Rate vs. Baseline |
|---|---|---|---|
| Adam (Baseline) | 300 | ReduceOnPlateau | 0.00% |
| Adam | 1000 | ReduceOnPlateau | -0.02% |
| Adam | 300 | Half Constant + Cosine | -0.05% |
| Adam | 500 | Half Constant + Cosine | -0.10% |
| SOAP | 300 | ReduceOnPlateau | **-3.49%** |
| SOAP | 1000 | ReduceOnPlateau | **-3.51%** |

### A.17.1 GRADIENT CLIPPING

Gradient clipping is standard practice in LIC training to prevent exploding gradients. We train models with gradient clipping thresholds of $\{1.0, 5.0, \infty \text{ (no clipping)}\}$, where 1.0 is the default value following CompressAI (Bégaint et al., 2020).

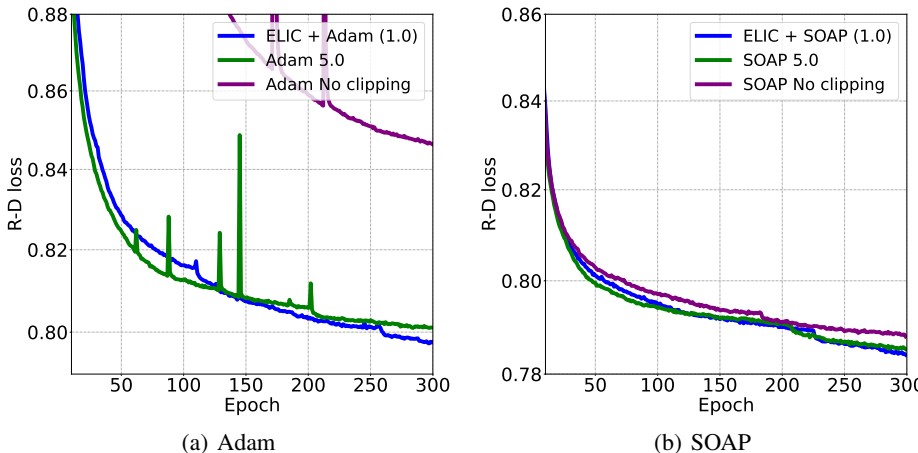

(a) Adam        (b) SOAP

Figure 11: **Comparison of Testing Loss: Epochs vs. R-D Loss under Different Gradient Clipping.** *(Best viewed zoomed in.)* The first 10 epochs are omitted for clarity. Evaluation is conducted on the Kodak dataset with $\lambda = 0.013$. The R-D loss is computed as $\lambda \cdot 255^2 \cdot \text{MSE} + \text{Bpp}$.

**Observation:** We observe distinct behaviors regarding sensitivity to gradient clipping. For Adam (Fig. 11(a)), the optimizer is more sensitive: removing clipping leads to immediate worse convergence results, and even relaxing the threshold to 5.0 results in noticeable training instability (loss spikes) and slightly suboptimal convergence. In contrast, SOAP (Fig. 11(b)) exhibits greater structural stability; it does not diverge even without explicit clipping, supporting the intuition that second-order preconditioning acts as an intrinsic normalization against curvature-induced explosions. However, while SOAP survives without clipping, its final R-D performance is slightly degraded compared to the clipped versions. Consequently, we find that the standard clipping threshold of 1.0 yields the best results for both optimizers, ensuring both stability and great convergence.

### A.17.2 LEARNING RATE WARMUP

Warmup strategies are typically employed to stabilize the variance of adaptive learning rates during the initial training phase. Interesting, it is not a widely used strategy in learned compressor training. We hypothesis it is because Adam is not sensitive to lr warmup as it adaptively scales the lr (Kingma & Ba, 2014). We evaluate performance by comparing training with no warmup (default) versus a linear warmup over the first 3 epochs.

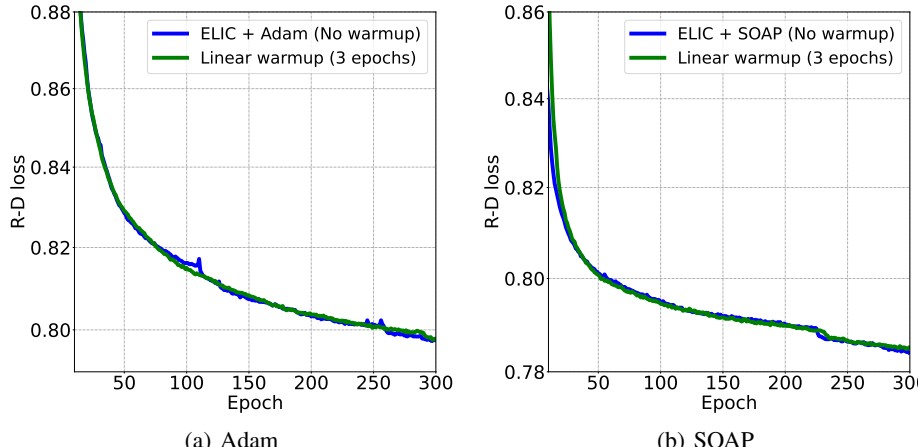

(a) Adam          (b) SOAP

Figure 12: **Comparison of Testing Loss: Epochs vs. R-D Loss under Different Warmup.** *(Best viewed zoomed in.)* The first 10 epochs are omitted for clarity. Evaluation is conducted on the Kodak dataset with $\lambda = 0.013$. The R-D loss is computed as $\lambda \cdot 255^2 \cdot \text{MSE} + \text{Bpp}$.

**Observation:** In our experiments, we observe that both Adam and SOAP are insensitive to the learning rate warmup strategy in this setting, as illustrated in Fig. 12. Neither optimizer exhibited significant differences in convergence or final rate-distortion performance when warmup was removed or applied. This suggests that the initial optimization dynamics are sufficiently stable to allow immediate training at the base learning rate without necessitating a gradual ramp-up.

### A.17.3 EXPONENTIAL MOVING AVERAGE (EMA)

EMA maintains a shadow copy of the model parameters with a decay factor (typically 0.999) to smooth out optimization noise and improve generalization. The dynamics and benefits of EMA have been extensively characterized by Morales-Brotons et al. (2024), who highlight its ability to stabilize training and improve final convergence even when the underlying optimization trajectory is noisy.

**Observation:** As shown in Fig. 13, applying EMA proves beneficial for both Adam and SOAP, though their underlying dynamics differ. For Adam, the raw training trajectory is highly oscillatory due to the destructive interference of gradients (as discussed in Sec. 4); consequently, EMA is critical to filtering this noise and revealing the true performance of the model. For SOAP, the raw optimization trajectory is significantly smoother, validating our theoretical findings on inter-step alignment, yet EMA still provides a consistent improvement in the final R-D performance. This suggests that while SOAP effectively resolves optimization conflicts, the regularization effect of EMA remains valuable for maximizing generalization performance.

Finally, regarding computational cost, we find that maintaining EMA shadow weights introduces negligible overhead. For example, one epoch of training takes approximately 7 minutes 20 seconds without EMA compared to 7 minutes 30 seconds with EMA for ELIC model with our setting, an increase of roughly 2% time. Given this minimal negligible cost relative to the universal performance gains observed, we employ EMA by default for all experiments in this work.

### A.18 EVOLUTION OF FEATURE OUTLIERS DURING TRAINING

To gain deeper insight into the genesis of the outlier features discussed in Sec. 5, we track the Kurtosis and MaxMed statistics of the latent representation $z$ throughout the entire training process. While Table 2 reports the final converged metrics, analyzing their trajectories reveals distinct differences in how Adam and SOAP manage feature dynamic ranges during the critical initial learning phases.

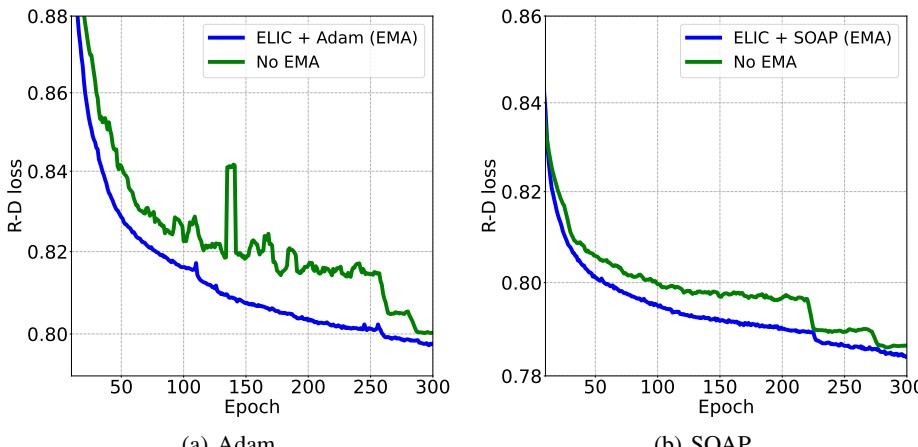

(a) Adam       (b) SOAP

Figure 13: **Comparison of Testing Loss: Epochs vs. R-D Loss under Different EMA.** *(Best viewed zoomed in.)* The first 10 epochs are omitted for clarity. Evaluation is conducted on the Kodak dataset with $\lambda = 0.013$. The R-D loss is computed as $\lambda \cdot 255^2 \cdot \mathrm{MSE} + \mathrm{Bpp}$.

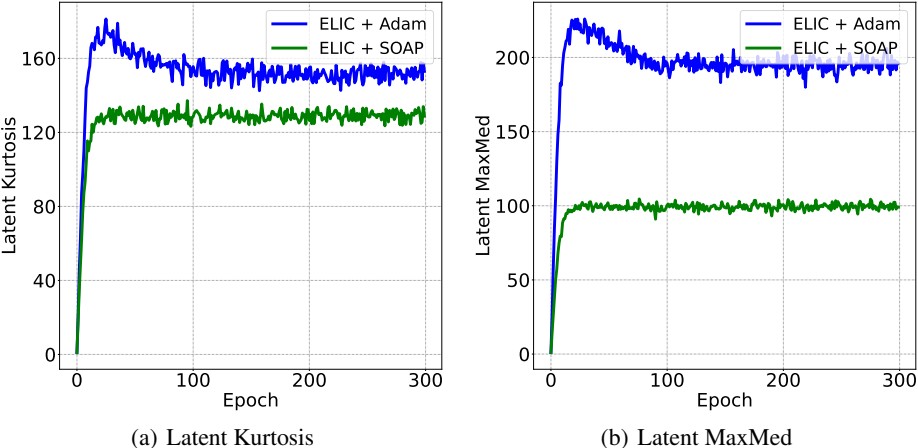

(a) Latent Kurtosis       (b) Latent MaxMed

Figure 14: **Evolution of Latent Outlier Metrics during Training (ELIC).** We track (a) Kurtosis and (b) MaxMed of the latent representation $z$ over 300 epochs. Both metrics rise during the initial feature learning phase (epochs 0–20). However, **Adam (Blue)** allows these metrics to spike to extreme levels and plateau there, indicating the permanent formation of high-magnitude outliers. **SOAP (Green)** significantly suppresses this growth, stabilizing at a much lower plateau—especially in MaxMed, where the peak deviation is nearly halved.

**Observation:** As illustrated in Figure 14, the training dynamics of the two optimizers diverge immediately during the early epochs:

- **Adam (Unchecked Growth):** Training with Adam leads to a rapid accumulation of outlier features within the first 20 epochs. Because Adam restricts preconditioning to coordinate-wise scaling, it cannot rotate the optimization basis to redistribute large gradients. Consequently, specific channels absorb excessive update energy, causing Kurtosis to spike to $\approx 160$ and MaxMed to exceed 200. Once these outliers are established, the model settles into a high-magnitude plateau, locking in the heavy-tailed distribution for the remainder of training.

- **SOAP (Active Suppression):** While SOAP also exhibits an initial rise as features are learned, it strictly bounds the magnitude of these outliers. The curvature-aware preconditioning effectively "diffuses" the update energy across coupled channels (as derived in Sec. 5.2). This results in a consistently lower plateau. The effect is most pronounced in the MaxMed metric (Fig. 14(b)), where SOAP reduces the peak outlier magnitude by approximately $50\%$ compared to Adam ($\approx 100$ vs. $\approx 200$). This confirms that SOAP's robustness to quantization is rooted in its ability to prevent extreme outliers from forming during the early optimization trajectory.

