# OpenReview forum: "Towards Second-Order Optimization in Learned Image Compression: Faster, Better, and More Deployable"
_ICLR.cc/2026/Conference — Submitted to ICLR 2026_

### Official Review · Reviewer_DAjx · 2025-10-25

**Soundness:** 4
**Presentation:** 4
**Contribution:** 3
**Rating:** 8
**Confidence:** 4

**Summary:**

This paper proposes using a second-order optimizer, Shampoo with Adam in the Conditioner (SOAP), instead of first-order optimizers, like Adam, to train Learned Image Compression (LIC) models. The paper finds that this optimizer results in faster training, better rate-distortion (RD) curves, and models that are more robust to quantization. Experiments include comparisons of state-of-the-art LIC models with Adam and with SOAP optimization as well as analysis of how gradients change during training.

**Strengths:**

1. The proposed method shows strong empirical results. Specifically, it is great to see both convergence time and RD metrics improving simultaneously.
2. The paper includes strong empirical evidence that this improvement occurs over a wide variety of LIC models and test datasets. Furthermore, this method works better than other recently proposed methods to solve training convergence issues.
3. The paper is clearly written and most claims are clearly validated by empirical or theoretical evidence. The concept of inter-step and intra-step scores was neat and useful for understanding how gradients change during training.
4. The section on evaluating the effect of this optimizer on quantized LIC models is neat and useful for potential deployment of compressed LIC models.

**Weaknesses:**

1. SOAP is an existing optimizer, so replacing Adam with SOAP has limited novelty.
2. The theoretical results rely on strong assumptions, like $L_H$-Lipschitz and local smoothness. It'd be interesting to know whether these hold for the models under investigation.

**Questions:**

1. Have you considered training the SOAP models longer than 300 epochs? From Figure 1, it looks like the loss might still be decreasing. I'm curious if you could see even more improvements in RD curves with longer training?
2. How do other optimizers perform? For example, Shampoo, SGD, AdamW.

---

> ### Author Response · Authors · 2025-11-21
> **Response to Reviewer DAjx (1)**
>
> **W1.** Contribution
>
> **Response.**
>
> We thank the reviewer for this comment and appreciate the opportunity to clarify the scope and nature of our contributions.
>
> We explicitly acknowledge that we did not invent the SOAP optimizer. If our contribution were merely applying an existing optimizer to a new task, we would agree that the novelty is limited. However, our work is motivated by a fundamental problem in Learned Image Compression (LIC): the optimization landscape is dominated by the intrinsic conflict between two competing objectives—Rate and Distortion (R-D). Our contribution is a comprehensive study revealing *why* standard first-order methods (like Adam) struggle with this landscape and how efficient Newton-style preconditioning fundamentally alters the training dynamics and the resulting representation quality.
>
> **Novel Theoretical Analysis of R-D Gradient Conflicts.**
>
> We provide a theoretical framework specific to the R-D objective (Section 4). We analyze the optimization trajectory through the lens of gradient conflicts. We theoretically derive and empirically validate that while Adam often exhibits negative cosine similarity between Rate and Distortion updates (indicating oscillating, conflicting gradients), SOAP’s curvature-aware preconditioning achieves high positive alignment. We decompose this into intra-step alignment (resolving conflict between R and D within a step) and inter-step alignment (stabilizing the trajectory over time). This mechanistic explanation of how curvature information resolves multi-objective conflicts in LIC is a novel contribution to the compression literature.
>
> **Discovery of Latent Outlier Suppression and Quantization Robustness.**
>
> Another significant and novel finding of our work (Section 5) is that the choice of optimizer dictates the statistical properties of the learned latent space. We discovered that models trained with Newton-style preconditioning exhibit significantly fewer outliers in their latent representations compared to Adam-trained models. We provide a theoretical justification for this based on "correlation energy" redistribution. Crucially, we show this has a major downstream impact: SOAP-trained models are substantially more robust to Post-Training Quantization (PTQ), a critical hurdle for deploying LICs on edge devices. Connecting the optimization algorithm to the quantization robustness of the entropy model is a new insight that goes beyond simple "acceleration."
>
> **Practical Impact.**
>
> Finally, while "replacing Adam" sounds simple, establishing that a specific class of second-order methods can serve as a drop-in replacement that outperforms complex, specialized training strategies (like Auxiliary Transforms or Balanced R-D, as detailed in Appendix A.3) is a valuable contribution. We show that one does not need to redesign the architecture or the loss function to solve R-D conflicts; one simply needs an optimizer that respects the curvature of the R-D landscape.
>
> In summary, our novelty lies in the *analysis* of R-D specific optimization dynamics and the *discovery* of the link between second-order optimization and quantization-friendly latent representations, rather than in the proposal of the optimizer itself.

---

> ### Author Response · Authors · 2025-11-21
> **Response to Reviewer DAjx (2)**
>
> **W2.** strong assumptions
>
> **Response.**
>
> We thank the reviewer for raising this important point regarding the theoretical assumptions. We agree that assumptions such as $L$-Lipschitz continuity and local smoothness are strong idealizations, particularly for deep neural networks that utilize non-linearities like ReLUs. Indeed, verifying whether these properties hold strictly for complex architectures like ELIC or TCM is a significant challenge.
>
> **Standard assumptions in optimization analysis.**
>
> While we acknowledge this limitation, we note that these assumptions are standard practice in the theoretical analysis of optimization algorithms ([1], [2], [3]). Almost all convergence guarantees and alignment analyses for first-order (Adam) and second-order methods rely on local quadratic approximations and smoothness to derive tractable bounds. As the reviewer notes, these serve as a mathematical model to build intuition about the optimization dynamics, rather than an exact description of the global landscape.
>
> **Difficulty of verification.**
>
> Empirically verifying these assumptions for modern LIC models is computationally prohibitive. Estimating the exact Lipschitz constant or the local smoothness requires computing the spectral norm of the full Hessian matrix. For models with millions of parameters (like the Transformers and CNNs used in our study), this is computationally intractable and constitutes a non-trivial research problem in its own right ([4]). A rigorous quantification of the "curvature drift" or smoothness violations in LIC models would indeed be a valuable contribution, but it lies beyond the scope of the current work, which focuses on the comparative dynamics of first- vs. second-order updates.
>
> Reference:
>
> [1]. Wright, S., & Nocedal, J. (1999). Numerical optimization. Springer Science, 35(67-68), 7.
>
> [2]. Kingma, D. P., & Ba, J. (2014). Adam: A method for stochastic optimization. arXiv preprint arXiv:1412.6980.
>
> [3]. Reddi, S. J., Kale, S., & Kumar, S. (2019). On the convergence of adam and beyond. arXiv preprint arXiv:1904.09237.
>
> [4]. Virmaux, A., & Scaman, K. (2018). Lipschitz regularity of deep neural networks: analysis and efficient estimation. Advances in Neural Information Processing Systems, 31.
>
>
>
> ------
>
>
>
> **Q1.** longer training period
>
> **Response.** We thank the reviewer for this insightful observation. While the loss curves in Fig. 1 may visually suggest a slight downward trend near epoch 300, the models have effectively reached stationarity in terms of Rate-Distortion performance. This is because the `ReduceLROnPlateau` scheduler typically decays the learning rate to extremely small values (less than $5 \times 10^{-6}$) by this stage, rendering subsequent updates negligible in practice and near 300 epoch is last decrease period.
>
> To rigorously address the possibility that longer training or different scheduling might bridge the gap between Adam and SOAP, we conducted extensive ablation studies, which we have added to **Appendix A.16** of the revised manuscript. We explored two dimensions: extending the training duration up to 1000 epochs, and employing a ‘Half Constant + Cosine’ scheduler. The latter scheduler maintains the initial high learning rate for the first 50% of the total epochs before applying a cosine decay, thereby forcing the optimizer to explore the landscape more thoroughly than the aggressive decay of `ReduceLROnPlateau`.
>
> As detailed in Table below, extending the training of SOAP from 300 to 1000 epochs yields a negligible additional gain (moving from $-3.49$% to $-3.51$% BD-Rate), confirming that SOAP converges efficiently within the standard budget. Crucially, even when we provide Adam with a more aggressive scheduler (Half Constant + Cosine) and extend its training to 500 epochs, the performance improvement is marginal ($-0.10$%). This stands in stark contrast to the $-3.49$% improvement SOAP achieves in just 300 epochs. These results confirm that the performance gap is not an artifact of training time or scheduling choices, but stems from the second-order optimizer's ability to resolve gradient conflicts and locate a fundamentally better minimum.
>
> *Comparisons of ELIC performance trained with different durations and schedulers. Evaluated on Kodak.*
>
> | **Optimizer**   | **Epochs** | **Scheduler**          | **BD-Rate vs. Baseline** |
> | --------------- | ---------- | ---------------------- | ------------------------ |
> | Adam (Baseline) | 300        | ReduceOnPlateau        | 0.00%                    |
> | Adam            | 1000       | ReduceOnPlateau        | -0.02%                   |
> | Adam            | 300        | Half Constant + Cosine | -0.05%                   |
> | Adam            | 500        | Half Constant + Cosine | -0.10%                   |
> | SOAP            | 300        | ReduceOnPlateau        | **-3.49%**               |
> | SOAP            | 1000       | ReduceOnPlateau        | **-3.51%**               |

---

> ### Author Response · Authors · 2025-11-21
> **Response to Reviewer DAjx (3)**
>
> **Q2.** How do other optimizers perform? For example, Shampoo, SGD, AdamW.
>
> Response:
>
> We thank the reviewer for this constructive suggestion. In response, we have conducted a comprehensive evaluation of these optimizers. The detailed results and discussions are provided in Appendix A.5 (for AdamW) and the newly added Appendix A.15 (for SGD, Shampoo, etc.).
>
> **AdamW:** As detailed in Appendix A.5, AdamW is mathematically equivalent to Adam when weight decay is zero. We performed an ablation study on the ELIC model with weight decay values of $\{0.01, 0.001, 0.0001\}$ using AdamW. We found that introducing weight decay yielded results that were either similar to or slightly more unstable than training without it. Consequently, the optimal configuration for this baseline is with weight decay set to $0$, making our Adam comparison effectively cover the optimal AdamW performance.
>
> **Shampoo:** We evaluated Shampoo as a representative of structured second-order methods (Appendix A.15). While Shampoo converged slightly faster than Adam, it remained slower and achieved worse Rate-Distortion performance than SOAP. Theoretically, we attribute this to the fact that Shampoo approximates the inverse *square root* of the Hessian ($H^{-1/2}$), whereas SOAP approximates the *full inverse* ($H^{-1}$). The full inverse is required to completely "whiten" the local landscape and resolve the gradient conflicts described in Section 4; the root-inverse only partially corrects the curvature. Empirically, we also observed the numerical instability often associated with Shampoo in this domain.
>
> **SGD and Adagrad:** As shown in Figure 10, SGD performed poorly, failing to reach competitive rate-distortion performance, which underscores the necessity of adaptive scaling in the complex LIC landscape. Adagrad provided stable training but was consistently outperformed by Adam and SOAP, confirming that simple diagonal approximations are insufficient to resolve the non-axis-aligned gradient conflicts inherent to the Rate-Distortion objective.
>
> **Muon:** We also attempted to apply the emerging Muon optimizer. However, because Muon is designed for 2D matrices (linear layers in LLMs), applying it to the 4D convolutional kernels prevalent in LICs resulted in a dimensionality mismatch that disrupted spatial inductive biases. We were unable to achieve stable convergence with Muon despite extensive tuning.
>
> In summary, SOAP demonstrates superior performance and stability compared to this broad spectrum of alternatives by effectively approximating the full Newton step to resolve gradient conflicts, while functioning as a seamless drop-in replacement.

---

> ### Author Response · Authors · 2025-11-26
>
> Dear Reviewer DAjx,
> As we approach the end of the discussion period, we wanted to make sure you had a chance to review the additional experiments and clarifications we added in response to your questions on convergence, baselines, and assumptions.
>
> *Longer training (Appendix A.16).* Following your suggestion, we extended Adam training to 1000 epochs. Adam improves by only (-0.02%) BD-Rate with 700 extra epochs, while SOAP maintains a (~ 3.5%) advantage, indicating that SOAP’s gains come from better optimization rather than training the baseline too briefly.
>
> *Other optimizers (Appendix A.5 and A.15).* We added comparisons against Shampoo, SGD, Adagrad, and AdamW. Shampoo, which approximates $(H^{-1/2})$, is less effective and less stable than SOAP’s full inverse approximation $(H^{-1})$ for the R–D landscape; SGD and Adagrad underperform, and AdamW with tuned weight decay performs similarly to or slightly worse than the Adam baseline.
>
> *Novelty and assumptions.* We clarified that our main contribution lies in the R–D-specific theoretical analysis of gradient conflicts and in the discovery that SOAP suppresses latent/activation outliers (Section 5), thereby improving quantization robustness. We also expanded the discussion of standard smoothness and Lipschitz assumptions and their role as local models of the loss landscape.
>
> We hope these additional analyses and ablations help strengthen your assessment of the work. Thank you again for your constructive and encouraging review.
>
> Best regards,
> The Authors

---

### Official Review · Reviewer_8Sdb · 2025-10-30

**Soundness:** 3
**Presentation:** 3
**Contribution:** 2
**Rating:** 4
**Confidence:** 5

**Summary:**

This paper investigates the advantages of the existing second-order optimizer SOAP for training learned image codecs, including (1) Assist in avoiding conflicts between different optimization objectives of RD within mini batches; (2) Help avoid gradient conflicts between different samples across mini-batches. (3) Help eliminate outliers in the latent and activation layers, and facilitate PTQ. In terms of final results, validation on models such as ELIC, TCM, LALIC, and DCAE found that the training appear more stable, resulting in a BD-Rate gain of about 3%. The performance after PTQ training was 1% -2% higher than the prior Adam training.

**Strengths:**

- From the results reported in the paper, it can be seen that using the SOAP optimizer has indeed saved training time, improved convergence performance, and assisted with the implementation of PTQ

**Weaknesses:**

- The paper only analyzes the advantages of using SOAP in LIC, without making any methodological improvements or targets analysis for the Compression task
- Some of the details in the paper are unclear, e.g., if the performance comparison conditions are consistent, and if the 300 epoch model fully converges, etc. See "Questions".
- A large number of "See Appendix" jumps, but there is no concise explanation of the relevant content in the main text, resulting in poor readability and logical fragmentation

**Questions:**

- We want to know if this 3% performance gain is compared to the original performance in models' paper or based on the final convergence performance of the Adam optimizer under the same training conditions. In addition, how does the performance of the original model compare according to the training configuration in the paper(such as EMA, Batch size set to, etc.)
- The paper claims that all models are trained for 300 epochs to ensure complete convergence. However, according to Fig.1, the model did not fully converge at 300 epochs.
- I wonder if the SOAP optimizer is robust enough to alleviate gradient conflicts: (1) For multiple optimization objectives within a mini batch, such as R&D, whether this effect is universally applicable when the amplitude and convergence speed differences of different optimization objectives are more diverse, such as with different R-D balance (lambda). (2) For gradient consistency between mini batches, whether it can be equally effective in small batch sizes, as batch size set to 64 in the paper requires too much GPU memory (H100 is used in the paper, which is too expensive for training learned image codecs). And also when the batch size is large, the randomness of samples is smaller, and the differences between samples between mini batches are more stable, these will also help mini batch gradients tend towards smaller sizes.
- I wonder if the EMA used in the paper is necessary for SOAP, as many codecs, especially lightweight CNN based ones, have weak performance gains with EMA, which slows down their training speed. At the same time, I also wonder how wild the training without the average effect of the EMA, and what is the effect of SOAP at this time.
- In Figure 3, after more training steps, does the gradient consistency still maintain the current trend, especially in the first 10000 steps where inter-step score fluctuates but seems to show an overall upward trend? Will Adam's performance be similar to SOAP after this initial training period?
- For the outliers in Latent, I think more in-depth analysis and visualization are needed, including the changes in this outlier during the training process, comparing the distribution of this outlier with the same training loss alignment, and so on.
- The paper only compares with Adam optimizer, there are also some learned image/video codecs that use AdamW for optimization. I wonder if there are any relevant experiments or analyses
- Regarding the application of SOAP to other types of data compression models, such as videos, gradient conflicts/error accumulation may also occur temporally. However, it also involves an allocation optimization problem trained on this dimension (such as frame level bitrate allocation in videos). Does the SOAP optimizer have any advantages in solving such problems?
- This is an innovative and practical research, but I have some doubts about the practicality and actual performance of this new optimizer, which is a fundamental component in network training. If the author can solve these problems, I will increase my score.

---

> ### Author Response · Authors · 2025-11-21
> **Response to Reviewer 8Sdb (1)**
>
> **W1.** methodological improvements or targets analysis
>
> Response:
>
> We thank the reviewer for this comment. We respectfully disagree with the assessment that our work lacks target analysis for the compression task. While we do not propose a new network architecture module (e.g., a new attention block), our contribution is a paradigm shift in the optimization strategy for LIC, supported by a novel, compression-specific theoretical analysis and the discovery of previously unknown phenomena regarding latent representations.
>
> We clarify our contributions regarding "methodological improvements" and "target analysis for compression" as follows:
>
> 1. The "Methodological Improvement" is the Discovery of Curvature-Aware R-D Optimization.
>
> In the current LIC literature, the standard practice is to default to Adam. Our work challenges this by demonstrating that the choice of optimizer is not merely a hyperparameter, but a fundamental determinant of how the Rate-Distortion (R-D) trade-off is navigated. The "improvement" we propose is the switch to efficient quasi-Newton optimization (SOAP), which we show serves as a seamless drop-in replacement that outperforms complex, tailored training strategies like Auxiliary Transforms (AuxT), CMD-LIC, or Balanced R-D (see Appendix A.3 and Table 3). This is a significant methodological contribution: we provide a simpler, faster, and more effective training pipeline that can be applied to any existing LIC architecture, achieving an average 3% BD-Rate improvement and 70% step reduction across four diverse architectures (ELIC, TCM, LALIC, DCAE).
>
> 2. Target Analysis 1: Resolving the Intrinsic R-D Gradient Conflict (Sec. 4).
>
> Section 4 is entirely dedicated to the specific dynamics of the R-D objective ($\mathcal{L} = \mathcal{L}_R + \lambda \mathcal{L}_D$). We do not simply apply SOAP; we derive a theoretical framework to explain why Newton-style preconditioning is uniquely suited to the R-D problem.
>
> Novel Insight: We identify that the Rate and Distortion objectives often generate conflicting gradient updates ("destructive interference"), leading to negative intra-step cosine similarity as shown in Figure 3. In the newly added Section 4.2 and Appendix A.13 (Proposition 5), we formally prove that maximizing the alignment between these update vectors strictly increases the lower bound of the loss reduction. We show that SOAP's curvature-aware preconditioning aligns these conflicting updates (intra-step alignment) and stabilizes the trajectory (inter-step alignment). This analysis is specific to multi-objective R-D optimization and constitutes a novel contribution to the theory of training learned compressors.
>
> 3. Target Analysis 2: Latent Outliers and Quantization Robustness (Sec. 5).
>
> Our analysis extends deeply into the properties of the compressed representation itself, which is central to the compression task.
>
> Discovery: We uncover a crucial phenomenon: second-order optimization systematically suppresses outliers in the latent space (Figure 4, Table 2). We provide a theoretical explanation based on "correlation energy" redistribution via the conservation law in Signal Propagation theory (Eq. 10 in Section 5.2).
>
> Compression Relevance: This is not just an optimization statistic; it directly impacts the entropy model and the model's deployability. We show that this outlier suppression leads to significantly improved robustness to Post-Training Quantization (PTQ), a critical hurdle in deploying compression models on edge devices. By proving that SOAP minimizes kurtosis growth (Eq. 11), we connect the choice of optimizer directly to the practical deployment of codecs, a target analysis that is absent in standard LIC literature.
>
> Summary.
>
> Our work goes beyond simply "using SOAP." We identify why the standard optimization approach (Adam) is suboptimal for the conflicting nature of Rate and Distortion, and we provide a comprehensive analysis of how curvature information resolves these conflicts and regularizes latent statistics for better quantization. We believe that discovering fundamental behaviors of the R-D objective and establishing a superior, generic training protocol is a substantial contribution to the field.
>
>
>
>
>
> ------
>
>
>
> **W3.** Improved logic and writing
>
> **Response:** We thank the reviewer for this constructive feedback. In the revised manuscript, we have ensured that the main text provides concise, intuitive explanations of the relevant content and key theoretical insights before referencing the Appendix.

---

> ### Author Response · Authors · 2025-11-21
> **Response to Reviewer 8Sdb (2)**
>
> **W2 & Q1.** Performance comparison and original performance
>
> **Response:** We confirm that the reported 3% BD-Rate gain is calculated relative to the **final convergence performance of the Adam optimizer trained under the exact same conditions** (same batch size, scheduler, EMA, epochs, etc.), rather than the numbers reported in the original papers. This ensures a strict ablation where the choice of optimizer is the only variable.
>
> Also, our Adam-trained models actually outperform the originally reported results. For instance, taking our Adam-trained ELIC model as the anchor (0.0% reference) on the Kodak dataset, our SOAP-trained ELIC achieves a **-3.49%** BD-Rate (significantly better). In contrast, the official ELIC model results (e.g., reported from the CompressAI https://github.com/InterDigitalInc/CompressAI/blob/master/results/image/kodak/paper-elic2022_mse.json) against our Adam anchor yields a **+1.8%** BD-Rate (worse than our baseline). This improvement in our baseline is largely attributable to our different setting (EMA, batch size, learning rate and training samples). Thus, the gains provided by SOAP are achieved on top of a highly competitive baseline, confirming their significance.
>
>
>
> ------
>
>
>
> **Q2.** Fig.1?
>
> **Response:** We thank the reviewer for this careful observation. We respectfully clarify that while the loss curves in Fig. 1 may appear to have a slight downward trend near epoch 300, the models have effectively reached a stationary point in terms of rate-distortion performance. By epoch 300, the `ReduceLROnPlateau` scheduler has typically decayed the learning rate to extremely small values (less than $5 \times 10^{-6}$), meaning any subsequent loss reduction is negligible in practice.
>
> To empirically verify this and address the concern, we conducted an additional ablation study (added to **Appendix A.16** in the revised paper) where we extended the training of the ELIC model with Adam to **1000 epochs**. As shown in the Table below, the performance difference between 300 and 1000 epochs is statistically insignificant ($-0.02$% BD-Rate). In contrast, switching to SOAP yields a massive $-3.49$% improvement within the standard 300 epochs.
>
> This confirms that the performance gap is due to SOAP's superior optimization of the landscape, not insufficient training time for the baseline.
>
> **Table 1:** Comparisons of ELIC performance trained with different durations and schedulers. Evaluated on Kodak.
>
> | **Optimizer**   | **Epochs** | **Scheduler**          | **BD-Rate vs. Baseline** |
> | --------------- | ---------- | ---------------------- | ------------------------ |
> | Adam (Baseline) | 300        | ReduceOnPlateau        | 0.00%                    |
> | Adam            | 1000       | ReduceOnPlateau        | -0.02%                   |
> | Adam            | 300        | Half Constant + Cosine | -0.05%                   |
> | Adam            | 500        | Half Constant + Cosine | -0.10%                   |
> | SOAP            | 300        | ReduceOnPlateau        | **-3.49%**               |
> | SOAP            | 1000       | ReduceOnPlateau        | **-3.51%**               |
>
> ------
>
>
> **Q3.** lambda and smaller bs
>
> Response:
>
> We thank the reviewer for these insightful questions regarding the robustness of our strategy.
>
> (1) Robustness to different R-D balances ($\lambda$).
>
> The effectiveness of SOAP in alleviating gradient conflicts holds regardless of the specific $\lambda$ value. In our theoretical framework, the loss is defined as $\mathcal{L} = \mathcal{L}_R + \lambda \mathcal{L}_D$. Mathematically, $\lambda$ acts as a scalar multiplier that affects the magnitude of the distortion gradient but not the underlying curvature structure of the loss landscape. Our analysis could treat the scaling factor as absorbed into the distortion term; thus, the mechanism by which the preconditioner rotates the basis to align update vectors remains valid across different scales. Empirically, the BD-Rate metrics reported in our paper are calculated by aggregating results from models trained with six different $\lambda$ values (ranging from $0.0018$ to $0.0483$). The improvements confirm that SOAP is effective across diverse trade-offs.
>
> (2) Robustness to smaller batch sizes.
>
> To address the reviewer's concern, we verified that SOAP remains effective at smaller batch sizes suitable for consumer-grade hardware (e.g., NVIDIA RTX 4090). We conducted an additional experiment training the ELIC model with a batch size of 32.
>
> Using the Adam-trained model (Batch Size 64) as the anchor, the model trained with Adam with Batch Size 32 achieves 0.07% BD-Rate, indicating that reducing the batch size negligibly impacts performance. The model trained with SOAP at Batch Size 32 achieves -3.42% BD-Rate. This is negligibly impacted compared to the -3.49% improvement obtained with Batch Size 64. These results demonstrate the effectiveness under smaller batches, making it a practical solution for consumer-grade hardware.

---

> ### Author Response · Authors · 2025-11-21
> **Response to Reviewer 8Sdb (3)**
>
> **Q4.** EMA usage
>
> Response:
>
> We appreciate the reviewer's question regarding the necessity and impact of EMA. To address this, we conducted an ablation study comparing Adam and SOAP both with and without EMA, using the ELIC model on the Kodak dataset ($\lambda = 0.013$). The detailed results and training curves are presented in the newly added Appendix A.17.3 and Figure 13.
>
> Impact of removing EMA:
>
> For Adam, removing EMA leads to a highly oscillatory training trajectory (Fig. 13a), confirming that first-order optimization struggles with noise and instability in the R-D landscape. EMA is critical here to smooth out these fluctuations and recover a stable model.
>
> For SOAP, the training trajectory without EMA (Fig. 13b) is significantly smoother than Adam's, which validates our theoretical finding that SOAP's preconditioning inherently stabilizes the optimization path (high inter-step alignment). However, enabling EMA still yields a consistent improvement in the final converged Rate-Distortion performance. This suggests that while SOAP effectively resolves gradient conflicts and reduces trajectory noise, the regularization provided by EMA remains beneficial for maximizing generalization, consistent with findings in other domains[1].
>
> computational cost:
>
> We also measured the overhead of EMA. In our setting with the ELIC model, one epoch takes approximately 7m 20s without EMA and 7m 30s with EMA—an increase of only $\approx 2$%. Given the negligible cost and the universal performance gain (even for the already-stable SOAP), we recommend using EMA as a default stabilizer. Nevertheless, SOAP's superior stability means it is much less dependent on EMA to prevent divergence compared to Adam.
>
>
> Reference:
> [1]. Morales-Brotons, D., Vogels, T., & Hendrikx, H. (2024). Exponential moving average of weights in deep learning: Dynamics and benefits. arXiv preprint arXiv:2411.18704.
>
>
> ------
>
>
>
>
>
> **Q5.** Figure 3,
>
> Response:
>
> We thank the reviewer for this observation. To address the question about long-term behavior, we extended the tracking of gradient consistency metrics in Figure 3 from 10,000 steps to 100,000 steps. We actually do not observe that Adam's inter-step score shows an overall upward trend as training progresses, and its behavior remains fundamentally different from SOAP. Specifically, Adam's alignment scores continue to exhibit significantly high-frequency fluctuations and chaotic behavior throughout the extended training period. In contrast, SOAP maintains a consistently high and stable alignment score. This confirms that Adam does not eventually "catch up" to SOAP's stability; rather, the lack of curvature information results in persistent oscillatory dynamics.
>
>
>
> ------
>
>
>
> **Q6.** changes in this outlier during the training process
>
> **Response:** We thank the reviewer for this constructive suggestion.
>
> In the revised manuscript, we have added a new subsection **Appendix A.18: Evolution of Feature Outliers During Training** along with **Figure 14** to address this. We tracked the Kurtosis and MaxMed of the latent representation throughout the entire 300-epoch training run.
>
> Summary of Results:
>
> Our analysis reveals that the divergence in outlier behavior occurs very early in the training process (within the first 20 epochs):
>
> **Adam (Spike and Plateau):** The optimizer allows feature statistics to spike sharply during the initial learning phase. These outliers persist, causing the metrics to plateau at high magnitudes (e.g., MaxMed $\approx 200$) for the remainder of training.
>
> **SOAP (Active Suppression):** In contrast, SOAP effectively limits this initial growth. While feature statistics rise as the model learns, they stabilize at a significantly lower plateau. Notably, the MaxMed metric for SOAP is nearly half that of Adam ($\approx 100$ vs. $\approx 200$).
>
> These results confirm that SOAP's superior robustness to post-training quantization is rooted in its ability to actively suppress the formation of extreme outliers throughout the optimization trajectory, rather than merely reducing them at convergence. Please refer to **Figure 14** and **Appendix A.18** for the detailed visualization and discussion.

---

> ### Author Response · Authors · 2025-11-21
> **Response to Reviewer 8Sdb (4)**
>
> **Q7.** AdamW
>
> **Response:** We thank the reviewer for this question. We respectfully point out that an experimental comparison with AdamW and an analysis of weight decay were included in our initial submission. Please refer to **Appendix A.5: WILL WEIGHT DECAY MAKE ADAM/ADAMW DIFFERENT?**.
>
> To summarize the relationship between these optimizers and our findings:
>
> Relationship between Adam and AdamW: AdamW (loshchilov2017decoupled) is a variant of Adam that decouples weight decay from the gradient update. Importantly, when the weight decay coefficient is set to $0$, Adam and AdamW are mathematically equivalent.
>
> **Experimental Results:** As detailed in Appendix A.5 and Figure 8, we compared the baseline Adam (weight decay $=0$) against AdamW with various weight decay settings ($\{0.01, 0.001, 0.0001\}$) on the ELIC model. Our results showed that adding weight decay (via AdamW) yielded results that were either similar to or slightly more unstable/worse than training without weight decay.
>
> Given that weight decay provided no performance benefit in this context, the optimal configuration for the baseline is with weight decay set to $0$. In this setting, Adam and AdamW are identical. Therefore, our comparison with Adam effectively covers the AdamW baseline as well.
>
>
>
> ------
>
>
>
> **Q8.** Application to videos
>
> **Response:** We thank the reviewer for raising this interesting point. We agree that learned video compression involves significantly more complex optimization dynamics compared to image compression, particularly due to temporal error accumulation and the need for effective bitrate allocation across frames.
>
> We believe that the general analysis of Rate-Distortion optimization and the benefits of SOAP applies to the video format as well. Fundamentally, video training is still a joint optimization problem where competing gradients (between rate and distortion, and potentially across temporal steps) can hinder convergence.
>
> As presented in **Appendix A.4: A PRELIMINARY EXPLORATION FOR LEARNED VIDEO COMPRESSION**, we have conducted initial experiments using the DCVC model. Our results (Figure 7) demonstrate that SOAP achieves faster convergence and better final R-D performance compared to Adam in this setting. This suggests that the advantages of curvature-aware optimization extend to the video domain.
>
> However, we emphasize that this is a **preliminary exploration**. We acknowledge that fully addressing the specific challenges of the temporal dimension, such as optimizing frame-level bit allocation and managing long-term error propagation requires more extensive study. We consider the in-depth exploration of second-order optimization for temporal dynamics to be a valuable direction for future work.

---

> ### Author Response · Authors · 2025-11-26
>
> Dear Reviewer 8Sdb,
> We are writing to ensure you have seen the new experimental results added to address your questions about convergence and robustness.
>
> *Convergence verification (Appendix A.16).* We extended Adam training to 1000 epochs (for ELIC on Kodak). The results show that Adam improves by only (-0.02%) BD-Rate with 700 extra epochs, while SOAP maintains a large (~ 3.5%) advantage, indicating that the gap is not due to insufficient baseline training.
>
> *Outlier analysis (Appendix A.18 and Fig. 14).* We now track latent kurtosis and MaxMed throughout training, demonstrating that SOAP actively suppresses outlier formation from the early epochs and stabilizes at much lower values than Adam.
>
> *Robustness checks.* We added experiments with batch size 32 (showing SOAP remains effective on consumer-grade GPUs) and AdamW comparisons is provided in Appendix A.5.
>
> We believe these ablations strengthen the case for SOAP’s practicality and robustness. If they address your doubts, we would be very grateful if you could take them into account in your final assessment.
>
> Best regards,
> The Authors

---

> > ### Comment · Reviewer_8Sdb · 2025-11-27
> > **Reviewer Response**
> >
> > Thank you for your detailed response and experiments. The explanation of training settings and testing conditions, as well as ablation experiments, further verified the effectiveness of SOAP for LIC training, solving most of my problems. Therefore, I will improve my score and look forward to the author's further expansion work in LVC.
> >
> > But there is still a small issue. I tried using the SOAP optimizer for other novel LIC architectures training and found some marginal situations. In some special LIC networks, using the default setting of SOAP with precondition frequency=10 directly can cause rapid training crashes (nan/inf values, which are not the case with Adam). We need to reduce precondition frequency, as frequent updates seem to degrade final performance. Explaining this boundary problem will further help enhance practicality.

---

> > > ### Author Response · Authors · 2025-11-27
> > >
> > > Thank you very much for your follow-up and for taking the time to try SOAP on other LICs. We really appreciate your willingness to update your score and your encouragement regarding future extensions to LVC.
> > >
> > > We also appreciate your note about marginal cases where using SOAP with the default preconditioner update interval (precondition_frequency = 10) can lead to numerical issues (NaN/Inf values), and that reducing the effective update frequency improves stability and final performance. In our experiments on ELIC, TCM, LALIC, and DCAE, update frequencies in {5, 10, 20, 50} were numerically stable and led to very similar convergence behavior. However, as you observed, novel architectures with particularly noisy or sharply varying gradients can push the optimizer into more fragile regimes.
> > >
> > > In the revised manuscript, we have added a short discussion of these potential boundary cases in Appendix A.6 (“What is the Impact of Preconditioner Update Frequency?”). In particular, we write that increasing the update interval (e.g., from 10 to 50, 100, or even 1000) allows each preconditioner refresh to aggregate curvature information over more optimization steps. This smooths the Kronecker-factored curvature estimates and makes their inverse less sensitive to transient gradient spikes, which in turn acts as a more conservative and stable preconditioner, exactly matching your empirical observation that less frequent updates can prevent crashes.
> > >
> > > Our SOAP implementation also includes several small numerical-stability tweaks compared to the original authors’ released code (e.g., careful dtype casting around eigendecomposition/QR, how moments are rotated, and defensive updates of the Kronecker factors). We will release our full training code upon acceptance so that others can directly reuse these stability improvements, and we plan to continue investigating such edge cases as we extend SOAP to broader LVCs.
> > >
> > > Thank you again for this very constructive and practically valuable feedback.

---

### Official Review · Reviewer_RnuL · 2025-10-31

**Soundness:** 3
**Presentation:** 4
**Contribution:** 3
**Rating:** 6
**Confidence:** 4

**Summary:**

This paper proposed SOAP, a second-order optimizer for training Learned Image Compressor (LIC). Applying this optimizer is simple, and only requires a few lines of change. Empirically, the paper demonstrate that SOAP is superior compared to Adam, in terms of step-efficiency, time-efficiency, and achieves a better BD-rate after convergence. The paper provide theoretical and intuitive explanation that why SOAP achieves a better performance. The first explanation is that SOAP "smooth" the training process and better aligns conflicting gradients given by the rate and distortion factor. Both theoretical explanation and empirical evidence that supports the theory are provided. The second explanation is that SOAP suppresses latent and activation outliers, which are negative for entropy coding. Also both theory and empirical evidence are provided.

**Strengths:**

1. This paper is very well structured and easy to follow. Placing the empirical results ahead of theoretical explanation makes it easy to focus on the main results and contributions without being absorbed into too much details, while providing the theoretical explanation afterwards answers the questions naturally raised when reading the empirical results.
2. This paper provides very good insights into why second-order optimizer works better than first-order methods. It manages to define quantitative scores to proof the point of theory, and provide additional empirical evidence to support the theory.
3. The empirical results (table 1) is rich and sound. Benchmarking is performed using multiple baseline models and datasets.

**Weaknesses:**

1. I think showing algorithm 1 is kind of trivial.
2. I am not sure "step-efficiency" and "time-efficiency" are flawless metrics. It is clear that SOAP introduce computation overhead as it needs to compute (even just an approximation) of Hessian. So that each step of SOAP is significantly more expensive than a step of Adam. Therefore I am not sure it is fair to use step-efficiency to compare. Hopefully the authors can address this concern in their response. Using "time-efficiency" is also a bit questionable in some sense, as given the same amount of time SOAP and Adam consumes, it makes lead to e.g. different CPU usage. Hopefully there are better metrics to more fairly quantitatively compare SOAP and Adam.

**Questions:**

1. Question on the validness of using "step-efficiency" and "time-efficiency" to compare Adam and SOAP is already mentioned in the "Weakness" session.
2. It is not fully convincing to me that why having a larger intra-step similarity score is a good thing. It is intrinsically conflicting to optimize rate and distortion, so that a small cosine similarity between p_R and p_D is expected. As both terms are used, it makes sure the model parameters are updated toward the right direction. Why having a larger cosine similarity translates to a good thing in parameter updating?

---

> ### Author Response · Authors · 2025-11-21
> **Response to Reviewer RnuL (1)**
>
> **W1.** showing algorithm 1 is kind of trivial.
>
> **Response:** We appreciate the reviewer's feedback. Our original intention with Algorithm 1 was to visually emphasize the "drop-in" nature of SOAP, contrasting it with prior acceleration methods that require complex architectural changes (e.g., AuxT) or loss reformulation (e.g., Balanced R-D). However, we agree that for the expert audience at ICLR, explicitly showing the code for a standard optimizer swap is unnecessary.
>
> We have **removed Algorithm 1** from the revised manuscript. We now simply state in the introduction and methodology that adopting SOAP requires only a standard optimizer substitution.
>
>
>
> ------
>
> **W2 & Q1.** "step-efficiency" and "time-efficiency"
>
> **Response:**
>  We thank the reviewer for this thoughtful comment. We agree that *step-efficiency* alone does not fully capture the cost of training, as second-order methods like SOAP indeed incur a computational overhead per step (calculating and applying the preconditioner) compared to first-order methods like Adam.
>
> However, we respectfully maintain that the combination of *step-efficiency* and *time-efficiency* provides a comprehensive and fair comparison for the following reasons:
>
> **1. Step-efficiency measures optimization quality.**
>  While not a measure of wall-clock speed, step-efficiency is a standard metric in optimization literature [1, 2] to quantify the *per-update effectiveness*. It isolates the algorithm's ability to navigate the loss landscape.
>
> **2. Time-efficiency is the practical “great equalizer”.**
>  We use *time-efficiency* (Wall-Clock Time) to address the concern regarding per-step overhead. Time-efficiency accounts for the extra computation required by SOAP. The fact that SOAP achieves a **57.7% reduction in wall-clock time** (Table 1) demonstrates that the drastic reduction in the *number* of steps (approx. 70%) far outweighs the increased *cost* per step. This “Time-to-Accuracy” metric is the standard benchmark in both optimization [1, 2, 3] and efficient training literature [4, 5].
>
> **3. Resource Usage (CPU/GPU).**
>  Regarding the concern about CPU usage: Deep Learning training is heavily GPU-bound. The preconditioner updates in SOAP are computed on the GPU, not the CPU. Therefore, CPU usage is not a bottleneck and does not vary significantly between optimizers. Furthermore, regarding GPU memory, as noted in the paper:
>  *“Please note that the additional VRAM overhead of SOAP relative to Adam is negligible (about a 1% increase in our setting) and is therefore not reported separately.”*
>
> In summary, because SOAP significantly reduces *both* the total steps and the total wall-clock time without requiring meaningful extra memory, we believe these metrics fairly represent its superiority over Adam.
>
> **References:**
>
> [1] Anil, R., Gupta, V., Koren, T., Regan, K., & Singer, Y. (2020). Scalable second order optimization for deep learning. arXiv preprint arXiv:2002.09018. (Scalable Shampoo)
>
> [2] Martens, J., & Grosse, R. (2015, June). Optimizing neural networks with kronecker-factored approximate curvature. In International conference on machine learning (pp. 2408–2417). PMLR. (K-FAC)
>
> [3] Kingma, D. P., & Ba, J. (2014). Adam: A method for stochastic optimization. arXiv preprint arXiv:1412.6980. (Adam)
>
> [4] Li, H., Li, S., Dai, W., Cao, M., Kan, N., Li, C., ... & Xiong, H. (2025). On disentangled training for nonlinear transform in learned image compression. arXiv preprint arXiv:2501.13751. (AuxT)
>
> [5] Zhang, Y., Duan, Z., Huang, Y., & Zhu, F. (2025). Accelerating Learned Image Compression Through Modeling Neural Training Dynamics. Transactions on Machine Learning Research, 2025, 4249. (CMD-LIC)

---

> ### Author Response · Authors · 2025-11-21
> **Response to Reviewer RnuL (2)**
>
> **Q2.** Why higher score?
>
> Response:
>
> We thank the reviewer for raising this fundamental point. We agree that the raw gradients of Rate and Distortion are intrinsically conflicting and naturally exhibit negative cosine similarity. However, the critical distinction lies between the raw gradients and the preconditioned update vectors. The goal of a second-order optimizer like SOAP is to rotate the optimization basis (via the preconditioner matrix) such that these conflicting gradients are transformed into update vectors that point in a shared descent direction.
>
> We have expanded the manuscript to clarify this mechanism in two ways. First, in the new **Section 4.2 (Geometric Intuition)**, we explain that low cosine similarity results in "destructive interference" or a geometric "tug-of-war," where the rate and distortion updates cancel each other out. This wastes gradient magnitude and leads to small effective steps. Conversely, high cosine similarity creates "constructive interference," where the magnitudes of the updates sum up, resulting in a larger and more efficient step.
>
> Second, to provide a rigorous justification, we added **Appendix A.13** and **Proposition 5** ("Alignment Maximizes Descent Efficiency"). We formally prove that the lower bound of the loss reduction $\Delta_t$ at each step is monotonically increasing with respect to the intra-step cosine similarity $\mathcal{S}_{\text{intra}}^t$. Specifically, we derive the bound:
>
> $$\Delta_t \ge C(\eta) \left( \|p_{R,t}\|^2 + \|p_{D,t}\|^2 + 2 \|p_{R,t}\| \|p_{D,t}\| \cdot \mathcal{S}_{\text{intra}}^t \right)$$
>
> where $C(\eta) > 0$ depends on the smoothness and preconditioner spectrum. This inequality demonstrates that maximizing the alignment term $\mathcal{S}_{\text{intra}}^t$ strictly increases the guaranteed reduction in loss, thereby theoretically validating why higher alignment translates to faster and more efficient parameter updating.

---

> ### Author Response · Authors · 2025-11-26
>
> Dear Reviewer RnuL,
> As the discussion phase concludes next week, we wanted to check whether our revisions have addressed your concerns. Based on your helpful suggestions, we made the following changes.
>
> First, we removed Algorithm 1 to keep the presentation focused on the core contributions rather than on the mechanics of swapping the optimizer.
>
> Second, we clarified our efficiency metrics: in Section 3 we emphasize that “Time efficiency” explicitly accounts for the computational overhead of the preconditioner, and SOAP still yields about a (57%) reduction in wall-clock time on average.
>
> Third, to justify the role of intra-step similarity, we added Appendix A.13 and Proposition 5, which show that increasing the alignment between the rate and distortion updates strictly improves a lower bound on the per-step loss reduction (constructive interference).
>
> We hope these updates improve both the clarity and theoretical soundness of the paper, and we would very much appreciate any final feedback you might have.
>
> Best regards,
> The Authors

---

### Official Review · Reviewer_Nu8g · 2025-11-01

**Soundness:** 3
**Presentation:** 3
**Contribution:** 2
**Rating:** 2
**Confidence:** 5

**Summary:**

The paper introduces a second-order optimization method for training learned image compression models. Experimental results demonstrate faster convergence and improved coding efficiency compared with first-order optimization. Furthermore, suppressing activation outliers is shown to facilitate effective post-training quantization of learned image compression networks.

**Strengths:**

1. Overall, the paper makes a meaningful contribution to the learned image coding community.

2. The experimental results appear solid and well supported.

3. The paper is clearly written and easy to follow.

**Weaknesses:**

The paper claims three primary contributions, however, each has certain weaknesses, as discussed below.

1. Although this may be a common concern, I am afraid the paper lacks strong novelty. The main contribution appears to be the application of an existing second-order optimization method (SOAP) to learned image compression (LIC). While the plug-and-play nature of the method demonstrates its versatility, it also highlights the limited originality of the work, especially since SOAP itself was not proposed by the authors. Besides, the proposed approach does not show a significant performance improvement compared with previous studies such as Zhang et al. (2025c).

2. The theoretical part is relatively weak. The core section (Section 4.2) mainly reproduces the SOAP derivation, which cannot be considered a novel contribution of this paper.

3. Regarding the PTQ part, it seems that the outlier reduction is performed only for activations (as shown in Fig. 4) rather than for weights. If this is the case, considering that AdaRound is designed solely for weight quantization, I wonder why the authors employed AdaRound in the experiments of Section 5.3. I am also curious why weight quantization is included in Table 2.

**Questions:**

1. In Section 5, the distinction between latent and activation is unclear. Does latent refer to the output activation of the final encoding layer?

2. From Fig. 4, it is difficult to observe a clear difference in the dynamic range of values.

3. In Table 3, it is unclear what the final results would be if the model were trained for more epochs (e.g., 1000 epochs).

**Details Of Ethics Concerns:**

No specific ethics concerns

---

> ### Author Response · Authors · 2025-11-21
> **Response to Reviewer Nu8g (1)**
>
> **W1. Contribution and performance improvement compared with Zhang et al. (2025c)**
>
> **Response:**
>  We thank the reviewer for raising the comparison with Zhang et al. (2025c) (Balanced R–D). While both works aim to improve LIC optimization, they address the problem through fundamentally different mathematical mechanisms, leading to distinct insights and performance characteristics.
>
> **1. Fundamental mechanism: curvature geometry vs. scalar reweighting.**
>  The core novelty of our work lies in diagnosing LIC optimization difficulties as a problem of *curvature*, not merely imbalanced gradient magnitudes.
>
> - **Zhang et al. (2025c)** take a Multi-Objective Optimization (MOO) perspective. They compute adaptive scalar weights ($w_R, w_D$) to balance the raw gradients $g_R$ and $g_D$. However, the final update is still a first-order step, limited by the diagonal preconditioning of common optimizers.
> - **Our approach** applies a layerwise quasi-Newton preconditioner ($H^{-1}$). Instead of only reweighting the losses, SOAP uses curvature information to *rotate and scale* the update direction in high-dimensional parameter space. Our theoretical analysis (Sec. 4) shows that this intrinsically resolves gradient conflicts in the preconditioned space, without requiring explicit auxiliary solvers as in Zhang et al.
>
> **2. Superior performance and efficiency.**
>  The reviewer expressed concern that our approach may not show significant gains over Zhang et al. Our empirical results demonstrate otherwise. As shown in Table 3 (Appendix A.3), we evaluate both methods on TCM-S under identical training protocols:
>
> - **Better R–D performance:** SOAP achieves an average BD-Rate reduction of **2.76%**, while Balanced R–D achieves only **1.71%**. This is a substantial improvement for image compression.
> - **Faster convergence:**
>   - SOAP reaches target performance using **0.28×** the training steps of Adam (vs. 0.67× for Balanced R–D).
>   - SOAP requires **0.39×** the wall-clock time (vs. 0.81× for Balanced R–D).
>
> Thus, SOAP provides **roughly 2× the speedup** to reach the same performance and **≈1% better coding efficiency** after full convergence.
>
> **3. Novel insights on outliers and deployment.**
>  Beyond speed, our work introduces a new contribution absent in Zhang et al. and other LIC papers: second-order optimization fundamentally alters representation statistics. In Sec. 5, we show that SOAP decreases feature kurtosis and suppresses activation outliers. This yields a major practical benefit: **improved robustness to Post-Training Quantization (PTQ)**, a key bottleneck in real-world LIC deployment. This positions SOAP not only as a training accelerator but also as a tool for producing more deployable models.
>
> **4. Usability and drop-in integration.**
>  Balanced R–D requires modifying the training loop, introducing Pareto-stationary solvers, and tuning multiple hyperparameters. In contrast, SOAP delivers second-order benefits as a simple **drop-in optimizer replacement**, requiring no changes to the architecture or loss function.
>
> In summary, while Zhang et al. propose a useful gradient-balancing scheme, our work provides a distinct, more effective, and easier-to-use solution based on curvature information, with additional deployment advantages not explored in prior optimization studies.

---

> > ### Comment · Reviewer_Nu8g · 2025-11-22
> > **Speedup is important?**
> >
> > The author mentions the training speedup, but I am wondering whether such a speedup is actually important for learned image coding.

---

> > > ### Author Response · Authors · 2025-11-23
> > >
> > > **Response.** We thank the reviewer for this question. We believe that training efficiency is a critical practical aspect for current LIC models, and our speedup is important for three reasons.
> > >
> > > 1. **Massive reduction in research and design cost.**
> > >
> > > Training state-of-the-art LIC models is extremely computationally expensive. As shown in Figure 2(c), training a single LALIC model to convergence requires approximately 211 H100 GPU hours. To evaluate the performance of a compression method typically requires training separate models for at least 6 bit-rates (different $\lambda$ values) to obtain a full R-D curve, leading to roughly $211 \times 6 \approx 1,266$ GPU hours. In practice, codec design involves multiple architectural ablations and hyperparameter searches, which further increase the training cost and make experimentation a major bottleneck.
> > >
> > > With SOAP, we reduce the number of training steps and wall-clock time by about 50-70% compared to Adam across four strong LIC architectures (Table 1). For the LALIC example above, this effectively reduces the cost of one R-D curve from about 1,266 to roughly 620 GPU hours, substantially lowering the barrier for designing and iterating on advanced codecs. We also note that related LIC works published in the same or similar venues, such as AuxT[1] and CMD-LIC[2], explicitly target training acceleration, further demonstrating that training cost is well recognized as a key challenge in the community.
> > >
> > > Also note that SOAP only changes the *training* optimizer. The architecture, bitstream, and decoding process remain unchanged, so there is no additional inference-time complexity, latency, or memory overhead.
> > >
> > > 2. **Improved R-D performance at fixed compute budget.**
> > >
> > > As shown in Table 1, when trained for the same number of steps or for the same wall-clock time, SOAP-trained models consistently converge to lower R-D loss and achieve better BD-Rate than Adam-trained models. Across Kodak, Tecnick, and CLIC2022 datasets, we show that SOAP provides roughly a 3% BD-Rate improvement on average compared to Adam at convergence. Thus, even under a fixed training budget, replacing Adam with SOAP yields strictly better compression performance.
> > >
> > > 3. **Beyond speed: improved deployability via outlier suppression.**
> > >
> > > Our contribution is not limited to acceleration. As discussed in Section 5, we observed that SOAP's curvature-aware updates naturally suppress outliers in both latent and activation spaces, leading to more regular feature statistics. This, in turn, makes models trained with SOAP substantially more robust to post-training quantization (PTQ) than those trained with first-order optimizers, as reflected by the consistent BD-Rate gains under W8A8 PTQ. Consequently, SOAP makes LIC models not only faster to train and stronger in terms of R-D, but also easier to deploy on resource-constrained hardware.
> > >
> > > Reference:
> > >
> > > [1] Li, H., Li, S., Dai, W., Cao, M., Kan, N., Li, C., ... & Xiong, H. On Disentangled Training for Nonlinear Transform in Learned Image Compression. International Conference on Learning Representations, 2025.
> > >
> > > [2] Zhang, Y., Duan, Z., Huang, Y., & Zhu, F. (2025). Accelerating Learned Image Compression Through Modeling Neural Training Dynamics. Transactions on Machine Learning Research, 2025, 4249.

---

> ### Author Response · Authors · 2025-11-21
> **Response to Reviewer Nu8g (2)**
>
> **W2. The core section (Section 4.2) mainly reproduces the SOAP derivation**
>
> **Response:**
>  We thank the reviewer for raising this point. We would like to clarify that Section 4.2 (now Section 4.3 in the revised manuscript) does **not** re-derive the SOAP optimizer nor reproduce the core derivation from the SOAP paper. For convenience, the SOAP paper is:
>  Vyas et al., *“SOAP: Improving and Stabilizing Shampoo using Adam for Language Modeling”*, ICLR 2025 (available at https://openreview.net/forum?id=IDxZhXrpNf).
>
> Instead, our Section 4.2 begins from the high-level interpretation of SOAP as a **local quasi-Newton method** and then develops **new theory that is specific to rate–distortion (R–D) optimization**, which does not appear in the SOAP paper.
>
> The only part of Section 4.2 that relies on prior work is the observation that SOAP can be viewed as performing a local Newton-like update,
> $$
>  p \approx - H^{-1} g,
> $$
>  an assumption formalized as Theorem 1 in Appendix A.7. This assumption simply states that SOAP provides a well-conditioned Newton-style preconditioner.
>
> **All remaining theoretical components of Section 4.2 are novel**, including:
>
> - **Lemma 1 (inter-step alignment):**
>    Proves that Newton-preconditioned updates for the R–D loss enjoy strong *inter-step* alignment, meaning $p_t$ and $p_{t+1}$ have high cosine similarity.
>    This explains the empirically observed stability of SOAP in Fig. 3(b).
> - **Proposition 1 (intra-step alignment of rate vs. distortion):**
>    Shows that under reasonable assumptions on the component Hessians $H_R$, $H_D$, and the combined Hessian $H$, the Newton-preconditioned updates $p_R$ and $p_D$ become asymptotically colinear near a local minimum.
>    This is the key theoretical justification for why curvature-aware updates resolve gradient conflicts between rate and distortion—an effect Adam cannot capture.
> - **Cosine-similarity metrics $S^{\text{inter}}_t$ and $S^{\text{intra}}_t$:**
>    These metrics and their predicted behaviors do *not* appear in the SOAP paper; they are introduced by us to analyze R–D optimization dynamics.
>
> Importantly, the SOAP paper focuses on:
>
> - algorithmic equivalence between Shampoo and Adafactor in the preconditioner eigenspace,
> - interpreting SOAP as “Adam in the preconditioner’s eigenbasis,”
> - efficiency improvements for large language model pre-training,
>
> and **does not** analyze:
>
> 1. rate–distortion objectives,
> 2. gradient conflict between $L_R$ and $L_D$,
> 3. Newton-preconditioned *component* updates ($p_R$, $p_D$),
> 4. inter- or intra-step cosine alignment properties of such updates.
>
> Our Section 4.2 fills this gap by using the Newton interpretation of SOAP to derive **new theoretical predictions** about how second-order preconditioning resolves gradient conflicts in R–D optimization. We then validate these predictions empirically in Fig. 3.
>
> We hope this clarifies that our Section 4.2 does not duplicate the derivation in the SOAP paper; instead, it builds new theory that connects Newton-style preconditioning (instantiated via SOAP) to stable and cooperative optimization dynamics in learned compression.

---

> ### Author Response · Authors · 2025-11-21
> **Response to Reviewer Nu8g (3)**
>
> **W3. Clarification of AdaRound and PTQ**
>
> **Response:**
>  We thank the reviewer for the careful observation. We apologize for any confusion regarding the quantization settings and the visualization in Figure 4. We have revised the manuscript accordingly and clarify the key points below.
>
> **1. Clarification on Figure 4.**
>  Figure 4 visualizes the **latent representations** (specifically the scaled deviation maps for ELIC latents), not intermediate backbone activations. The purpose of this figure is to demonstrate that SOAP suppresses latent-space outliers (lower maximum values), which is important because reducing outliers leads to tighter entropy modeling after quantization.
>
> **2. Clarification on the W8A8 protocol.**
>  We agree that AdaRound is designed for weight quantization. Our W8A8 experiments quantize **both** weights and activations to simulate realistic hardware deployment.
>
> - **Weights:** AdaRound is used to optimize rounding of weights ($\mathbf{W}$).
> - **Activations:** A channel-wise dynamic quantization scheme is applied. During each forward pass, per-channel min/max values are used to compute quantization parameters. This component is non-learnable and applied on-the-fly.
>
> **3. Justification for using AdaRound.**
>  We selected AdaRound as it is a widely used and well-established PTQ baseline. While more advanced PTQ methods (e.g., [1]) may yield stronger absolute performance, our goal is not to benchmark PTQ algorithms. Rather, our purpose is to demonstrate the **robustness of SOAP-trained models** to activation outliers under a standard PTQ pipeline.
>
> **4. Why SOAP improves W8A8 performance with AdaRound.**
>  Activation outliers (as quantified in Table 2) induce high quantization noise. During AdaRound, the reconstructed model tries to solve:
> $$
>  |\mathbf{W}x - \hat{\mathbf{W}}\hat{x}|^2,
> $$
>  but if the quantized activations $\hat{x}$ suffer significant error due to outliers, even the best $\hat{\mathbf{W}}$ cannot adequately recover accuracy. Since SOAP-trained models exhibit far fewer activation outliers (e.g., Kurtosis reduced from $64.96 \to 4.28$ for ELIC), activation quantization noise is dramatically lowered. Consequently, AdaRound’s weight optimization becomes more effective, yielding better overall W8A8 performance.
>
> **5. Removal of W8A32 results.**
>  As noted by the reviewer, W8A32 (weight-only quantization) keeps activations in floating point and therefore is insensitive to activation outliers. To maintain clarity and focus on deployability benefits under full quantization, we have **removed W8A32 results** from Table 2 and now report only W8A8.
>
> We have clarified all quantization protocol details in the revised Section 5.3 (Empirical Validation).
>
> **Reference**
>  [1] Shi, J., Lu, M., & Ma, Z. (2023). *Rate-distortion optimized post-training quantization for learned image compression.* IEEE Transactions on Circuits and Systems for Video Technology, 34(5), 3082–3095.
>
>
>
> ------
>
> **Q1. Clarification of latent**
>
> **Response:**
>  We thank the reviewer for requesting this clarification. Yes, the reviewer’s understanding is correct.
>
> - **Latent:** This refers to the bottleneck representation (typically denoted as $y$ or $z$) produced by the final layer of the encoder in learned compression. This tensor is the one that undergoes quantization and entropy coding to form the bitstream.
>
> In our experiments (Table 2), all “Latent’’ metrics are computed on this final bottleneck output. We have clarified this definition in the revised manuscript.
>
> ------
>
> **Q2. Dynamic range of Fig. 4**
>
> **Response:**
>  We appreciate the reviewer’s comment. We clarify that the key indicator of improved feature quality in Figure 4 is **not** the overall dynamic range of the visualization, but the **reduction in the maximum value** (peak deviation).
>
> The figure visualizes the *scaled deviation map*, representing the normalized error between the quantized latent $\hat{y}$ and the original latent $y$. In this context:
>
> - **Ideal state:** A value of zero indicates perfect reconstruction; lower values correspond to less information loss.
> - **Outliers vs. max value:** Large outliers in the latent space cause large quantization errors, which appear as “hot spots’’ (high values) in the deviation map. A key effect of SOAP is suppressing these outliers.
>
> As shown in the color bars and labels in Figure 4, SOAP consistently reduces the **maximum scaled deviation** relative to Adam (e.g., $0.216 \to 0.188$ for *kodim01* and $0.108 \to 0.084$ for *CLIC17*). This reduction demonstrates that SOAP lowers worst-case latent errors, even if the overall dynamic range of the plot looks visually similar at first glance.

---

> > ### Comment · Reviewer_Nu8g · 2025-11-22
> > **Any theoretical reason?**
> >
> > Is there any theoretical justification for why SOAP is able to reduce the dynamic range of the activations?

---

> > > ### Author Response · Authors · 2025-11-23
> > >
> > > **Response.** A detailed theoretical analysis is provided in Section 5.2 and Appendix A.14 of the paper. Our analysis is two-fold:
> > >
> > > 1. **Conservation of correlation energy (Eq. (10).)**
> > >
> > > Using tools from Signal Propagation theory, we derive an exact identity that decomposes the total “correlation energy” of features into a diagonal term and an off-diagonal term. This identity shows that the total energy is conserved under rotations. Diagonal optimizers such as Adam can only rescale coordinates and are inefficient at moving energy into the off-diagonal terms, so excess energy tends to accumulate in the diagonal term, manifesting as high kurtosis and large-magnitude outliers. In contrast, SOAP’s Newton-like preconditioning performs rotations in a curvature-aligned basis and then maps back, which enables it to redistribute correlation energy into off-diagonal terms and thereby suppress the build-up of outliers on the diagonal. This directly translates into a reduced outlier for activations and latents.
> > >
> > > 2. **Small-step kurtosis bound (Eq. (11).)**
> > >
> > > We further derive a small-step expansion showing that the expected growth of feature kurtosis during a SOAP update is upper-bounded by that of a diagonal optimizer. In other words, in non-diagonal curvature regimes, SOAP provably does not increase kurtosis faster than a diagonal method like Adam, and in practice, strictly limits outlier formation relative to Adam.
> > >
> > > To empirically validate this theory, we track the evolution of feature kurtosis and MaxMed statistics throughout training in **Appendix A.18** and **Figure 14**. These results show that, while Adam allows these metrics (and thus the activation dynamic range) to spike to large values in the early stages and remain high, SOAP actively suppresses their growth and stabilizes at a significantly lower level.

---

> ### Author Response · Authors · 2025-11-21
> **Response to Reviewer Nu8g (4)**
>
> **Q3. Longer training period**
>
> **Response:**
>  We thank the reviewer for this important question. To verify that SOAP’s superior performance is not simply due to insufficient training of the Adam baseline, we added a new section in the revised manuscript: **Appendix A.16:  Will a Longer Training Period Make Any Difference?**.
>
> In this appendix, we performed ablations on the ELIC model with **extended training durations** (up to 1000 epochs) and **alternative learning-rate schedulers**. The key findings are summarized below.
>
> **Extended epochs (1000 epochs).**
>  Increasing training from 300 to 1000 epochs yields only a **−0.02% BD-Rate** improvement. This is because the ReduceOnPlateau scheduler already decays the learning rate to extremely small values ($< 5 \times 10^{-6}$). By ~300 epochs, the model has effectively reached a stationary point, so further training brings negligible benefit.
>
> **Alternative scheduler (Half Constant + Cosine).**
>  To check whether the baseline scheduler was the bottleneck, we tested a “Half Constant + Cosine’’ schedule for 300 and 500 epochs, which maintains a high learning rate for much longer. Even here, improvements were marginal (best result: **−0.10% BD-Rate**).
>
> **In contrast**, SOAP achieves a **−3.49% BD-Rate** improvement within just 300 epochs.
>  This confirms that the advantage of SOAP is *not* due to training longer but due to how curvature-aware preconditioning resolves intra- and inter-step gradient conflicts in the R–D landscape.
>
> Below is the comparison table included in the revised manuscript:
>
> ------
>
> ### ELIC performance under different training durations and schedulers (Kodak dataset)
>
> | Optimizer       | Epochs   | Scheduler              | BD-Rate vs. Baseline |
> | --------------- | -------- | ---------------------- | -------------------- |
> | Adam (Baseline) | 300      | ReduceOnPlateau        | 0.00%                |
> | Adam            | 1000     | ReduceOnPlateau        | −0.02%               |
> | Adam            | 300      | Half Constant + Cosine | −0.05%               |
> | Adam            | 500      | Half Constant + Cosine | −0.10%               |
> | **SOAP**        | **300**  | ReduceOnPlateau        | **−3.49%**           |
> | **SOAP**        | **1000** | ReduceOnPlateau        | **−3.51%**           |
>
> ------
>
> These results show that simply training Adam longer does not close the gap, whereas SOAP maintains a large benefit even under extended training.

---

> ### Author Response · Authors · 2025-11-26
>
> Dear Reviewer Nu8g,
> Thank you again for your thoughtful and detailed comments. We wanted to briefly follow up on our responses to your questions about the practical importance of training speedup and the theoretical basis for outlier reduction.
>
> *Importance of speedup.* Section 3 and Figure 2 quantify that replacing Adam with SOAP reduces the training cost for one full R–D curve by roughly a factor of two (e.g., for LALIC, from about (1200) to (~600) GPU hours), which substantially lowers the barrier to architectural iteration and codec design.
>
> *Theoretical justification for outlier reduction.* Section 5.2 and Appendix A.14 develop a “conservation of correlation energy” view, showing how diagonal optimizers tend to accumulate energy in outliers, while SOAP’s curvature-aligned rotations redistribute this energy and mathematically bound kurtosis growth.
>
> We also clarified the W8A8 quantization protocol in Section 5.3 and added the 1000-epoch ablation in Appendix A.16, confirming that SOAP’s gains are not due to under-training the Adam baseline.
>
> We hope these clarifications address your concerns about contribution and novelty, and we would be grateful for any further thoughts you may have.
> Best regards,
> The Authors

---

### Official Review · Reviewer_xcTm · 2025-11-06

**Soundness:** 3
**Presentation:** 3
**Contribution:** 3
**Rating:** 6
**Confidence:** 4

**Summary:**

This paper investigates second-order optimization for training Learned Image Compression (LIC) models. The authors argue that common first-order optimizers (SGD, Adam) suffer from gradient conflicts between rate and distortion terms in the rate–distortion (R–D) loss, causing slow and unstable convergence. They propose replacing Adam with a quasi-Newton optimizer (SOAP). With this simple two-line substitution the authors are able to show the acceleration in training by 57–70% in wall-time and steps and improvement in BD-Rate by ~3% across different learned image compression models. Furthermore, they showed empirically that the SOAP optimizer reduces the outliers in the activations and latents, improving entropy modeling and post-training quantization (PTQ) robustness. Finally, they also present theoretical analysis to show why the SOAP optimizer is able to overcome the conflicting gradient updates.

**Strengths:**

1. Simple yet useful approach in the Learned image compression. Replacing Adam with SOAP is a minimal, drop-in change requiring no architectural modification. Despite its simplicity, it yields significant practical benefits — faster training, better BD-Rate.

2. The paper evaluates on four advanced LICs (ELIC, TCM, LALIC, DCAE) across three datasets (Kodak, Tecnick, CLIC2022). The improvements are consistent and robust.

3.  The authors provide an elegant geometric interpretation of how Newton preconditioning aligns gradients from rate and distortion objectives (intra-step) and across consecutive iterations (inter-step). I did not check the accurateness of the theorems.

4. Section 5 provides an interesting secondary contribution: showing that second-order updates reduce activation outliers, thus tightening entropy models and enhancing quantization robustness. This bridges optimization dynamics with hardware-friendly deployment — a fresh perspective.

**Weaknesses:**

1. The main idea—using a second-order optimizer—is not new in deep learning. SOAP (Vyas et al., 2024) is external work, and applying quasi-Newton updates to LICs is a contextual extension, not a fundamental algorithmic advance. The novelty lies more in the application and analysis than in the optimizer itself. For a top-tier acceptance, stronger conceptual framing (e.g., a LIC-specific curvature model or adaptive R–D Hessian decomposition) would strengthen the contribution.

2. The comparison is primarily between Adam and SOAP. The authors mention Shampoo and AdaHessian but do not present direct comparisons, even though those are obvious baselines.
Empirical results would be more convincing if SOAP’s advantages were shown against other second-order or preconditioned methods (e.g., Shampoo, Muon, AdaHessian).

3. Proposition 1 assumes that the rate and distortion Hessians are locally proportional or jointly diagonalizable — a strong assumption that is unlikely to hold in practice for deep networks. The analysis is therefore qualitative rather than rigorous, though still informative.

**Questions:**

1) Can the authors empirically verify the “shared curvature structure” assumption (e.g., via eigenvalue overlap or gradient covariance between rate and distortion losses)?
2)How does SOAP interact with other training stabilizers (e.g., gradient clipping, EMA, learning-rate warmup)?
3) Is SOAP’s benefit additive to other acceleration techniques (CMD-LIC, Balanced-RD)?
4) Are the preconditioners reused across layers, or is SOAP applied per-layer independently?

---

> ### Author Response · Authors · 2025-11-21
> **Response to Reviewer xcTm (1)**
>
> **W1. Strengthen the contribution with a LIC-specific curvature model or adaptive R–D Hessian decomposition.**
>
> **Response.**
>  We appreciate the reviewer for this thoughtful comment. We agree that applying an existing optimizer like SOAP is not an algorithmic innovation in itself. However, our contribution lies in the theoretical and empirical characterization of Rate–Distortion (R–D) optimization dynamics, uncovering why standard first-order methods fail in this specific landscape and how Newton-style preconditioning resolves these pathologies.
>
> **1. LIC-specific curvature analysis: resolving gradient conflicts.**
>  We do not merely apply SOAP; we provide an R–D–specific analysis (Sec. 4) demonstrating that the central challenge in LIC training is the *gradient conflict* between rate and distortion terms. We show that diagonal optimizers (Adam) fail to capture the off-diagonal correlations required to align these competing objectives, leading to oscillatory and adversarial updates. In contrast, we demonstrate that SOAP’s Newton-like preconditioning ($H^{-1} g$) mathematically rotates the rate and distortion gradients into a cooperative descent direction. This is a specific characterization of the R–D landscape, not a generic application of optimization.
>
> **2. Specificity of the solution: not all second-order methods work.**
>  To further strengthen the conceptual contribution, we have expanded Appendix A.15 to empirically compare SOAP against other curvature-aware optimizers such as Shampoo, SPlus, Muon, and Adagrad. Crucially, we find that methods approximating $H^{-1/2}$ (like Shampoo) do *not* yield the same efficiency or alignment benefits as SOAP ($H^{-1}$). This highlights that simply “using second-order information’’ is insufficient; the optimizer must effectively approximate the full inverse Hessian to whiten the R–D landscape and resolve the specific conflicts.
>
> **3. Novelty in deployability: the outlier–curvature connection.**
>  Another novelty of our work is the discovery connecting optimization curvature to signal-processing statistics (Sec. 5).
>  We provide a theoretical view based on conserved *correlation energy*, showing that diagonal methods concentrate variance into activation outliers (hindering quantization), whereas SOAP’s basis rotation diffuses this variance. This provides a mechanistic explanation for why SOAP-trained models are inherently more robust to post-training quantization (improving W8A8 performance by $\sim 2$ % BD-Rate), a critical property for practical video/image codecs that is completely independent of convergence speed.
>
> **4. Future direction: adaptive R–D Hessian decomposition.**
>  We thank the reviewer for the suggestion regarding adaptive R–D Hessian decomposition. While a fully adaptive decomposition is beyond the scope of this work, our findings provide the theoretical foundation for such future developments. We have updated our Conclusion and Future Work section to explicitly identify adaptive R–D Hessian decomposition as a promising research direction that can be explored based on the baseline established in this paper.
>
> In summary, our work establishes that efficient Newton-style optimization is structurally better suited for the multi-objective R–D landscape than first-order methods. We believe this analysis provides the conceptual framing the reviewer seeks, moving beyond a simple benchmark to explain the interaction between Hessian approximation, gradient alignment, and latent statistics in learned compression.

---

> ### Author Response · Authors · 2025-11-21
> **Response to Reviewer xcTm (2)**
>
> **W2. Against other second-order or preconditioned methods.**
>
> **Response:**
>  Thank you for the constructive suggestion. We agree that comparisons beyond Adam and SOAP strengthen the evaluation. We have revised **Appendix A.15, Comparison with Other Optimization Paradigms**, to compare SOAP against a broader spectrum of optimizers: first-order (SGD), diagonal root-inverse (Adagrad), and structured root-inverse (Shampoo/SPlus). We also discuss our attempts to apply Muon.
>
> Our results highlight the following:
>
> **1. Shampoo/SPlus (Structured $H^{-1/2}$).**
>  In our experiments, Shampoo/SPlus converged slightly faster than Adam but remained slower and less effective than SOAP. We also observed the instability often reported in literature, requiring careful gradient crafting to avoid divergence. Theoretically, Appendix A.15 shows that Shampoo approximates the inverse *square root* of the Hessian ($H^{-1/2}$), whereas SOAP approximates the *full inverse* ($H^{-1}$). The full inverse is necessary to fully “whiten’’ the local landscape and maximize alignment between conflicting gradients (Constructive Interference); the square-root inverse only partially corrects curvature.
>
> **2. Adagrad (Diagonal $H^{-1/2}$).**
>  Adagrad provided stable training but was consistently outperformed by Adam and SOAP. This confirms that simple diagonal approximations struggle to resolve the non–axis-aligned gradient conflicts inherent to the Rate–Distortion objective.
>
> **3. Muon.**
>  We attempted to adapt the emerging Muon optimizer for LIC. However, Muon is designed for 2D parameters (linear layer matrices), whereas LIC architectures rely heavily on 4D convolutional kernels. Flattening these kernels creates a dimensionality match but disrupts spatial inductive biases. Despite extensive hyperparameter tuning, we were unable to achieve stable convergence with Muon in this setting. Further investigation is needed to understand LIC-specific Muon behavior.
>
> **4. SGD.**
>  As a baseline, SGD failed to reach competitive rate–distortion performance under the same training steps, underscoring the necessity of curvature-aware or adaptive methods in this optimization landscape.
>
> In summary, SOAP demonstrates superior performance and stability compared to these alternatives by effectively approximating the full Newton step to resolve gradient conflicts, while remaining as easy to use as a drop-in replacement.
>
>
>
>
>
> ------
>
> **W3 & Q1. Empirically verify the “shared curvature structure” assumption.**
>
> **Response:**
>  We appreciate the reviewer for highlighting this critical aspect of our theoretical motivation. While we did not include a direct spectral analysis of the full Hessian (computationally prohibitive for large LIC models due to the $O(n^3)$ cost), our manuscript provides strong empirical verification of the *shared curvature structure* through the analysis of update-vector alignment in Section 4.4 and Figure 3(a).
>
> Specifically, our verification relies on the logical implication established in Proposition 1 (Appendix A.9). The proposition shows that a shared second-order preconditioner (such as SOAP) can align the update vectors of competing objectives ($p_R$ and $p_D$) if their underlying Hessians ($H_R$ and $H_D$) are locally proportional or jointly diagonalizable.
>
> In Figure 3(a), we empirically plot the intra-step cosine similarity $S(p_R, p_D)$ throughout training. The results indicate that with SOAP, this similarity consistently approaches 1.0 (strong positive alignment). In contrast, Adam—ignoring the shared off-diagonal curvature—produces misaligned or adversarial updates (negative cosine similarity).
>
> This high alignment observed with SOAP thus serves as a direct empirical proxy supporting the shared-curvature assumption.

---

> ### Author Response · Authors · 2025-11-21
> **Response to Reviewer xcTm (3)**
>
> **Q1 (2). Training stabilizers (e.g., gradient clipping, EMA, learning-rate warmup)**
>
> **Response:**
>  We thank the reviewer for this insightful question. To evaluate how the optimizer interacts with standard training heuristics, we conducted an ablation study using the ELIC model on the Kodak dataset, examining **gradient clipping**, **learning-rate warmup**, and **exponential moving average (EMA)**.
>
> A detailed discussion and results have been added to **Appendix A.17** of the revised manuscript. The main findings are summarized below:
>
> **Gradient clipping.**
>  Adam is highly sensitive to clipping; removing it or relaxing the threshold leads to instability or suboptimal convergence. In contrast, SOAP exhibits substantially greater structural stability and does not diverge even without clipping (likely due to its second-order intrinsic normalization). However, since both optimizers achieve their best final R–D performance using the standard clipping threshold of 1.0, we retain this setting.
>
> **Learning-rate warmup.**
>  Both optimizers showed negligible sensitivity to warmup. The initial optimization dynamics were sufficiently stable to train at the base learning rate from the start, so no warmup is used.
>
> **Exponential moving average (EMA).**
>  EMA provides universal benefits. For Adam, it is essential for smoothing the oscillatory noise caused by gradient conflicts. For SOAP, although the raw training trajectory is already smooth—supporting our claims about its stability—EMA still improves final generalization. Given its negligible computational overhead (approximately 2%), we enable EMA by default.
>
>
>
>
>
> ------
>
> **Q2. Additive to other acceleration techniques**
>
> **Response:**
>  We thank the reviewer for the excellent suggestion to investigate SOAP’s benefit when combined with other acceleration techniques. In response, we added new experiments combining SOAP with Auxiliary Transform (AuxT) [1]. The results appear in **Appendix A.3 (Comparison with Other Methods)**. The resulting **AuxT + SOAP** variant achieves the best overall performance: it further reduces the Steps-to-Adam and Time-to-Adam ratios from $0.28$ and $0.39$ (SOAP alone) to $0.23$ and $0.35$, and improves the average BD-Rate from $-2.76$% to $-2.91$%. This demonstrates that SOAP provides complementary gains on top of an existing accelerator.
>
> To clarify our choice of AuxT:
>
> - **CMD-LIC** is not publicly released, so we only have access to the authors’ reported results and cannot confidently modify their training code to plug in SOAP while ensuring fairness.
> - **Balanced-RD** is implemented as its own optimizer; combining it with SOAP would require designing a hybrid optimization rule and further hyperparameter tuning—nontrivial and beyond this paper’s scope.
>
> In contrast, **AuxT + SOAP** serves as a clean and controlled example showing that SOAP’s benefits are *additive* to those of other acceleration techniques. We believe exploring such combinations is an exciting direction for future work. In particular, integrating SOAP with other optimizer-level accelerators or architectural-speedup methods may reveal additional synergies. Developing principled hybrid optimization strategies that unify SOAP with approaches like AuxT or CMD-LIC could yield even greater acceleration and stability, which we see as a promising avenue for follow-up work.
>
> For the reviewers’ convenience, we reproduce the comparison table below. Please refer to **Appendix A.3** for the full R–D loss curves.
>
> ------
>
> ### Computational Complexity and BD-Rate Comparison on TCM-S
>
> | Method          | Steps-to-Adam ↓ | Time-to-Adam ↓ | Kodak      | Tecnick    | CLIC2022   | Avg.       |
> | --------------- | --------------- | -------------- | ---------- | ---------- | ---------- | ---------- |
> | Adam            | 1               | 1              | 0%         | 0%         | 0%         | 0%         |
> | AuxT [1]        | 0.43            | 0.46           | -1.11%     | -1.24%     | -1.66%     | -1.34%     |
> | CMD-LIC         | 0.49            | 0.50           | -0.47%     | -0.55%     | -0.68%     | -0.57%     |
> | Balanced-RD     | 0.67            | 0.81           | -1.37%     | -1.91%     | -1.87%     | -1.71%     |
> | SOAP            | 0.28        | 0.39       | -2.86%     | -2.40%     | -3.01%     | -2.76%     |
> | **AuxT + SOAP** | **0.23**        | **0.35**       | **-2.97%** | **-2.53%** | **-3.22%** | **-2.91%** |
>
> **Training conditions:** 1× NVIDIA H100 GPU, 2× Intel Xeon Platinum 8480+ CPU, 1 TB RAM.
>  **Bold** indicates best performance. The “Avg.” column reports mean BD-Rate across Kodak, Tecnick, and CLIC2022.
>
> ------
>
> **Reference**
>
> [1] Li, H., Li, S., Dai, W., Cao, M., Kan, N., Li, C., ... & Xiong, H. (2025). *On disentangled training for nonlinear transform in learned image compression.* arXiv:2501.13751.

---

> ### Author Response · Authors · 2025-11-21
> **Response to Reviewer xcTm (4)**
>
> **Q3. Are the preconditioners reused across layers, or is SOAP applied per-layer independently?**
>
> **Response.**
>  In our experiments, SOAP is applied **per layer**, and the preconditioners are **not** reused across layers.
>
> Concretely, we follow the standard SOAP implementation (Vyas et al., 2024), where each matrix-shaped parameter $W \in \mathbb{R}^{m \times n}$ (i.e., each layer) maintains its own Kronecker-factored curvature estimates $L \in \mathbb{R}^{m \times m}$ and $R \in \mathbb{R}^{n \times n}$, together with its own Adam-style moment statistics in the rotated basis. Appendix A.7 in our paper explicitly assumes a **layerwise** Kronecker structure, and all subsequent analysis is performed at the layer level, ignoring inter-layer curvature.
>
> Thus, the “shared preconditioner’’ discussed in Sec. 4 refers to sharing between the rate and distortion components **within the same layer**, not across different layers.
>
> In practice, this means each layer maintains its own preconditioner state
>  $(L, R, Q_L, Q_R, \text{moments})$, and these states are never shared across layers. They are only reused temporally for the **same** layer, with the preconditioner updated every $f = 10$ optimization steps as described in Sec. 3 and Appendix A.6.

---

> ### Author Response · Authors · 2025-11-26
>
> Dear Reviewer xcTm,
> As the discussion phase concludes next week, we wanted to briefly check whether you had any further feedback on our revisions.
> In particular, we have addressed your suggestions on stronger baselines and conceptual framing:
>
> *Comparison with other optimizers (Appendix A.15).* We added experiments comparing SOAP against Shampoo, SPlus, Adagrad, and discussed our attempts with Muon. Across these baselines, SOAP is the most effective optimizer for the R–D landscape, while others (such as Shampoo) face stability or efficiency issues.
>
> *Additivity (Appendix A.3).* Following your suggestion, we combined SOAP with AuxT. This AuxT+SOAP variant achieves our best results (average BD-Rate (-2.91%)), showing that the benefits are additive to existing acceleration techniques.
>
> *Curvature assumption.* We had empirical support for the shared-curvature structure via the update-vector alignment measurements in Section 4.4 and Figure 3(a).
>
> We hope these help clarify and strengthen the contribution. Please let us know if there are any remaining questions we can address.
> Best regards,
> The Authors

---

### Author Response · Authors · 2025-11-29
**Summary of Contributions and Rebuttal Outcome for the Area Chair (part 1)**

Dear Area Chair,

In light of the recent OpenReview incident, review revert, and reassignment, we provide this summary of our submission, *“Towards Second-Order Optimization in Learned Image Compression”*, and how we addressed reviewer concerns during the discussion period.



### **1. High-Level Contribution**



Our work studies optimization for the Rate–Distortion (R–D) objective in learned image compression (LIC), showing that an efficient second-order optimizer (SOAP) improves both training dynamics and deployability.

Our main contributions are:

1. **R–D gradient conflict analysis and Newton preconditioning.**

   We view the R–D loss as a multi-objective problem with conflicting gradients between rate and distortion. We show theoretically (Sec. 4 and appendices) that Newton-style preconditioning, as implemented by SOAP: (1) aligns the preconditioned rate and distortion updates within a step (high intra-step cosine similarity), (2) stabilizes updates across steps (high inter-step similarity), transforming “destructive interference” into “constructive interference” and leading to faster, more stable convergence.

2. **Quantization-friendly latent and activation statistics.**

   We develop a “conservation of correlation energy” view (Sec. 5.2) that explains why diagonal optimizers (Adam) tend to accumulate energy into diagonal terms (high kurtosis/outliers), whereas curvature-aware updates redistribute energy across channels. Empirically, SOAP reduces latent and activation kurtosis and outlier measures, and improves robustness to W8A8 post-training quantization by about 2% BD-Rate across models.

3. **Efficiency and performance across strong LIC baselines.**

   Across four advanced LIC models (ELIC, TCM-S, LALIC, DCAE), replacing Adam with SOAP as a drop-in optimizer yields approximately: (1) 70% fewer training steps (Steps-to-Adam around 0.25–0.35), (2) 58% reduction in wall-clock time (Time-to-Adam around 0.35–0.49), and (3) 2–4% BD-Rate improvement after convergence, with no changes to architecture or inference-time complexity.



### **2. Novelty: Beyond “Just Applying an Optimizer”**



Reviewers (Nu8g, xcTm) raised the question of whether this work is “just applying SOAP” to LIC. During the discussion, we clarified that the main novelty lies in R–D–specific analysis and in connecting optimization to quantization robustness.

**R–D gradient alignment theory.**

Starting from the quasi-Newton interpretation of SOAP, we derive results specific to the R–D objective: (1) Lemmas and propositions show that Newton-preconditioned steps have strong inter-step alignment, explaining the empirically smooth trajectories. (2) Propositions show that under reasonable structural assumptions, the Newton-preconditioned rate and distortion updates become asymptotically colinear near an optimum, resolving R–D conflicts in the preconditioned space. (3) We introduce intra- and inter-step cosine scores and prove that higher intra-step and inter-step alignment increases a lower bound on the loss decrease. This directly links gradient alignment to optimization efficiency.

These analyses do not appear in the original SOAP paper; they are developed to explain R–D optimization behavior.

**Latent outliers and PTQ.**

In Sec. 5 and the appendices, we introduce a new link between the optimizer and latent/activation statistics: (1) A “conservation of correlation energy” identity decomposes feature statistics into diagonal and off-diagonal components. (2) A small-step expansion shows that SOAP yields a smaller upper bound on kurtosis growth than diagonal methods, explaining why it suppresses outliers. (3) We track kurtosis and a MaxMed metric throughout training, showing that SOAP prevents early spikes and stabilizes at a much lower plateau than Adam. This directly improves W8A8 post-training quantization.

---

### Author Response · Authors · 2025-11-29
**Summary of Contributions and Rebuttal Outcome for the Area Chair (part 2)**

### **3. Baselines, Convergence, and Additional Experiments from the Discussion**



Reviewers 8Sdb and DAjx raised questions about the strength of baselines, convergence, and alternative optimizers. We added several ablations in response.

**Extended training and schedulers.**

For ELIC, we extended Adam training from 300 to 1000 epochs and additionally tested a “Half Constant + Cosine” learning-rate schedule. The best Adam variants improved BD-Rate by at most 0.10% relative to our baseline. In contrast, SOAP achieves about a 3.5% BD-Rate improvement within 300 epochs. This indicates that SOAP reaches a better solution, not merely the same solution faster.

**Other optimizers.**

We compared against: (1) AdamW with weight decay: tuned variants perform similarly to or slightly worse than Adam; with optimal settings, Adam and AdamW effectively coincide. (2) Adagrad and SGD: stable but clearly worse in R–D performance, underscoring the need for adaptive/curvature-aware methods. (3) Shampoo/SPlus: converge somewhat faster than Adam but remain less efficient and less stable than SOAP. The main difference is that Shampoo approximates $H^{-1/2}$, whereas SOAP approximates $H^{-1}$, which is more effective for resolving R–D conflicts. (4) Muon: difficult to apply to 4D convolutional kernels in LIC; in our trials, we could not obtain stable convergence despite tuning.

**Practical robustness checks.**

To address concerns about practicality and robustness: (1) We repeated experiments with batch size 32, showing that SOAP’s gains persist on smaller, consumer-grade GPU configurations. (2) We performed EMA ablations, showing that SOAP is substantially more stable than Adam even without EMA; EMA still gives modest gains for both, at negligible extra cost. (3) We analyzed the effect of SOAP’s preconditioner update frequency and discussed boundary cases where less frequent updates improve numerical stability. (4) We added a preliminary learned video compression experiment (DCVC), indicating similar qualitative benefits for video R–D optimization.



### **4. Reviewer Trajectory and Current Status**



**Positive reviewers (scores 8, 6, 6).**

Reviewers DAjx, xcTm, and RnuL consistently emphasized: (1) strong empirical improvements in both efficiency and BD-Rate across multiple LIC models, (2) clear and well-structured exposition, (3) useful analysis via intra-/inter-step scores and the outlier/PTQ study.

Their questions about longer training and additional baselines were addressed by the new ablations; they did not raise any remaining major concerns during the discussion.

**Reviewer 8Sdb (initially 4, then explicitly improving).**

Reviewer 8Sdb initially raised doubts about: (1) fairness and convergence of the comparisons, (2) robustness to different R–D trade-offs, batch size, EMA, and AdamW, (3) depth of the outlier analysis.

After we added the 1000-epoch Adam ablation, small-batch experiments, AdamW comparisons, outlier-evolution plots, and discussion of preconditioner frequency, they wrote that our explanation and ablations “further verified the effectiveness of SOAP for LIC training, solving most of my problems” and explicitly stated that they would improve their score.

**Reviewer Nu8g (score 2).**

Reviewer Nu8g’s main concerns were: (1) limited novelty beyond the SOAP paper, (2) improvement over Zhang et al. (2025c), (3) the practical importance of training speedup, and the reason for outlier reduction.

In the discussion, we clarified that: (1) Sec. 4 and Sec. 5 introduce R–D–specific theory and analysis not present in the original SOAP paper, (2) Our method shows significantly better results than Zhang et al. (2025c) (See A.3 COMPARISON WITH OTHER METHODS) (3) training cost is a major bottleneck in modern LIC (hundreds of H100 hours per R–D curve), so a roughly $2\times$ reduction in training cost is practically significant. Our outlier reduction is supported by both theory (correlation-energy view and kurtosis bound) and experiments, and is directly linked to W8A8 PTQ improvements.

The reviewer did not respond further after these clarifications. We believe these concerns are substantively addressed in the paper and rebuttal.



### **5. Conclusion**



In summary, this work proposes a change in optimization strategy for learned compression: using efficient second-order quasi-Newton updates via SOAP as a drop-in replacement for Adam. This leads to: (1) substantially faster training (about 70% fewer steps and approximately 58% less wall-clock time), (2) consistently better R–D performance (about 2–4% BD-Rate gain), (3) more quantization-friendly latent and activation statistics, improving W8A8 post-training quantization robustness.

These claims are supported by extensive experiments across four strong LICs and by R–D–specific theoretical analysis. We hope this overview is helpful for your assessment under the reverted-review setting.

Sincerely,

The Authors

---

### Meta-Review · Area_Chair_pzcU · 2026-01-06

**Summary:**

One of the key concern is novelty (xcTm, Nu8g, DAjx, 8Sdb), several reviewers point out that using a second-order optimizer is not new in deep learning, the main contribution is the application of an existing second-order optimization method (SOAP) to learned image compression (LIC). To address this concern, the authors point out they derive results specific to the R–D objective, showing that the Newton-preconditioned rate and distortion updates become asymptotically colinear near an optimum (assuming that the rate and distortion Hessians are locally proportional or jointly diagonalizable), resolving R–D conflicts in the preconditioned space (PROPOSITION 1). The in the added Proposition 5&6, the authors show this alignment helps optimization efficiency though rate and distortion are intrinsically conflicting objectives. However, similar insights and derivations already exist in Wang, Sifan, et al., 2025c: "Gradient alignment in physics-informed neural networks: A second-order optimization perspective." arXiv preprint arXiv:2502.00604 (2025), which appears on arxiv on 2 Feb 2025 and is later publised in NeurIPS 2025. I understand the policy on arxiv paper and concurrent work. However, this paper cites Wang, Sifan, et al., 2025c but did not discuss the conection, which may need another round of review if the authors had discussed the correlation. It is currently not clear to me which part is specific for R-D in LIC which cannot be covered by Wang, Sifan, et al., 2025c.



Another major concern is the importance of training speedup for LIC (Nu8g). The AC has the same doubt to some extent, especially when the proposed solution is an existing optimizer SOAP. It is good to see models trained with SOAP are substantially more robust to post-training quantization (PTQ) than those trained with first-order optimizers, however, this is known in previous works that SOAP decreases feature kurtosis and suppresses activation outliers [1]. More important thing is what will happen if we train longer (Nu8g, DAjx, 8Sdb) as the R-D Loss curve shown in Figure 1for both Adam and SOAP are still decreasing in a sharp way.




[1] He, Bobby, et al. "Understanding and minimising outlier features in transformer training." Advances in Neural Information Processing Systems 37 (2024): 83786-83846.

**Reviewer Concerns:**

Addressed:
Comparison with Shampoo.
Empirically verify the shared curvature structure assumption.
The comparison with Zhang et al. (2025c) (Balanced R–D).
The core section (Section 4.2) mainly reproduces the SOAP derivation.
Why the authors employed AdaRound in the experiments of Section 5.3.
Some other concerns on writing and clarity.
"step-efficiency" and "time-efficiency" are flaw metrics.
Why having a larger cosine similarity translates to a good thing in parameter updating?
A large number of "See Appendix" jumps result in poor readability and logical fragmentation.
The presentation of training settings and testing conditions, as well as ablation experiments.



Still outstanding:
Novelty is limited as the manuscript mainly applies existing second-order optimizer SOAP to learned image compression (xcTm, Nu8g, 8Sdb, DAjx).
Comparison with Muon, AdaHessian (xcTm).
Whether such a training speedup is actually important for learned image coding (Nu8g).
Proposition 1 assumes that the rate and distortion Hessians are locally proportional or jointly diagonalizable — a strong assumption that is unlikely to hold in practice for deep networks (xcTm).

**Reviewer Scores:**

xcTm may keep 6 or increase.
Nu8g may keep 2 as Nu8g doubts the novelty and the relevance of training speedup.
RnuL may keep 6.
8Sdb may increase the score as stated, maybe from 4 to 6.
DAjx may keep 8.

---

### Decision · Program_Chairs · 2026-01-26

Reject